# MIDSTEER: Optimal Affine Framework for Steering Generative Models

## Abstract

The idea of steering intermediate representations of generative models has recently emerged as a simple yet powerful approach for controlling aspects of generated texts and images. However, despite the simplicity of the approach, no theoretical framework has yet been built around steering. In this paper, we aim to bridge this gap, building theory around concept steering. First, we provide a theoretical link between steering and affine concept erasure, showing that the widely used steering approach for erasing unwanted behaviours or concepts from generative models is a special case of LEACE, a recently proposed closed-form method for affine concept erasure in neural networks. Next, we consider the task of concept switching, the aim of which is to change information about an unwanted concept or behaviour in the model's representations into another, more desired concept or behaviour. Here our contribution is two-fold: first, we formulate a theoretical framework for this task, adapting the existing affine concept erasure framework used for concept erasure. Then, we identify weaknesses of the resulting framework, and propose a new, improved one, that we call **MiDSteer** (**Mi**nimal **D**isturbance concept **Steer**ing). Our results show that MiDSteer performs favourably on a variety of tasks, modalities, and models, including vision diffusion models and LLMs.

## 1 Introduction

Generative models such as Large Language Models (LLMs) and vision diffusion models have achieved remarkable progress in recent years Yang et al. (2024b) Naveed et al. (2023). However, controlling model outputs to enforce desirable behaviors or suppress harmful ones remains challenging Bartoszcze et al. (2025). Yet, this capability is necessary for improving model safety, reliability, alignment, and usefulness in downstream applications.

Concept steering of intermediate representations is an increasingly popular technique that has already proven to be simple yet powerful for controlling behaviour in LLMs. Recently it was also shown to be applicable to vision diffusion models Gaintseva et al. (2025). The underlying idea is to change the intermediate representations of a generative model during generation by adding or subtracting a "steering vector" that encodes a target concept. This approach has proven effective for tasks such as erasing unwanted behaviors (toxicity, nudity) or amplifying desirable features (helpfulness, truthfulness). However, despite the simplicity of the approach, its theoretical foundations remain underdeveloped with most of the work around it being highly empirical. Existing methods largely rely on heuristic vector manipulations, which can introduce unintended side effects and lack solid theoretical basis and guarantees. Moreover, naive steering often perturbs unrelated features, undermining the minimal disturbance principle that is critical to maintaining model quality and coherence.

Recently, strong theoretical foundations have been developed around concept erasure. Ravfogel et al. (2023) introduced the notion of log-linear guardedness. Based on this work, Belrose et al. (2025) developed LEACE, an affine concept erasure framework to remove undesired information from model representations for downstream tasks. However, these methods do not naturally extend to other forms of concept manipulation—such as switching—where the goal is to introduce a desired attribute or replace one with another.

In this work, we address these gaps by developing a unified theoretical framework for affine steering of generative models. We first show that the widely used steering method for concept deletion is a

special case of LEACE. We then define the task of concept switching, and show how to extend LEACE to this setting. Then we identify limitations of such approach and close this gap by introducing MidSteer (Minimal Disturbance Concept Steering), a new theoretical framework designed to achieve precise concept switching while minimizing interference with other properties of the representation.

Through experiments with LLMs and vision diffusion models, we demonstrate that MidSteer achieves more reliable concept switching than prior methods, allowing controllable generation with minimal side effects. Our results highlight the value of grounding steering methods in theory and provide practical tools for aligning generative models with desired behaviors.

In summary, our contributions are as follows:

- We **provide the first theoretical connection** between two widely used methods for concept erasure in generative models.
- We **introduce a formal framework for concept switching** by directly extending the existing affine concept erasure approach to enable the controlled replacement of an undesired attribute with a desired one.
- We **unify approaches to both tasks** by introducing MidSteer, a new minimal-disturbance steering framework designed to enable precise concept manipulation with provably minimal interference to unrelated directions in representation space.
- We **empirically validate MidSteer** across tasks, modalities, and architectures—including LLMs and vision diffusion models—showing improved controllability and reduced side effects compared to prior steering and erasure methods.

## 2 RELATED WORK

**Steering generative models.** Early work in LLMs demonstrated that adding or subtracting steering vectors derived from contrasting prompts can provoke or suppress targeted concepts such as sentiment, bias, or style from appearing in the generated text Turner et al. (2023). Later, Rimsky et al. (2024) proposed to construct steering vectors of concepts based on the mean activation difference, which was then established as an effective approach to steering by Bartoszcze et al. (2025) and Zou et al. (2023). Multiple approaches based on steering have since been developed to effectively steer LLMs representations from undesired behaviours or towards desired ones for different tasks. Wang et al. (2025) adaptively adjust steering intensity to improve truthfulness, Stickland et al. (2024) proposed KL-then-steer, a fine-tuning technique that decreases the side effects of steering applied to an LLM, Lu & Rimsky (2024) used steering for bias mitigation, Scialanga et al. (2025) showed that activation steering can be applied to knowledge editing, Rahn et al. (2024) applied steering to LLM agents.

**In vision generative models**, similar approaches to LLM steering have been developed to control image generations. Kwon et al. (2023) Park et al. (2023), Si et al. (2024), Tumanyan et al. (2023) focus on finding interpretable directions in various intermediate spaces of diffusion models, which can then be used to control the semantics of generated images. SDID Li et al. (2024) constructs learnable concept vectors, which are then added to intermediate activation of a bottleneck layer of the diffusion model during inference to control the level of this concept in generated images. However, these methods rely on different methods of constructing steering vectors than what is used in LLM, in most cases their steering vectors are learnable. Recently, Gaintseva et al. (2025) proposed to apply the steering of cross-attention outputs of the diffusion models to control generation, in a similar manner to the way it was done in LLMs, with steering vectors constructed using mean differences of intermediate activations of the diffusion model.

**Affine concept erasure.** One related line of research seeks to use affine transformations to remove concepts from representations altogether. Early methods such as INLP Ravfogel et al. (2020) iteratively trained linear classifiers to project out protected attributes like gender or sentiment. More recently, Ravfogel et al. (2023) introduced log-linear guardedness as a theoretical basis for concept erasure, formalising the conditions under which a representation can be considered free of a concept. Building on this, Belrose et al. (2025) proposed LEACE, a closed-form solution for affine concept erasure that minimizes disturbance to the representation space. Later, Holstege et al. (2025), Singh et al. (2024) extended this framework for preserving task-relevant information under concept removal. These advances provide rigorous guarantees for concept deletion, but they do not directly address

the broader goal of concept switching. In our work, we bridge this gap by extending LEACE to the framework of optimal concept switching, and build a new theory-based framework of optimal concept switching.

## 3 PRELIMINARIES

### 3.1 STEERING INTERNAL REPRESENTATIONS OF MODELS

In this section we formalise the widely used idea of manipulating internal model representations to influence the presence of a certain concept $c$ in the model output. It works by adding or subtracting a fixed steering vector $s_c$ to or from the intermediate activations of a model during inference.

The steering vector $s_c$ is usually constructed by collecting the neural activity from a specific part of a network based on paired sample of input stimuli. Formally, let $h$ be a random vector in $\mathbb{R}^d$ representing the activity at a particular layer $l$, and $C$ be a random binary variable taking values in $\{0, 1\}$ representing the presence or absence of a concept $c$. Then $s_c \in \mathbb{R}^d$ is a difference of concept-conditional means of the internal representation:

$$s_c = \mathbb{E}[h|C = 1] - \mathbb{E}[h|C = 0], \tag{1}$$

$s_c$ can be optionally post-processed, e.g. normalized to have unit norm.

Now let us define the steering intervention $f$ that controls the expressiveness of the concept $c$ in the model output during inference. Let $\alpha \in \mathbb{R}$ be a parameter controlling the steering strength and direction. Omitting subscript $c$ for clarity $s = s_c$,

$$f(h, s) = h + \alpha s \tag{2}$$

Now we highlight two special cases of steering, varying in the choice of $\alpha$.

**Concept Deletion**: In this case we aim to prevent any information about the concept $c$ to be present in the activation vector $h$. The dot product is used as an estimation of the amount of concept $c$ in current activation vector $h$:

$$f_{\text{delete}}(h, s) = h - \langle h, s \rangle s \tag{3}$$

**Concept switch**: Now note, that if we use 2 as the multiplier in the dot product, we get Householder reflection of the vector $h$ across the hyperplane orthogonal to $s$:

$$f_{\text{switch}}(h, s) = h - 2\langle h, s \rangle s \tag{4}$$

In this case, the resulting transformation substitutes the representation of concept $c$ with a representation of its absence (i.e., representation opposite to $c$).

If steering is applied to the outputs of self-attention layers in an LLM or cross-attention layers of a vision diffusion model, it is possible to incorporate Eq. 3 and Eq. 4 into weight matrices of the model, thus achieving zero inference overhead. This is a desirable property for any large-scale application of steering. Refer to Sec.7.2 in supplementary for more details.

### 3.2 AFFINE GUARDEDNESS FRAMEWORK

For the task of concept erasure from model representations, Ravfogel et al. (2023) introduced the notion of log-linear guardedness. Belrose et al. (2025) generalized it to the following:

**Definition 1** (Guardedness). *Consider a $k$-class classification task over jointly defined random vectors $X$ (the input data) and $Z$ (the one-hot labels), with $X$ of finite first moment and taking values in $\mathbb{R}^d$, and $Z$ taking values in $\mathcal{Z} = \{\mathbf{z} \in \{0, 1\}^k \mid \|\mathbf{z}\|_1 = 1\}$[1] with each $\mathbb{P}(Z = j) > 0$. Let $\eta(\cdot; \boldsymbol{\theta}) : \mathbb{R}^d \to \mathbb{R}^k$ be a predictor chosen from a function class $\mathcal{V} = \{\eta(\cdot; \boldsymbol{\theta}) \mid \boldsymbol{\theta} \in \Theta\}$ (presumed to contain all constant functions) so as to minimize the expectation $\mathbb{E}[\mathcal{L}(\eta(X), Z)]$ of some $\mathcal{L} : \mathbb{R}^k \times \mathcal{Z} \to [0, \infty)$ in a class $\mathfrak{L}$ of loss functions. Let $\chi$ be the set of all random vectors of finite first moment taking values in $\mathbb{R}^d$, jointly defined with $Z$.*

---

[1] Integer $j \le k$ is used to refer to the element of $\mathcal{Z}$ which is 1 at the $j^{\text{th}}$ index and 0 elsewhere.

*We say $X$ $(\mathcal{V}, \mathfrak{L})-$**guards** $Z$ if, for all losses $\mathcal{L} \in \mathfrak{L}$, it maximizes the minimum expected loss:*

$$X \in \underset{X' \in \chi}{\mathrm{argmax}} \; \inf_{\boldsymbol{\theta} \in \Theta} \; \mathbb{E}\Big[\mathcal{L}(\eta(X'; \boldsymbol{\theta}), Z)\Big].$$

*In other words, its conditional distribution $\mathbb{P}(X \mid Z = \cdot)$ is among the worst possible distributions for predicting $Z$ from $X$ using a predictor of the form $\eta(\cdot; \boldsymbol{\theta}) \in \mathcal{V}$ and a loss function in $\mathfrak{L}$.*

Building on Definition 1, note that guardedness characterizes inputs whose conditional distributions make the target $Z$ maximally unpredictable under a given model class and loss. In particular, for linear–affine predictors and squared loss, Belrose et al. (2025) show that this worst-case unpredictability is achieved precisely when the representation is uncorrelated with the concept, i.e., when $\mathrm{Cov}(h, Z) = 0$. Thus, enforcing guardedness in a linear representation amounts to removing all linear statistical dependence between $X$ (or an internal representation $h$ derived from it) and $Z$.

**Theorem 2** (Belrose et al.). *The following statements are equivalent:*

- *The data X linearly guards the labels Z (Def. 1)*

- *Every component of $X$ has zero covariance with every component of $Z$: $\mathrm{Cov}(X, Z) = 0$.*

At the same time, when erasing concepts from model representations, we typically wish to alter the representations as little as possible, so that downstream performance on unrelated tasks is preserved. Thus, it is natural to seek the mildest affine transformation of $X$ that achieves this covariance-removal property.

Guided by these two principles — (i) guardedness corresponds to zero covariance, and (ii) we prefer minimal deviation from the original representation — Belrose et al. (2025) prove the following result:

**Theorem 3** (LEACE, Belrose et al.). *Let $X$, $Z$ be random vectors taking values in $\mathbb{R}^d$ and $\mathbb{R}^k$ respectively, each of finite second moments. Define $\Sigma_{XX} = \mathrm{Cov}(X, X) \in \mathbb{R}^{d \times d}$ and $\Sigma_{XZ} = \mathrm{Cov}(X, Z) \in \mathbb{R}^{d \times k}$. Assume $\mathrm{Im}(\Sigma_{XZ}) \subseteq \mathrm{Im}(\Sigma_{XX})$. The following optimization problem:*

$$\min_{\substack{A \in \mathbb{R}^{d \times d} \\ b \in \mathbb{R}^d}} \mathbb{E}\Big[\|AX + b - X\|_2^2\Big] \quad s.t. \quad \mathrm{Cov}(AX + b, Z) = 0 \tag{5}$$

*has the following solution (almost surely):*

$$\widehat{A} = I - W^+(W\Sigma_{XZ})(W\Sigma_{XZ})^+ W, \tag{6}$$

$$\widehat{b} = \mathbb{E}[X] - \widehat{A} \cdot \mathbb{E}[X], \tag{7}$$

*where $W = (\Sigma_{XX}^{1/2})^+$ is whitening transformation.*

Here and later we use $A^+$ to denote the pseudo-inverse of (any) matrix $A$ and $A^{1/2}$ to denote the square root of a positive semi-definite symmetric matrix $A$, i.e. for the singular value decomposition $A = VSV^\top$ with orthonormal matrix $V$ and diagonal matrix $S$ with non-negative singular values on the diagonal, the matrix $A^{1/2}$ is defined as $A^{1/2} := VS^{1/2}V^\top$, where the square root of the diagonal entries of $S$ is computed.

## 4 THEORETICAL RESULTS

In this section, we theoretically analyse connections between steering approaches (sec. 3.1) and LEACE (sec. 3.2), and derive a novel framework for affine concept steering called MidSteer, that unifies and generalizes all these approaches.

This section is organized as follows: First, we consider the task of concept erasure and derive a connection between steering setup for erasure and LEACE. In particular, we show that Eq. 3 is a special case of LEACE. Next, we extend LEACE for the task of optimal affine concept switching and show that Eq. 4 is a special case of the newly proposed framework. Finally, we identify limitations of LEACE-based approach to concept switching, and improve the proposed framework by restricting it to only affecting one concept. We derive MiDSteer, an affine optimal concept manipultion framework. In experimental section, we show that MiDSteer outperforms vanilla steering and LEACE in both LLMs and Vision Generative Diffusion Models, enabling precise concept switch while leaving other features of the images or texts intact.

## 4.1 CONNNECTION BETWEEN STEERING IN ERASURE MODE AND LEACE

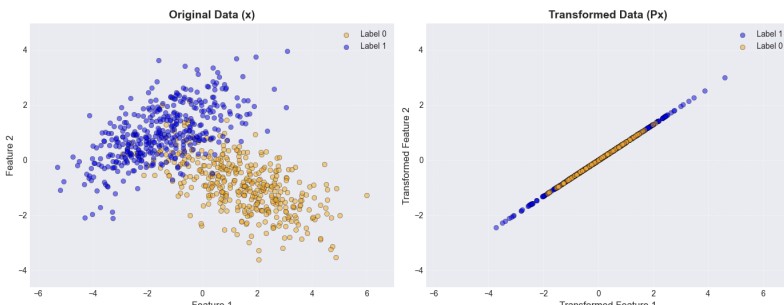

(a) Illustrative example of affine concept erasure. In this case the affine transformation is satisfying $\mathrm{Cov}(AX + b, Z) = 0$. This figure is inspired by Belrose et al. (2025)

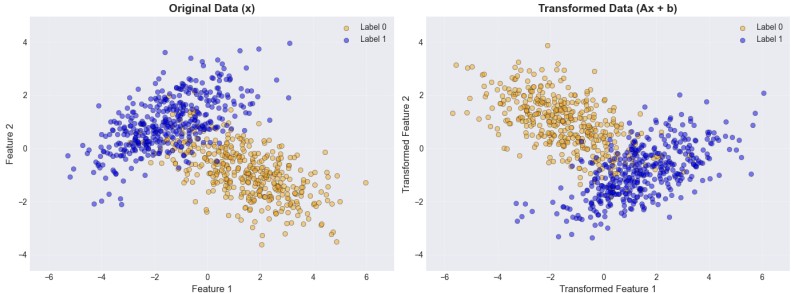

(b) Illustrative example of affine concept switching. In this case the affine transformation is satisfying $-\mathrm{Cov}(AX + b, Z) = \mathrm{Cov}(X, Z)$.

Figure 1: Illustrative example of affine concept erasure and affine concept flipping frameworks

In this section, we show that steering in deletion mode (3) is a special case of LEACE.

**Theorem 4.** *Let $X$ be a standardized random vector in $\mathbb{R}^d$, i.e. it has zero mean $E[x] = 0$ and unit covariance matrix $\Sigma_{XX} = I$. Let $C \in \{0, 1\}$ be a concept indicator variable. Let $s$ be defined as in equation 1. Let $f_{delete}$ be defined as in Eq. 3. Then $f_{delete}$ as a function of $h$ minimizes*

$$\min_{f \in \mathrm{Aff}(\mathbb{R}^d \mapsto \mathbb{R}^d)} \mathrm{E}[\|f(X) - X\|^2] \quad s.t. \quad \mathrm{Cov}(f(X), C) = 0 \tag{8}$$

This theorem states that steering in deletion mode can be seen as LEACE under the assumptions that the whitening matrix is identity and mean of all vectors is zero. Proof is found in appendix 7.4.1.

## 4.2 CONCEPT SWITCHING

In this section, we generalize beyond a single binary concept, and consider the task of transforming the representation of one concept $c_1$ into another $c_2$. We call this task of *concept switching*.

More formally, let $c_1$ and $c_2$ denote two distinct concepts, each associated with a subset of the data distribution, and their corresponding binary indicators $C_1, C_2 \in \{0, 1\}$. The goal of a *concept switch* operation is to construct a transformation $f$ such that samples exhibiting concept $c_1$ after transformation exhibit concept $c_2$, while preserving all other factors of variation as much as possible. Formally, this requires modifying the dependence of the representation on $C_1$ and introducing a desired dependence on $C_2$, without enforcing any relationship between $C_1$ and $C_2$ outside their observed support.

In this section, we adapt LEACE 3 framework to the task of concept switching, and formulate a theoretical framework for the optimal affine concept switching, LEACE-Switch. Additionally, we show that steering in switching mode Eq. 4 can be seen as a special case of this framework. Then we identify limitations of LEACE-Switch and in the next section proceed to formulating a new framework, MidSteer, which addresses these limitations.

### 4.2.1 LEACE FOR CONCEPT SWITCHING

Recall that $Z \in \{0,1\}^k$ denotes a concept vector. For the task of concept switching, we formulate the constraint analogous to that of Theorem 3 in the following way:

$$\text{Cov}(f(X), \mathbf{1}^k - Z) = \text{Cov}(X, Z), \tag{9}$$

where $\mathbf{1}^k$ denotes a $k$-dimensional vector where every entry is one. Due to linearity of the covariance this is equivalent to

$$-\text{Cov}(f(X), Z) = \text{Cov}(X, Z).$$

Visualisation to this can be found in fig. 1b.

The constraint in Eq. 9 is the natural affine analogue of "switching" the concept label within the covariance-based guardedness framework. In LEACE (Theorem 3), concept erasure is obtained by enforcing $\text{Cov}(f(X), Z) = 0$, which guarantees that the transformed representation carries no linear signal about $Z$. For concept switching, however, the goal is different: rather than removing linear dependence on the concept, we want the transformed representation to encode the *opposite* linear signal. The label vector $\mathbf{1}_k - Z$ provides the natural formalization of such a concept flip. This guarantees that (i) the magnitude of the linear dependence on $Z$ is preserved, while (ii) its sign is reversed—meaning that samples that previously correlated positively with concept $c_1$ now correlate positively with concept $c_2$, and vice versa.

Analogous to Theorem 3, let us now characterise the optimal affine transformation that satisfies the proposed constraint.

**Theorem 5** (LEACE-Switch, Optimal Concept Switching). *Let $X, Z, \Sigma_{XX}, \Sigma_{XZ}$ be defined as in Theorem 3. Assume $\text{Im}(\Sigma_{XZ}) \subseteq \text{Im}(\Sigma_{XX})$. Then the optimization problem*

$$\min_{\substack{A \in \mathbb{R}^{d \times d}, \\ b \in \mathbb{R}^d}} \text{E}\left[\|AX + b - X\|_2^2\right] \ s.t. \ \text{Cov}(AX + b, Z) = -\text{Cov}(X, Z) \tag{10}$$

*has the following solution (almost surely):*

$$\widehat{A} = I - 2W^+(W\Sigma_{XZ})(W\Sigma_{XZ})^+W, \tag{11}$$

$$\widehat{b} = \mathbb{E}[X] - \widehat{A}\mathbb{E}[X], \tag{12}$$

Proof is found in appendix 7.4.2. Also note, since the transform is affine, it can be incorporated into the weight matrix of the model, thus achieving zero inference overhead.

We also show that vanilla concept switching as defined in Eq. 4 is a special case of Thm. 5.

**Theorem 6.** *Let $X$ be a standardized random vector in $\mathbb{R}^d$, i.e. it has zero mean $\mathbb{E}[x] = 0$ and unit covariance matrix $\Sigma_{XX} = I$. Let $C \in \{0,1\}$ be a concept indicator variable. Let $s$ be defined as in equation 1. Let $f_{switch}$ be defined as in Eq. 4. Then $f_{switch}$ as a function of $h$ minimizes*

$$\min_{f \in \text{Aff}(\mathbb{R}^d \mapsto \mathbb{R}^d)} \text{E}[\|f(X) - X\|^2] \quad s.t. \quad \text{Cov}(f(X), C) = -\text{Cov}(X, C) \tag{13}$$

Proof is found in appendix 7.4.3.

Basically, this theorem states that steering in switching mode can be seen as LEACE-Switch under the assumptions that the whitening matrix is the identity and the mean of all vectors is zero.

**Limitations.** While the constraint above captures the desired algebraic effect of inverting the linear signal of a concept, it also has two important limitations.

First, Eq. 9 relies on the assumption that the binary concept variable $Z$ partitions the dataset, i.e.,

$$P(Z = 1) + P(Z = 0) = 1.$$

This ensures that $\text{Cov}(X, Z)$ is well-defined across all samples. When considering two concepts $c_1$ and $c_2$ with corresponding indicators $Z_1$ and $Z_2$ that do not jointly cover the entire distribution, this assumption no longer holds. Therefore, Theorem 5 cannot be appplied and there is no guarantee that an optimal transformation exists.

Second, even when $Z$ does partition the dataset, the condition of eq. 9 enforces a complete inversion of the linear dependence on the concept label. This corresponds to flipping *all* samples with respect to the concept. In many practical applications — such as switching between a weak and strong attribute, or between partially overlapping concepts — it may be undesirable to invert the label for all samples in the dataset. Instead, one may wish to transform representations of $c_1$ into those of $c_2$ without forcing the reverse transformation.

These limitations motivate the need for a more flexible formulation of concept manipulation that does not require dataset-wide label inversion or full support of the concept labels. In the next section, we introduce such a generalization.

### 4.3 OPTIMAL AFFINE CONCEPT MANIPULATION

Now we aim to formulate constraint and corresponding theorem that would allow us to solve the task of concept switching, while avoiding limitations discussed above. Assume that $Z = (Z_1, Z_2)$, where $Z_1, Z_2 \in \{0,1\}^l$. $Z_1$ and $Z_2$ represent indicators of two groups of concepts $C_1 = \{c_1^{(1)}, \ldots, c_l^{(1)}\}$ and $C_2 = \{c_1^{(2)}, \ldots, c_l^{(2)}\}$. Our goal would be to have $\mathrm{Cov}(f(X), Z_1) = \mathrm{Cov}(X, Z_2)$, meaning for each $i$, the concept $c_i^{(1)}$ maps to to $c_i^{(2)}$. For reasons that are clarified in the proof, we will require $\mathrm{Cov}(X, Z_1)$ to be full rank. Now we formulate the following theorem:

**Theorem 7** (MidSteer, affine optimal concept manipulation). *Let $X, Z$ be defined as in Theorem 5 and assume $k = 2l$ for $l > 0$. Let $Z = (Z_1, Z_2)$, where $Z_1, Z_2 \in \{0,1\}^l$. Let $\Sigma_{XZ_i} = \mathrm{Cov}(X, Z_i), i \in \{1,2\}$ be the cross-covariance matrices between $X$ and $Z_i$. Assume $\mathrm{Im}(\Sigma_{XZ_i}) \subseteq \mathrm{Im}(\Sigma_{XX})$ for $i \in \{1,2\}$. Assume $\Sigma_{XZ_1}$ has full column rank: $\mathrm{rk}\left(\Sigma_{XZ_1}\right) = l$. Let $W = (\Sigma_{XX}^{1/2})^+$ be the whitening transformation. Let $\Sigma_{WX,Z_i} = W\Sigma_{XZ_i}$.*

*Then we have the following optimization problem:*

$$\min_{\substack{A \in \mathbb{R}^{d \times d} \\ b \in \mathbb{R}^d}} \mathrm{E}[\|AX + b - X\|_2^2] \quad s.t. \quad \mathrm{Cov}(AX + b, Z_1) = \mathrm{Cov}(X, Z_2) \tag{14}$$

*which has the following solution (almost surely):*

$$\widehat{A} = I + W^+(\Sigma_{WX,Z_2} - \Sigma_{WX,Z_1})\Sigma_{WX,Z_1}^+ W \tag{15}$$

$$\widehat{b} = \mathbb{E}[X] - \widehat{A}\mathbb{E}[X] \tag{16}$$

Proof can be found in sec. 7.4.4. Unlike LEACE, we now have two covariance matrices for each group of concepts, and do not require the labels to be flipped, but translated from one group to another.

**Connection between MidSteer and LEACE** Note that if $Z_2$ is constant indicator (e.g. representing no item in the dataset), then $\mathrm{Cov}(X, Z_2) = 0$ and MidSteer formula turns into concept erasure (LEACE). Thus we refer to MidSteer as *affine optimal concept manipulation*, as it generalizes several concept manipulation tasks (erasure, switch) under one framework.

### 4.4 STEERING STRENGTH

Let us now introduce the steering strength $\beta$ for erasure and switching. Note that in vanilla steering, we can unify eq. 3 and eq. 4in the following way:

$$f(h, s) = h - \beta \cdot \langle h, s \rangle s \tag{17}$$

For LEACE, $\widehat{b}$ is unaffected, and eq. 6 and eq. 11 become:

$$\widehat{A} = I - \beta \cdot W^+(W\Sigma_{XZ})(W\Sigma_{XZ})^+ W \tag{18}$$

Now, $\beta = 1$ represents concept erasure, $0 < \beta < 1$ represents lesser degree of erasure. $\beta = 2$ represents concept switching, and $\beta > 2$ refers to switching where the switched concept is more expressed than the base concept.

For MidSteer, we update eq. 15 as:

$$\widehat{A} = I + \beta \cdot W^{+}(\Sigma_{WX,Z_2} - \Sigma_{WX,Z_1})\Sigma_{WX,Z_1}^{+}W \tag{19}$$

$\beta = 1$ represents normal steering mode, $\beta < 1$ represent steering where $Z_2$ has less expression compared to what $Z_1$ had in original representation. Vice versa, $\beta > 1$ represent more expression for $Z_2$ compared to $Z_1$.

## 5 EXPERIMENTS

### 5.1 EXPERIMENTAL SETUP

We conduct experiments comparing MidSteer to vanilla steering and LEACE-Switch on the task of concept switching. We consider two modalities: text, for which we use Large Language Models (LLMs); and image, for which we use Vision Generative Models (diffusion models). Details on models considered in each case are given below.

**Evaluation setup** In each case, given a pair of concepts ($c_1$, $c_2$) to switch, we estimate class-conditional covariances $\Sigma_{XZ_1}, \Sigma_{XZ_2}$ based on a sample of data of size $N = 1000$, and self-covariances $\Sigma_{XX}$ on a sample of data of size $M = 50000$. We ablate the number of prompts needed for estimating $\Sigma_{XX}$ in Sec. 7.8. We provide pseudocode for the algorithm on covariance estimation in the appendix (sec. 7.1).

To test switching from the source concept $c_s$ to the target concept $c_t$, we use 80 template prompts prompting the model to generate output related to $c_s$ or $c_t$. For each prompt we run 10 such generations varying the random seed. We then run the generation on these prompts with and without steering. Templates for LLMs and diffusion models can be found in sec. 7.7. We then use *Concept Score (CS)* to estimate the amount of concepts $c_s$ and $c_t$ present in the model's output. In the case of ideal switching, CS for concept $c_2$ should be high, and CS for concept $c_1$ should be low for prompts related to both $c_s$ or $c_t$. We also generate outputs for four additional ("testing") concepts $\{c_i\}_{i=1}^4$ to test the capability of the method to preserve content not related to the concepts being switched. $\{c_i\}_{i=1}^4$ include semantically close and semantically far concepts from $c_s, c_t$. Based on $\{c_i\}_{i=1}^4$, we calculate CS to measure the amount of $c_i$ present in the generated output, which should ideally not lower when steering is applied.

We test switching of pairs of the following concepts $p_i = (c_s \to c_t)$: $p_1 = $ ("Horse" $\to$ "Motorcycle"), $p_2 = $"Dog" $\to$ "Cat", $p_3 = $"Chihuahua" $\to$ "Muffin"). Corresponding testing concepts are $t_i = \{c_i\}_{i=1}^4$: $t_1 = $ ("Cow", "Dog", "Pig", "Legislator"), $t_2 = $ ("Cow", "Wolf", "Pig", "Legislator"), $t_3 = $ ("Cat", "Dog", "Wolf", "Legislator").

Below we give details on metrics and models used in the case of LLMs and Diffusion Models.

**Details on LLM experiments** For LLMs experiments, we test on instruction-tuned Llama 2 Touvron et al. (2023) and Qwen 2.5 Yang et al. (2024a) models. We apply steering at every self-attention (SA) layer. SA activations corresponding to the last token in prompt are used for this process.

The dataset we used to estimate class-conditional covariances was obtained by prompting GPT o4-mini to generate various questions about each concept. Details of the prompt used and examples can be found in Sec. 7.5. To estimate $\Sigma_{XX}$, we use a sample of size $M$ from the Alpaca dataset Taori et al. (2023).

We calculate the *Concept Score (CS)* for a concept $c$ using LLM as a judge. More specifically, we use the Llama-3.1-8B-Instruct Dubey et al. (2024) model, and prompt it to estimate the amount of $c$ present in the generated output on a scale from 0 to 10. Prompts used are outlined in the appendix sec. 7.6.

We additionally test how much outputs for testing concepts differ with and without steering by calculating BERT scores Zhang et al. (2020) on the MMLU Hendrycks et al. (2021) dataset generations. Lower values of BERT Precision represent more change in the underlying output.

**Details on Diffusion Models experiments** For visual diffusion models, we test on SDXL Podell et al. (2024) and SANA Xie et al. (2025) models. Following recent work, we apply steering on activations of every cross-attention (CA) layer. CA activations corresponding to all images patches are used.

We use a sample of image captions from RELAION Schuhmann et al. (2022) to estimate both the class-conditional covariances $\Sigma_{XZ_i}$, and self-covariances $\Sigma_{XX}$. For $\Sigma_{XZ_i}$ we filtered the dataset to only captions that contain a specific word, mentioning the required concept.

We define *Concept Score (CS)* for a specific concept $c$ as the CLIP score Hessel et al. (2021) of $c$. Additionally, we measure the change between generations based on the testing concepts when steering is applied, by calculating FID Heusel et al. (2017) between images generated by steered model and vanilla model. Higher values of FID represent more change to the underlying image.

## 5.2 EXPERIMENTAL RESULTS

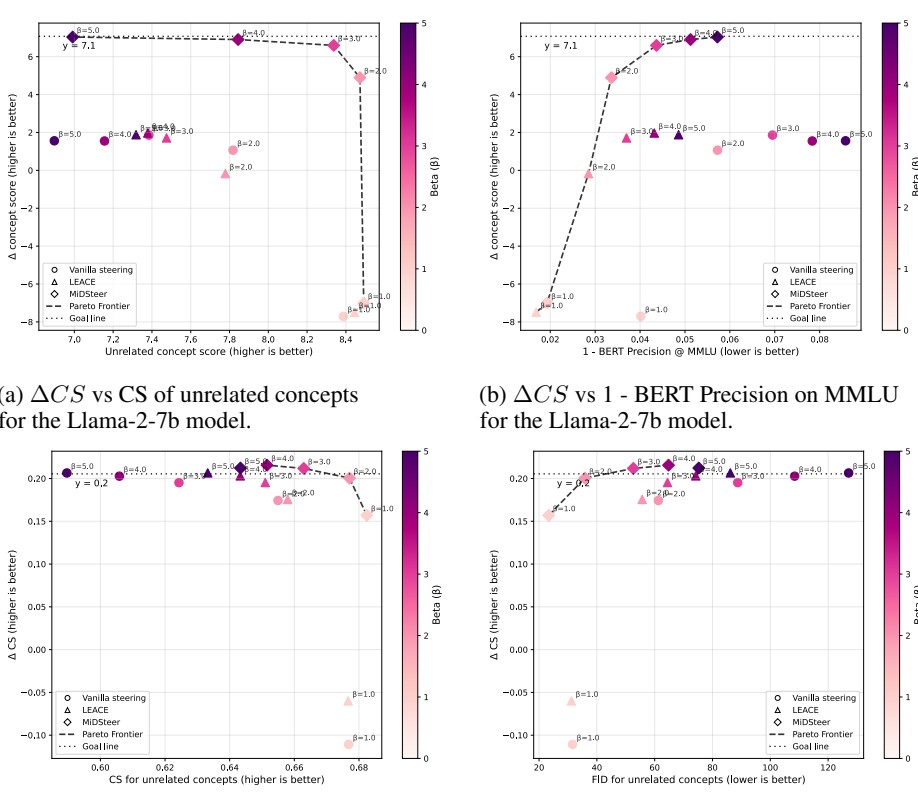

(a) $\Delta CS$ vs CS of unrelated concepts for the Llama-2-7b model.

(b) $\Delta CS$ vs 1 - BERT Precision on MMLU for the Llama-2-7b model.

(c) $\Delta CS$ vs CS of unrelated concepts for the SDXL model.

(d) $\Delta CS$ vs FID of unrelated concepts for the SDXL model.

Figure 2: Pareto efficiency frontiers for concept *switching* experiments with vanilla steering, LEACE and MiDSteer highlighting different values of $\beta$.

We compare results on concept switching by applying vanilla steering, LEACE-Switch and MidSteer with different values of $\beta$. We present results on LLama-2-7b and SDXL models aggregated for all three concept pairs $p_1, p_2, p_3$ in Fig. 2. It can be clearly seen, that in each case, MidSteer achieves much better balance between level of concept switch between $c_1$ and $c_2$ and preservation of other concepts across different values of $\beta$.

Pareto plots for other models, as well as Pareto plots for individual concept pairs can be found in the appendix sec. 7.9.1 and sec. 7.9.2.

To better illustrate differences between vanilla steering, LEACE and MidSteer for concept flipping, in Tab. 1 we also present results on switching a concept of $c_1$ = "horse" to $c_2$ = "motorcycle" on the SDXL model. We compare Vanilla switching and LEACE with $\beta = 2$ and MidSteer with $\beta = 1$, as these are default parameters for these methods as suggested by Eqs. 21, 11, and 15. The full table with results on all values of $\beta$ can be found in the supplementary. First note, that all the methods successfully flip "horse" to "motorcycle", having similar CS scores on source ("horse") and target ("motorcycle") concepts. Second, it can be seen that as suggested by definitions Eqs. 21, 11,

Table 1: Results on SDXL when flipping from "horse" to "motorcycle". Reported are CLIP-scores (cs) and FID for target and non-target concepts.

| method | strength | horse src-cs | horse tgt-cs | motorcycle src-cs | motorcycle tgt-cs | motorcycle fid | cow cs | cow fid | pig cs | pig fid | dog cs | dog fid | legislator cs | legislator fid |
|---|---|---|---|---|---|---|---|---|---|---|---|---|---|---|
| orig | - | 71.0 | 49.1 | 51.8 | 70.7 | - | 72.7 | - | 71.8 | - | 66.3 | - | 60.8 | - |
| CASteer | 2.0 | 52.1 | 69.5 | 68.3 | 52.9 | 212.4 | 70.9 | 42.7 | 71.9 | 18.9 | 66.1 | 28.6 | 60.9 | 24.6 |
| LEACE | 2.0 | 51.2 | 68.8 | 67.6 | 53.3 | 207.6 | 72.2 | 25.2 | 71.7 | 12.6 | 66.1 | 20.8 | 60.6 | 28.2 |
| MiDSteer (ours) | 1.0 | 51.2 | 68.7 | 51.9 | 70.7 | 12.7 | 72.2 | 23.9 | 71.8 | 12.4 | 66.1 | 20.7 | 60.7 | 27.2 |

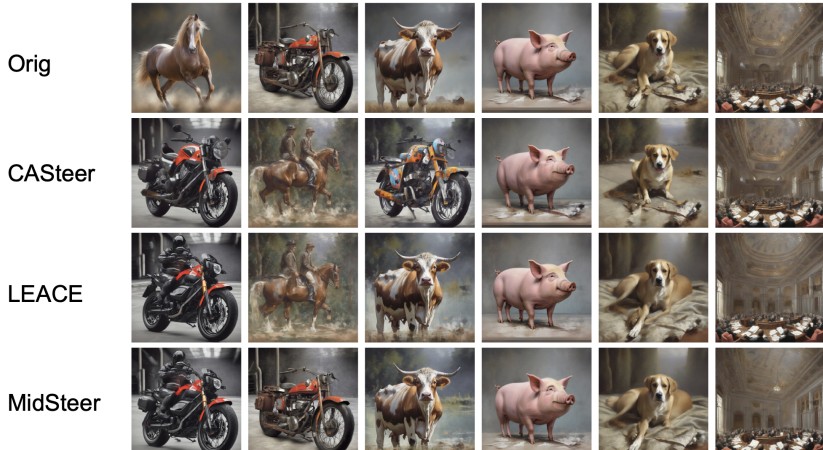

Figure 3: Qualitative results on switching to steer "horses" source concept into target "motorcycles". While all methods similarly successfully performed switching from "horse" to "motorcycle", CASteer and LEACE failed when presented with prompt for the target concept ("motorcycle"), since they do not distinguish between forward and reverse steering. CASteer also additionally failed on "cow" concept, and more significantly altered images of concept "dog"

vanilla steering and LEACE fail to keep the "motorcycle" concept intact when flipping "horse" to "motorcycle", as target CS score goes down. In contrast, MiDSteer keeps "motorcycles" intact. This is also illustrated in Fig. 3. Next, CS score of "cow" and FID scores of "cow", "pig" and "dog" are worse for vanilla than for other methods, showing superiority of LEACE and MiDSteer over vanilla steering in ability to keep unrelated concepts intact. Results on other concepts flipping on both LLMs and diffusion models show similar patterns.

We observe the same trend over all the concept pairs and models. Tables with detailed breakdown of scores per each concept and model are presented in the appendix, sec. 7.9.3 and sec. 7.9.4.

## 6 CONCLUSION

In this work, we bridge the gap between previous empirical research in steering generative models and the theory of affine concept steering. We extend this theoretical framework to concept switching. We define the corresponding optimisation problem and solve it in closed form.

We present MiDSteer, a universal steering method, that is theoretically optimal under certain conditions. It outperforms other methods on concept switching for both LLMs and image diffusion models, while having the advantage of clear matrix form representation. To our knowledge, this is the first theoretical treatment of steering beyond deletion, connecting empirical heuristics and principled affine methods.

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

# 7 APPENDIX

## CONTENTS

## 7.1   ALGORITHM FOR COMPUTING COVARIANCES

To estimate the covariances we use the algorithm by Welford (1962) on a sample of broad prompts (unrelated to the steering concepts). Given $X$ with the dimension of $batch\_size \times num\_heads \times seq\_len \times hidden\_dim$, the algorithm estimates the covariance matrix $\Sigma_X X$ of size $hidden\_dim \times hidden\_dim$. It does this by maintaining sample-level statistics of size $O(hidden\_dim^2)$ in memory and takes $O(batch\_size * seq\_len * hidden\_dim^2)$ time to update them for the output of a particular layer on a particular batch. In practice, estimating the covariances for 50,000 samples for SANA 1.6 finished in under 15 minutes, and for Qwen2.5 14B in under 30 minutes on our hardware setup.

In Algorithm 1 we provide pseudocode of the algorithm.

---

**Algorithm 1** Welford's Algorithm for Online Mean and Covariance Estimation

---

1:  **Input:** Stream of data batches $\{\mathbf{X}_1, \mathbf{X}_2, \ldots, \mathbf{X}_K\}$ where $\mathbf{X}_k \in \mathbb{R}^{h \times m_k \times d}$
2:  **Output:** Mean $\boldsymbol{\mu}$ and covariance $\boldsymbol{\Sigma}$ estimates
3:  **Initialization:**
4:  $n \leftarrow 0$                                         ▷ Total count of samples seen
5:  $\mathbf{M} \leftarrow \mathbf{0} \in \mathbb{R}^{h \times d}$                                        ▷ Running sum
6:  $\mathbf{S} \leftarrow \mathbf{0} \in \mathbb{R}^{h \times d \times d}$                            ▷ Running sum of squared differences
7:
8:  **for** $k = 1$ to $K$ **do**                                         ▷ Process each batch
9:      $m_k \leftarrow$ number of samples in batch $\mathbf{X}_k$
10:     **if** $n = 0$ **then**                                         ▷ First batch - initialize
11:         $n \leftarrow m_k$
12:         $\mathbf{M} \leftarrow \sum_{j=1}^{m_k} \mathbf{x}_{k,j}$                        ▷ Sum over samples
13:         $\boldsymbol{\mu} \leftarrow \mathbf{M}/n$
14:         $\boldsymbol{\Delta} \leftarrow \mathbf{X}_k - \boldsymbol{\mu} \otimes \mathbf{1}_{m_k}^\top$                        ▷ Center vectors
15:         $\mathbf{S} \leftarrow \boldsymbol{\Delta}^\mathsf{H} \boldsymbol{\Delta}$
16:     **else**                                         ▷ Subsequent batches - update
17:         $\boldsymbol{\mu}_{\text{old}} \leftarrow \mathbf{M}/n$                                         ▷ Mean before update
18:         $n \leftarrow n + m_k$                                         ▷ Update count
19:         $\mathbf{M} \leftarrow \mathbf{M} + \sum_{j=1}^{m_k} \mathbf{x}_{k,j}$                        ▷ Update sum
20:         $\boldsymbol{\mu}_{\text{new}} \leftarrow \mathbf{M}/n$                                         ▷ Mean after update
21:         $\boldsymbol{\Delta}_{\text{old}} \leftarrow \mathbf{X}_k - \boldsymbol{\mu}_{\text{old}} \otimes \mathbf{1}_{m_k}^\top$
22:         $\boldsymbol{\Delta}_{\text{new}} \leftarrow \mathbf{X}_k - \boldsymbol{\mu}_{\text{new}} \otimes \mathbf{1}_{m_k}^\top$
23:         $\mathbf{S} \leftarrow \mathbf{S} + \boldsymbol{\Delta}_{\text{old}}^\mathsf{H} \boldsymbol{\Delta}_{\text{new}}$                        ▷ Welford update
24:     **end if**
25: **end for**
26:
27: **Finalization:**
28: $\boldsymbol{\mu} \leftarrow \mathbf{M}/n$                                         ▷ Final mean estimate
29: $\boldsymbol{\Sigma} \leftarrow \frac{1}{n-1} \cdot \frac{\mathbf{S}+\mathbf{S}^\mathsf{H}}{2}$                        ▷ Final covariance estimate
30: **return** $\boldsymbol{\mu}, \boldsymbol{\Sigma}$

---

## 7.2 INCORPORATING STEERING INTO MODEL WEIGHTS

Recall that the last layer of self-attention block in LLMs or cross-attention block in SDXL/SANA is a Linear layer with no bias and no activation function, i.e., essentially is a matrix multiplication with bias correction term: $h_{out} = W_{proj\_out}h_{in} + b$. Here $W_{proj\_out}$ is a weight matrix of the last $proj\_out$ layer of SA/CA block of LLM/SDXL/SANA, $h_{in}$ and $h_{out}$ are input and output to that layer, $h_{out}$ being the final output of SA/CA layer.

Assuming $s$ is normalized ($\|s\| = 1$), vanilla concept erasure (CASteer) represents orthogonal projection onto the subspace orthogonal to $s$ and can be written in a matrix form:

$$f_{\text{delete}}(h, s) = (I - ss^T)h \tag{20}$$

Vanilla concept switching is a Householder reflection of the vector $h$ across the hyperplane orthogonal to $s$:

$$f_{\text{switch}}(h, s) = (I - 2ss^T)h \tag{21}$$

LEACE / MiDSteer are already presented in this paper in matrix form.

Thus, by combining last layer of SA/CA block with matrix formulation of steering/LEACE/MiDSteer, we can incorporate the transformation directly into weights of the model, by multiplying weight matrix of the last layer of SA/CA block with $A^*$ matrix from of steering/LEACE/MiDSteer:

$$h_{out} = A^*(W_{proj\_out}h_{in} + b) + b^* = W^s_{proj\_out}h_{in} + b^s \tag{22}$$

$W^s_{proj\_out}$ is a matrix of the same size as $W_{proj\_out}$. This results in having zero inference overhead compares to original LLM/SDXL/SANA models.

| | Horse | Motorcycle | Cow | Dog |
|---|---|---|---|---|
| **Vanilla model** | In the heart of a vast, rolling green valley lay a secluded meadow embraced by ancient oaks and wildflowers in bloom. Here, the **horses** lived a simple, joyous life... | In the heart of the bustling city of San Francisco, there lived a **motorcycle** named Thunder. Unlike the other shiny, new **bikes** lining the streets, Thunder was an old Kawasaki with rusted edges... | In the lush, rolling hills of Green Meadows Farm, there lived a herd of contented **cows** that grazed under the warm sun and shared the soft, dewy grass. Among them was Daisy, a golden **cow** with spots as vivid as a child's imagination... | In the quiet town of Willowbrook, nestled between rolling hills and whispering woods, lived a small **terrier** named Bella. Bella was not just a **dog**; she was the heart of the household... |
| **Steering** | In the heart of a bustling city, there was once a forgotten corner, nestled between towering skyscrapers and hidden behind a rusty iron gate, where a vintage **bike** shop and a small garage sat side by side. Among the neglected treasures within the **garage** were forgotten car parts... | In the heart of a bustling city, amidst towering skyscrapers and winding streets, lived an old **stallion** named Max. Unusual for the concrete jungle, Max was never your typical **beast**... | In the quiet countryside of Green Valley, there lived a community of **cows**, but not just any **cows**. This was a town of the most unique and quirky cows you could imagine. Each cow had its own style of living, its way of seeing the world, and even its own personality, much like humans... | In the heart of a bustling city where skyscrapers kissed the sky and the streets were a maze of hurried footsteps and whispering winds, lived a small, golden **retriever** named Buddy. Buddy wasn't your ordinary family pet; his world was the park near the city's old library... |
| **LEACE-Switch** | In the heart of a quaint countryside stood a grand old barn that was more than just a shelter; it was the soul of a **motorcycle** club known as The Thunder, but hidden deep within its walls was a story of grace, freedom, and unexpected friendship... | In the heart of a sprawling, bustling city, there lived an old man named Jack who harbored a long-lost passion for **horses** until the day he met his first **motorcycle**. It was a sleek, black **beast** he spotted outside a garage one rainy evening... | In the peaceful, sun-drenched fields of Green Meadows Farm, there lived a herd of contented **cows**, each with its own personality and quirks. At the heart of this herd was a gentle giant named Bessie, known for her creamy coat and kind eyes... | In the quiet town of Millbrook, nestled among rolling hills and vibrant green landscapes, lived a **dog** named Buddy. Buddy wasn't just any **dog**; he was a loyal companion to the Smith family, known for his boundless energy and affectionate nature... |
| **MidSteer** | In the heart of the sprawling Wilden Plains, there thrived a community of **motorcycles** and cars, but it was the Rusty Rims Speedway that drew the greatest attention. Yet, hidden within the shadow of this bustling speedway, lived a group of **motorcycles**... | In the heart of a bustling city, where the sound of the urban jungle was constant, there lived a **motorcycle** named Racer. More than just a machine, Racer was an embodiment of freedom and adventure, cherished by his owner, Jake... | In the verdant fields of Sunny Meadows, where the sun bathed golden light onto rolling hills blanketed in lush grass, there lived a community of cheerful **cows**. Among them was Daisy, known not only for her creamy coat and gentle moo but for her adventurous spirit... | Once upon a time, in a small town nestled between rolling hills and a whispering forest, there lived a **dog** named Rusty. Rusty wasn't like the other **dogs** in town. While they were mostly interested in chasing balls and squirrels, Rusty loved nothing more than exploring the woods that bordered the town... |

Figure 4: Qualitative text steering results for four content categories (*horse*, *motorcycle*, *cow*, *dog*). Results are reported using vanilla Qwen2.5-14B-instruct model, and three steering methods: *Vanilla Steering*, *LEACE-Switch*, *MidSteer*). Each cell shows the generated text for the prompt *"Write a short story about a X"*, where X is a corresponding category.

## 7.3 LLM QUALITATIVE RESULTS

In this section in fig. 4 we present qualitative results for different steering methods on LLM (Qwen2.5-14B-instruct). Results show similar pattern to that of images (see fig. 3). While all methods similarly successfully performed switching from "horse" to "motorcycle", vanilla steering and LEACE failed when presented with prompt for the target concept ("motorcycle"), since they do not distinguish between forward and reverse steering. Vanilla steering also more significantly altered texts of concepts "cow" and "dog".

### 7.4 THEOREM PROOFS

#### 7.4.1 PROOF OF THM. 4 (VANILLA ERASURE IS A SPECIAL CASE OF LEACE)

*Proof.* We have $k = 1, Z = C$. According to **Theorem 3**, $f(X) = A^* X + b^*$, where $A^*$ is defined as in equation 6 and $b^*$ is defined as in equation 7, minimizes equation 5.

We conclude $b^* = 0$ since $\mathrm{E}[X] = 0$. Further, it can be shown that $W = \Sigma_{XX}^{-1/2} = I^{-1/2} = I$; hence, the transform $f$ (that is optimal according to **Theorem 3**) simplifies to

$$f(X) = \left(I - \Sigma_{XZ}\Sigma_{XZ}^+\right)X . \tag{23}$$

Recall that we are working in $k = 1$, so $\Sigma_{XZ} \in \mathbb{R}^{d \times 1}$ is a column-vector. By definition of the Moore-Penrose inverse for column-vectors,

$$\Sigma_{XZ}^+ = \frac{\Sigma_{XZ}^T}{\|\Sigma_{XZ}\|^2} ,$$

hence

$$f(X) = X - s's'^T X = X - \langle X, s'\rangle s',$$

for $s' = \Sigma_{XZ}/\|\Sigma_{XZ}\|$,

Now,

$$\Sigma_{XZ} = \mathrm{Cov}(X, Z) = \mathrm{E}[XZ] - \mathrm{E}[X] \cdot \mathrm{E}[Z] = \mathrm{E}[X \cdot 1 | Z = 1] \cdot P(Z = 1) +$$
$$\mathrm{E}[X \cdot 0 | Z = 0] \cdot P(Z = 0) - \mathrm{E}[X] \cdot P(Z = 1) =$$
$$P(Z = 1) \cdot \Big(\mathrm{E}[X | Z = 1] - \mathrm{E}[X]\Big)$$

Now recall that $P(Z = 1) + P(Z = 0) = 1$, so

$$\Sigma_{XZ} = P(Z = 1) \cdot \Big(\mathrm{E}[X | Z = 1] - \mathrm{E}[X]\Big) =$$
$$P(Z = 1) \cdot \Big(\mathrm{E}[X | Z = 1] - \mathrm{E}[X | Z = 1]P(Z = 1) - \mathrm{E}[X | Z = 0]P(Z = 0)\Big) =$$
$$P(Z = 1) \cdot P(Z = 0) \cdot \Big(\mathrm{E}[X | Z = 1] - \mathrm{E}[X | Z = 0]\Big)$$

, so $s' = s$ and $f$ is equivalent to $f_{delete}$. Hence, $f_{delete}$ is the transformation that minimises equation 8.

$\square$

#### 7.4.2 PROOF OF THM. 5 (LEACE-SWITCH)

*Proof.* The sketch for the rest of the proof will look like this:

1. Find necessary conditions for optimality using Lagrange multipliers method

2. Show that $A^*, b^*$ satisfy the necessary conditions

3. Show that optimisation problem is convex over linear constraints, and such, if a local solution exists, it is globally optimal and unique.

Let us formulate the Lagrangian. Here $\Lambda \in \mathbb{R}^{d \times k}$, because we have $d \cdot k$ constraints on covariance matrix.

$$\mathcal{L}(A, b, \Lambda) = \frac{1}{2}\mathrm{E}\Big[\|AX + b - X\|_2^2\Big] + \langle \Lambda, \mathrm{Cov}(AX + b, Z) + \mathrm{Cov}(X, Z)\rangle_F =$$

$$\frac{1}{2}\mathrm{E}\Big[(AX + b - X)^T(AX + b - X)\Big] + \mathrm{Tr}\Big(\Lambda^T(A + I)\Sigma_{XZ}\Big) =$$

$$\mathrm{E}\Big[\frac{1}{2}X^T A^T A X + b^T A X - X^T A X - X^T b + \frac{1}{2}b^T b + \frac{1}{2}X^T X\Big] + \mathrm{Tr}\Big(\Lambda^T(A + I)\Sigma_{XZ}\Big) \quad (24)$$

The partial derivatives of the Lagrangian with respect to $A, b, \Lambda$ are

$$\frac{\partial \mathcal{L}}{\partial A} = \mathrm{E}[AXX^T + bX^T - XX^T] + \Lambda\Sigma_{XZ}^T,$$
$$= A\mathrm{E}[XX^T] + b\mathrm{E}[X]^T - \mathrm{E}[XX^T] + \Lambda\Sigma_{XZ}^T,$$
$$\frac{\partial \mathcal{L}}{\partial b} = \mathrm{E}[AX + b - X],$$
$$= A\mathrm{E}[X] + b - \mathrm{E}[X],$$
$$\frac{\partial \mathcal{L}}{\partial \Lambda} = (A + I)\Sigma_{XZ}.$$

Next, we use $\mu = \mathrm{E}(X)$ and $\mathrm{E}[XX^T] = \Sigma_{XX} + \mu\mu^T$ to formulate the necessary conditions

$$0 = \frac{\partial \mathcal{L}}{\partial A} = (A - I)\Big(\Sigma_{XX} + \mu\mu^T\Big) + b\mu^T + \Lambda\Sigma_{XZ}^T, \quad (25)$$

$$0 = \frac{\partial \mathcal{L}}{\partial b} = A\mu + b - \mu, \quad (26)$$

$$0 = \frac{\partial \mathcal{L}}{\partial \Lambda} = (A + I)\Sigma_{XZ}. \quad (27)$$

We note that the optimal $b^*$ as defined in equation 12 satisfies 26. Plugging equation 26 in equation 25 leads to

$$(A - I)\Big(\Sigma_{XX} + \mu\mu^T\Big) + (\mu - A\mu)\mu^T + \Lambda\Sigma_{XZ}^T,$$
$$= A\Sigma_{XX} - \Sigma_{XX} + A\mu\mu^T - \mu\mu^T + \mu\mu^T - A\mu\mu^T + \Lambda\Sigma_{XZ}^T,$$
$$= (A - I)\Sigma_{XX} + \Lambda\Sigma_{XZ}^T = 0. \quad (28)$$

Now let us check that $A^*$ satisfies 27 and 28. By plugging $A^*$ into 27 we get

$$0 = (2I - 2W^+(W\Sigma_{XZ})(W\Sigma_{XZ})^+W)\Sigma_{XZ},$$
$$= 2\Sigma_{XZ} - 2W^+(W\Sigma_{XZ})(W\Sigma_{XZ})^+(W\Sigma_{XZ}),$$
$$= 2\Big(\Sigma_{XZ} - W^+W\Sigma_{XZ}\Big),$$
$$= 2\left(\Sigma_{XZ} - \big(I - P_{\mathcal{N}(W)}\big)\Sigma_{XZ}\right),$$
$$= 2P_{\mathcal{N}(W)}\Sigma_{XZ}, \quad (29)$$

because Moore-Penrose inverses $B^+$ of $B$ satisfy $BB^+B = B$ and $B^+B = I - P_{\mathcal{B}}$. Here $P_{\mathcal{B}}$ denotes the orthogonal projection onto the nullspace $\mathcal{N}(B)$ of $B$. Since the columns of $\Sigma_{XZ}$ always lie within the image of $\Sigma_{XX}$ (which is the orthogonal complement of the kernel of $\Sigma_{XX}$, which is also the kernel of $W$), we can conclude that equation 29 is always satisfied.

Plugging $A^*$ into 28 we observe

$$- 2 \cdot (W^+(W\Sigma_{XZ})(W\Sigma_{XZ})^+W)\Sigma_{XX} + \Lambda\Sigma_{XZ}^T =$$
$$- 2 \cdot W^+(W\Sigma_{XZ})(W\Sigma_{XZ})^+W^+ + \Lambda\Sigma_{XZ}^T = 0 \quad (30)$$

The identity $W\Sigma_{XX} = W^+$ holds because $\Sigma_{XX}$ is symmetric p.s.d., so $\Sigma_{XX} = UDU^T$ and $\Sigma_{XX}^{-1/2}\Sigma_{XX} = UD^{-1/2}U^T UDU^T = UD^{1/2}U^T = \Sigma_{XX}^{1/2}$ for some orthogonal $U$ and non-negative diagonal $D$, and because $D^{-1/2}$ ignores zero diagonal values.

Next, multiplying equation 30 by $W$ from both sides leads to

$$- 2WW^+(W\Sigma_{XZ})(W\Sigma_{XZ})^+W^+W + W\Lambda\Sigma_{XZ}^TW =$$
$$- 2(W\Sigma_{XZ})(W\Sigma_{XZ})^+ + W\Lambda(W\Sigma_{XZ})^T = -2\Sigma_{WX,Z}\Sigma_{WX,Z}^+ + \Lambda_W\Sigma_{WX,Z}^T = 0,$$
$$\text{(almost surely)}$$

where again $WW^+ = W^+W = I$ on a subspace covered by $X$, and thus, almost surely.

We seek $\Lambda_W$ such that $-2\Sigma_{WX,Z}\Sigma_{WX,Z}^+ + \Lambda_W\Sigma_{WX,Z}^T = 0$. Let $\Lambda_W = 2(\Sigma_{WX,Z}^+)^T$. This choice satisfies the condition because $\Sigma_{WX,Z}\Sigma_{WX,Z}^+$ is an orthogonal projection matrix, and is thus symmetric, so

$$(\Sigma_{WX,Z}^+)^T\Sigma_{WX,Z}^T = (\Sigma_{WX,Z}\Sigma_{WX,Z}^+)^T = \Sigma_{WX,Z}\Sigma_{WX,Z}^+,$$

which again proves the Lagrange conditions for partial derivative w.r.t. A.

Thus we have shown that the said optimisation problem has a local solution. But because the constraint is linear in $A$, and it follows from the triangle inequality that $\|\cdot\|$ is convex, the local optimum is actually the global minimum.

$\square$

### 7.4.3 PROOF OF THM. 6 (VANILLA CONCEPT SWITCHING IS A SPECIAL CASE OF LEACE-SWITCH)

*Proof.* Let $k = 1, Z = C, M = I$. According to **Theorem 5**, $f(X) = A^*X + b^*$, where $A^*$ is defined in equation 11 and $b^*$ is defined in equation 12 minimizes equation 10.

$b = 0$ since $\mathrm{E}[X] = 0$. Also it can be shown $W = \Sigma_{XX}^{-1/2} = I^{-1/2} = I$, so the transform becomes:

$$f(X) = \left(I - 2 \cdot \Sigma_{XZ}\Sigma_{XZ}^+\right)X \quad (31)$$

Recall that we are working in $k = 1$, so $\Sigma_{XZ} \in \mathbb{R}^{d \times 1}$ is a column-vector. So

$$\Sigma_{XZ} = \mathrm{Cov}(X, Z) = \mathrm{E}[XZ] - \mathrm{E}[X] \cdot \mathrm{E}[Z] = \mathrm{E}[X \cdot 1 | Z = 1] \cdot P(Z = 1) +$$
$$\mathrm{E}[X \cdot 0 | Z = 0] \cdot P(Z = 0) - \mathrm{E}[X] \cdot P(Z = 1) =$$
$$P(Z = 1) \cdot \left(\mathrm{E}[X | Z = 1] - \mathrm{E}[X]\right)$$

Now recall that $P(Z = 1) + P(Z = 0) = 1$, so

$$\Sigma_{XZ} = P(Z = 1) \cdot \left(\mathrm{E}[X | Z = 1] - \mathrm{E}[X]\right) =$$
$$P(Z = 1) \cdot \left(\mathrm{E}[X | Z = 1] - \mathrm{E}[X | Z = 1]P(Z = 1) - \mathrm{E}[X | Z = 0]P(Z = 0)\right) =$$
$$P(Z = 1) \cdot P(Z = 0) \cdot \left(\mathrm{E}[X | Z = 1] - \mathrm{E}[X | Z = 0]\right)$$

, which is equal to $s$ up to normalization constant. By definition of Moore-Penrose inverse for column-vectors,

$$\Sigma_{XZ}^+ = \frac{\Sigma_{XZ}^T}{\|\Sigma_{XZ}\|^2}$$

, so

$$f(X) = X - 2 \cdot ss^T X = X - 2 \cdot s(s^T X) = X - 2 \cdot (s^T X)s = X - 2 \cdot \langle X, s \rangle s$$

, so $f$ is equivalent to $f_{switch}$.

$\square$

### 7.4.4 Proof of Thm. 7 (MidSteer)

*Proof.* We will use the same method as in 7.4.2 to prove this. Indeed, the objective is same, and thus convex. The constraint is still linear:

$$\text{Cov}(Ax + b, Z_1) = A\Sigma_{XZ_1} = \Sigma_{XZ_2} = \text{Cov}(X, Z_2) \tag{32}$$

So let us define the Lagrangian, where $\Lambda \in \mathbb{R}^{d \times l}$:

$$\mathcal{L}(A, b, \Lambda) = \frac{1}{2}\text{E}\left[(AX + b - X)^T(AX + b - X)\right] + Tr\left(\Lambda^T(A\Sigma_{XZ_1} - \Sigma_{XZ_2})\right) \tag{33}$$

The derivatives w.r.t. parameters are the following:

$$\frac{\partial \mathcal{L}}{\partial A} = (A - I)(\Sigma_{XX} + \mu\mu^T) + b\mu^T + \Lambda\Sigma_{XZ_1}^T \qquad = 0 \tag{34}$$

$$\frac{\partial \mathcal{L}}{\partial b} = A\mu - \mu + b \qquad = 0 \tag{35}$$

$$\frac{\partial \mathcal{L}}{\partial \Lambda} = A\Sigma_{XZ_1} - \Sigma_{XZ_2} \qquad = 0 \tag{36}$$

Trivially $b^*$ satisfies equation 35 for suitable $A^*$.

Let us see that equation 36 is satisfied. We can plug $A^*$ and then multiply by $W$ on the left, to get:

$$W\Sigma_{XZ_1} + WW^+(\Sigma_{WX,Z_2} - \Sigma_{WX,Z_1})\Sigma_{WX,Z_1}^+ W\Sigma_{XZ_1} - W\Sigma_{XZ_2} =$$
$$\Sigma_{WX,Z_1} - \Sigma_{WX,Z_1}\Sigma_{WX,Z_1}^+\Sigma_{WX,Z_1} + \Sigma_{WX,Z_2}\Sigma_{WX,Z_1}^+\Sigma_{WX,Z_1} - \Sigma_{WX,Z_2} =$$
$$\Sigma_{WX,Z_2}I_l - \Sigma_{WX,Z_2} = 0 \quad \text{(almost surely)}$$

Here we used $YY^+Y = Y$ for any $Y$. Furthermore, since $\Sigma_{WX,Z_1}$ has linearly independent columns (full column rank), we use the property $Y^+Y = I$.

Next, plugging equation 35 into equation 34 we get:

$$(A - I)\Sigma_{XX} + \Lambda\Sigma_{XZ_1}^T = 0 \tag{37}$$

Let us now proceed to show that for $A^*$ there exists $\Lambda \in \mathbb{R}^{d \times l}$ so this equality holds. After plugging in $A^*$ and using previously shown fact $W\Sigma_{XX} = W^+$:

$$W^+(\Sigma_{WX,Z_2} - \Sigma_{WX,Z_1})\Sigma_{WX,Z_1}^+ W^+ + \Lambda\Sigma_{XZ_1}^T = 0 \tag{38}$$

Again, multiplying by $W$ on both sides and recalling that $W$ is symmetric we get:

$$(\Sigma_{WX,Z_2} - \Sigma_{WX,Z_1})\Sigma_{WX,Z_1}^+ + \Lambda_W\Sigma_{WX,Z_1}^T = 0 \quad \text{(almost surely)}$$

Now, $\Sigma^+_{WX,Z_1}$ is also full column rank, so $\Sigma^+_{WX,Z_1} = \left(\Sigma^T_{WX,Z_1}\Sigma_{WX,Z_1}\right)^{-1}\Sigma^T_{WX,Z_1}$. Thus,

$\Lambda_W = -\left(\Sigma_{WX,Z_2} - \Sigma_{WX,Z_1}\right)\left(\Sigma^T_{WX,Z_1}\Sigma_{WX,Z_1}\right)^{-1}$ satisfies the equation.

$\square$

### 7.5 EXAMPLE LLM PROMPTS FOR ESTIMATING CLASS-CONDITIONAL COVARIANCE

To obtain class-conditional means for LLMs, we constructed 1000 prompts across several categories. In each case, last token activation was used to compute steering vectors, before first token of model output and after corresponding chat template tokens. Below we provide illustrative subsets of the prompts used:

- **Horse-related prompts**
  - How did horses evolve from their early ancestors?
  - What factors contributed to the domestication of horses?
  - What anatomical features enable a horse to run at high speeds?
- **Motorcycle-related prompts**
  - What was the significance of the earliest gasoline powered motorcycle in the evolution of personal transportation?
  - How does the engine cycle mechanism influence power delivery and efficiency in a motorbike?
  - What are the primary considerations when selecting the appropriate tire for varying riding conditions?
- **Dog-related prompts**
  - What is believed to be the origin of dog domestication?
  - How do dogs communicate through body language?
  - Which genetic factors influence coat color in dogs?
- **Cat-related prompts**
  - What is the origin of domestic cats and how did they become companion animals?
  - How did ancient Egyptians incorporate cats into their society and religion?
  - Why did cats spread globally alongside human trade routes?

### 7.6 PROMPT FOR CONCEPT SCORING IN LLMS

To assess the quality of steering, we compute a *concept score* that gauges the presence of a specific concept in generated text. The score ranges from 0 to 10:

- A score of 0 indicates the text is completely irrelevant to the concept.
- A score of 10 indicates strong and explicit relevance to the concept.
- Scores from 1 through 9 represent gradations of partial relevance.

The following prompt was used to query the model for concept scoring:

```
Score the text on a scale of 0 to 10, where the score reflects the extent
to which it demonstrates relevance to the concept of {concept}. A score of 0
indicates the text is completely irrelevant and shows no connection to {concept},
whereas a score of 10 signifies strong and explicit relevance to {concept}. For
texts that suggest varying degrees of relevance, use the intermediate numbers
(1 through 9) to represent the gradation of connection to {concept}.
Output just a single number.

Text to evaluate: ```
{text}
```
```

The exact score is obtained by doing a forward pass of Llama3.1-7B model and taking argmax across token probabilities corresponding to score tokens (from 0 to 10).

## 7.7 TEMPLATE PROMPTS FOR STEERING QUALITY ASSESSMENT

To evaluate the effectiveness of steering, we constructed a set of template prompts. In each case, the {} placeholder was substituted with the concept under consideration (as defined in **??**). Each template was used to generate multiple model responses. These responses were then evaluated either by a judge model (for text/LLM outputs) or by CLIP score (for images), to compute the Concept Score as well as related metrics such as FID and BERTScore.

### 7.7.1 LLM ASSESSMENT PROMPTS

The following template prompts were used for large language model (LLM) evaluation. The placeholder {} is substituted with a concept.

```
 1.  Write a short story about a {}
 2.  Write a poem about a {}
 3.  What is the history of {}
 4.  What is the most famous {}?
 5.  What is the most expensive {}?
 6.  How to make a {}?
 7.  How to maintain a {}?
 8.  How to dispose of a {}?
 9.  How to store a {}?
10.  How to transport a {}?
11.  How to use a {}?
12.  What is important to know about {}?
13.  How to tell age of a {}?
14.  What is a {}?
15.  What types of {} are there?
16.  What are the most common {}?
17.  Describe an appearance of {} in detail.
18.  How does {} look like?
19.  How does {} sound like?
20.  How does {} feel like?
21.  How does {} behave like?
22.  What is the purpose of {}?
23.  What are the main components of a {}?
24.  How to identify a {}?
25.  Where can you find a {}?
26.  What are the dangers of a {}?
27.  What tools do you need for a {}?
28.  How much does a {} typically cost?
29.  What are alternatives to a {}?
30.  How to choose a good {}?
31.  What are common problems with a {}?
32.  How long does a {} typically last?
33.  What size is a typical {}?
34.  How to clean a {}?
35.  What skills are needed to handle a {}?
36.  What are the benefits of having a {}?
37.  How has {} changed over time?
38.  What cultures use {} the most?
39.  How to test if a {} is working properly?
40.  What safety precautions are needed for a {}?
41.  How to upgrade or improve a {}?
42.  How does weather affect a {}?
43.  What are the environmental impacts of a {}?
```

44.  How to measure the quality of a {}?
45.  What accessories go with a {}?
46.  How to protect a {} from damage?
47.  What are myths about {}?
48.  How to teach someone about a {}?
49.  What industries use {}?
50.  How is a {} different from similar things?
51.  What are the legal considerations for owning a {}?
52.  How to pack a {} for moving?
53.  What are seasonal considerations for a {}?
54.  How to customize a {}?
55.  What are expert tips for using a {}?
56.  How to troubleshoot issues with a {}?
57.  What is the lifecycle of a {}?
58.  How to estimate the value of a {}?
59.  What are cultural significances of a {}?
60.  How to take a picture of a {}?
61.  How to make a sculpture of a {}?
62.  What is the future of {}?
63.  How to draw a {}?
64.  When was {} first mentioned in human history?
65.  Can one ride a {}?
66.  Write a song about {}
67.  Define a {}
68.  Write a positive review on a book about {}
69.  Write a negative review on a book about {}
70.  Do people make toys of {}?
71.  How is {} used in the economy?
72.  Write an abstract for a science paper about {}
73.  How does temperature affect a {}?
74.  What are the origins of the word {}?
75.  What are superstitions about {}?
76.  How to simulate a {} digitally?
77.  What are the physics of a {}?
78.  How to teach children about {}?
79.  What are famous artworks featuring {}?
80.  What are the nutritional aspects of a {}?
81.  Describe the most famous {} competitions.

### 7.7.2    IMAGE ASSESSMENT PROMPTS

The following template prompts were used for image model evaluation. The placeholder {} was substituted with a concept.

1.  a bad photo of a {}.
2.  a photo of many {}.
3.  a sculpture of a {}.
4.  a photo of the hard to see {}.
5.  a low resolution photo of the {}.
6.  a rendering of a {}.
7.  graffiti of a {}.
8.  a bad photo of the {}.
9.  a cropped photo of the {}.
10. a tattoo of a {}.
11. the embroidered {}.
12. a photo of a hard to see {}.
13. a bright photo of a {}.
14. a photo of a clean {}.
15. a photo of a dirty {}.

```
16.  a dark photo of the {}.
17.  a drawing of a {}.
18.  a photo of my {}.
19.  the plastic {}.
20.  a photo of the cool {}.
21.  a close-up photo of a {}.
22.  a black and white photo of the {}.
23.  a painting of the {}.
24.  a painting of a {}.
25.  a pixelated photo of the {}.
26.  a sculpture of the {}.
27.  a bright photo of the {}.
28.  a cropped photo of a {}.
29.  a plastic {}.
30.  a photo of the dirty {}.
31.  a jpeg corrupted photo of a {}.
32.  a blurry photo of the {}.
33.  a photo of the {}.
34.  a good photo of the {}.
35.  a rendering of the {}.
36.  a {} in a video game.
37.  a photo of one {}.
38.  a doodle of a {}.
39.  a close-up photo of the {}.
40.  a photo of a {}.
41.  the origami {}.
42.  the {} in a video game.
43.  a sketch of a {}.
44.  a doodle of the {}.
45.  a origami {}.
46.  a low resolution photo of a {}.
47.  the toy {}.
48.  a rendition of the {}.
49.  a photo of the clean {}.
50.  a photo of a large {}.
51.  a rendition of a {}.
52.  a photo of a nice {}.
53.  a photo of a weird {}.
54.  a blurry photo of a {}.
55.  a cartoon {}.
56.  art of a {}.
57.  a sketch of the {}.
58.  a embroidered {}.
59.  a pixelated photo of a {}.
60.  itap of the {}.
61.  a jpeg corrupted photo of the {}.
62.  a good photo of a {}.
63.  a plushie {}.
64.  a photo of the nice {}.
65.  a photo of the small {}.
66.  a photo of the weird {}.
67.  the cartoon {}.
68.  art of the {}.
69.  a drawing of the {}.
70.  a photo of the large {}.
71.  a black and white photo of a {}.
72.  the plushie {}.
73.  a dark photo of a {}.
74.  itap of a {}.
```

```
75.   graffiti of the {}.
76.   a toy {}.
77.   itap of my {}.
78.   a photo of a cool {}.
79.   a photo of a small {}.
80.   a tattoo of the {}.
```

## 7.8 NUMBER OF PROMPTS FOR COVARIANCE CALCULATION

To find the optimal number of prompts used to calculate unconditional covariances $\Sigma_{XX}$ for concept switching, we perform the following ablation study. For each number of prompts used for covariances generation from the set $\{100, 500, 1000, 5000, 10000, 20000\}$ we run the same base experiment as outlined in Sec. 5. We then compute $\Delta CS$ and 1 - BERT Precision @ MMLU metrics on a small set of MiDSteer steering strengths (to show if the steering strength can affect the optimal number of prompts). We do it on LLM experiment setup (sec. 5.1) and use LLama2-7B model.

We then plot these values of a 2D plane similar to Pareto charts, but this time varying the number of prompts instead. This in essence forms a curve that, after a certain threshold, settles in a small region of metric space. As can be seen from the chart below, increasing the number of prompts used beyond 5000 has limited impact.

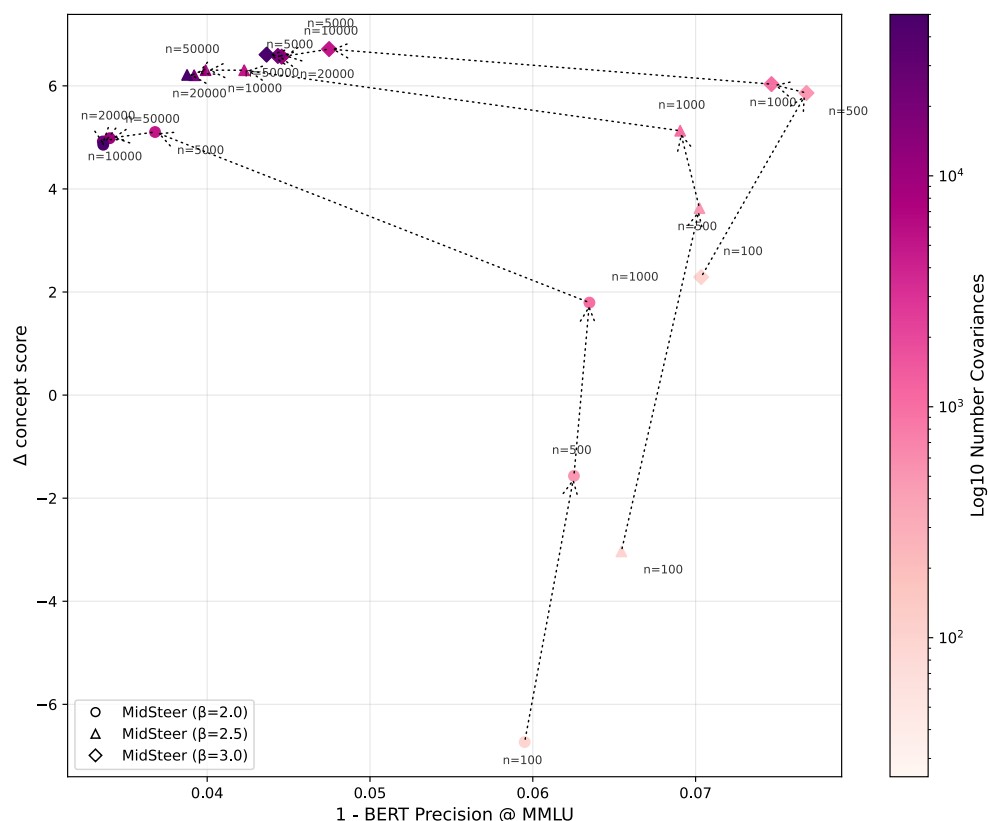

## 7.9 MORE RESULTS ON CONCEPT SWITCHING

### 7.9.1 PARETO CHARTS FOR LLM CONCEPT SWITCHING

In this section, we provide more Pareto charts for LLM concept switching. When switching concept $c_s$ to $c_t$, for each LLM model we provide 9 types of Pareto plots:

- 1 - BERT Precision score for unrelated concepts (horizontal axis) vs $\Delta$ Concept Score (CS) for the target $c_t$ and source $c_s$ concepts (fig. 6, 9, 12)
- 1 - BERT Precision score for MMLU (horizontal axis) vs$\Delta$ Concept Score (CS) for the target $c_t$ and source $c_s$ concepts (vertical axis) (fig. 5, 8, 11)
- Average Concept Score (CS) for unrelated concepts $c_i$ (horizontal axis) vs $\Delta$ Concept Score (CS) for the target $c_t$ and source $c_s$ concepts (vertical axis) (fig. 7, 10, 13)
- 1 - BERT Precision score for unrelated concepts (horizontal axis) vs Concept Score (CS) for the **source** $c_s$ concept (vertical axis) (fig. 15, 18, 21)
- BERT Precision score for MMLU (horizontal axis) vs Concept Score (CS) for the **source** $c_s$ concept (vertical axis) (fig.14, 17, 20)
- Average Concept Score (CS) for unrelated concepts $c_i$ (horizontal axis) vs Concept Score (CS) for the **source** $c_s$ concept (vertical axis) (fig.16, 19, 22)
- 1 - BERT Precision score for unrelated concepts (horizontal axis) vs Concept Score (CS) for the **target** $c_s$ concept (vertical axis) (fig. 15, 27, 21)
- BERT Precision score for MMLU (horizontal axis) vs Concept Score (CS) for the **target** $c_s$ concept (vertical axis) (fig.14, 17, 20)
- Average Concept Score (CS) for unrelated concepts $c_i$ (horizontal axis) vs Concept Score (CS) for the **target** $c_s$ concept (vertical axis) (fig.16, 19, 22)

In each case, we see clear superiority of MidSteer over other steering approaches.

We additionally provide detailed breakdown of scores for all $\beta$ values and all concepts $c_s, c_t, c_i$ in the tables in sec. 7.9.3

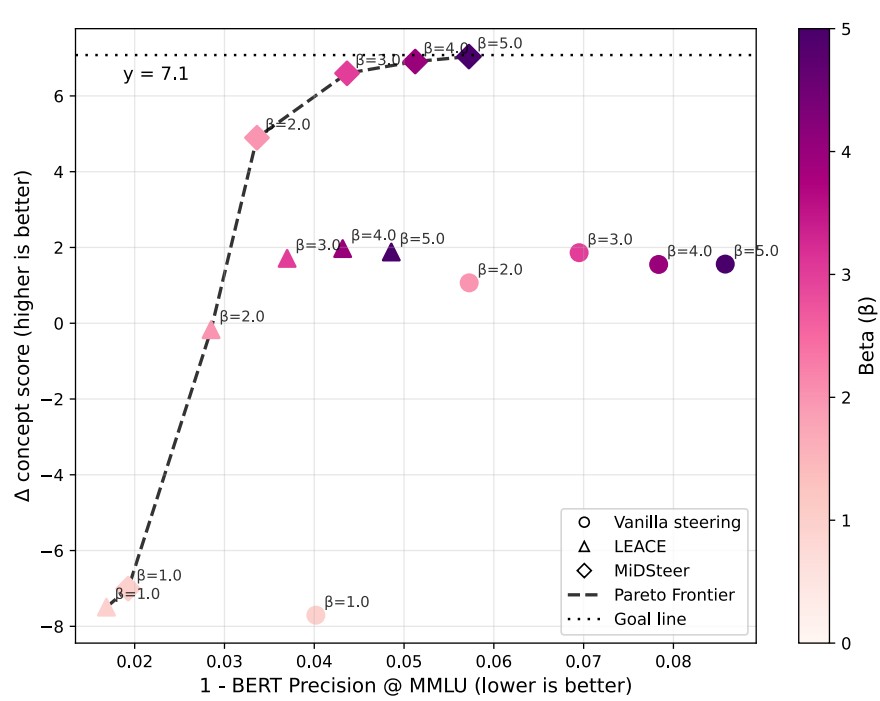

Figure 5: Pareto plot for concept flip on model llama2-7b

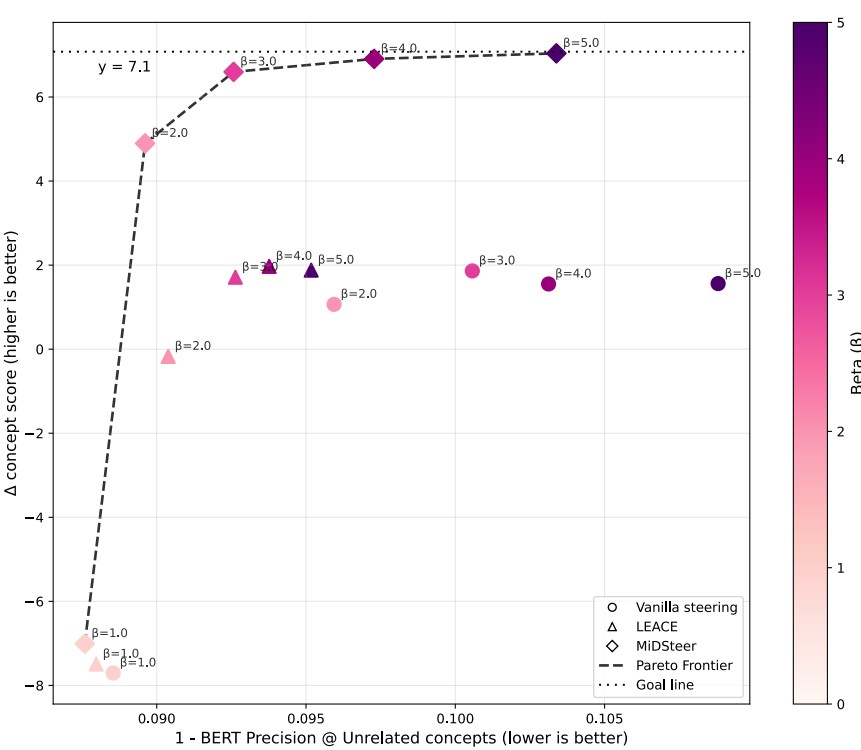

Figure 6: Pareto plot for concept flip on model llama2-7b

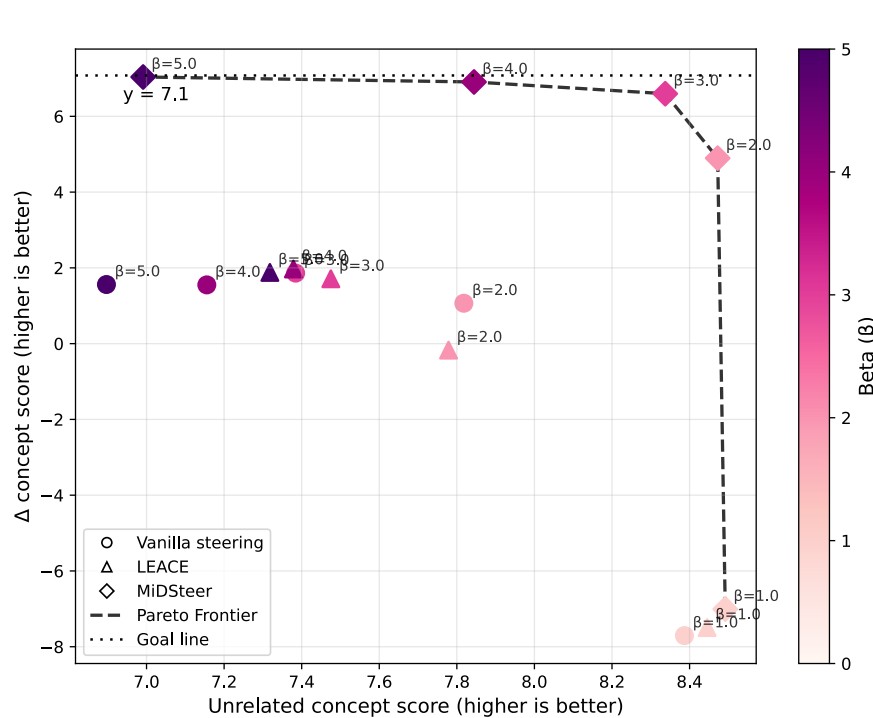

Figure 7: Pareto plot for concept flip on model llama2-7b

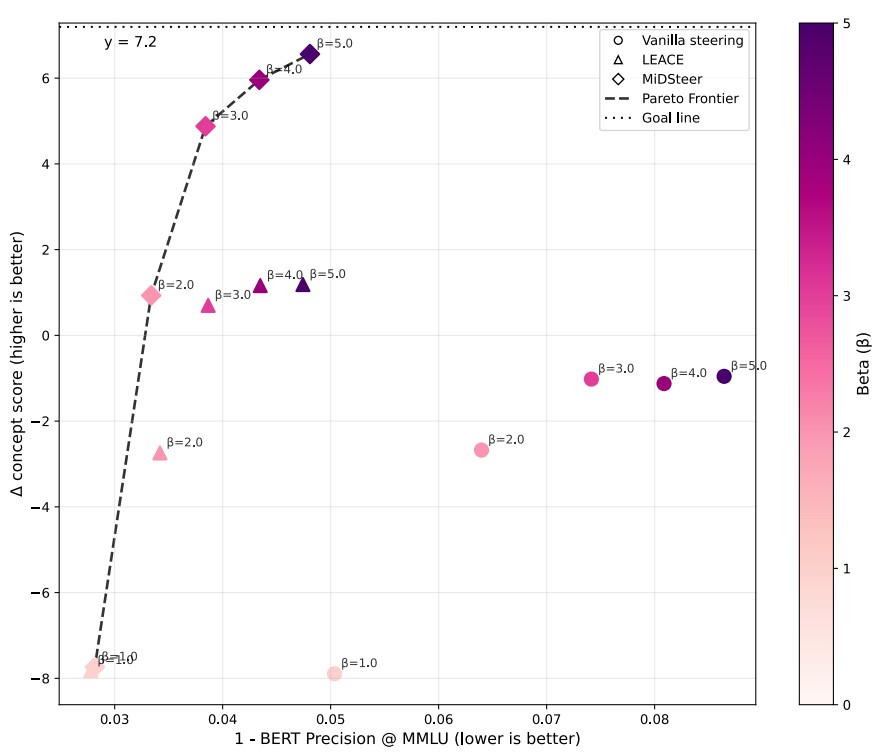

Figure 8: Pareto plot for concept flip on model qwen-14b

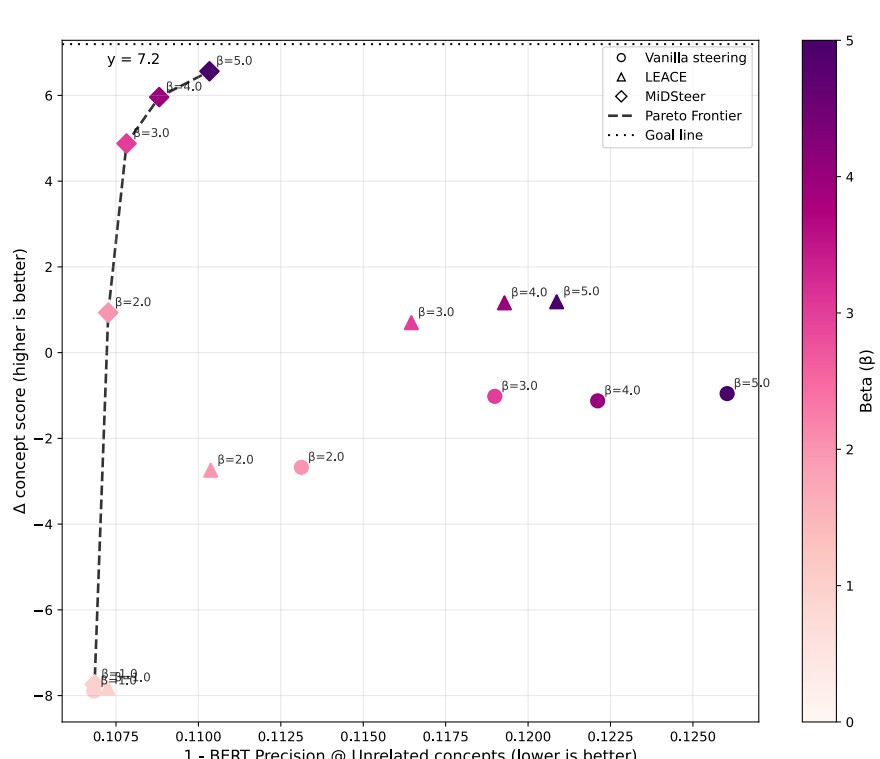

Figure 9: Pareto plot for concept flip on model qwen-14b

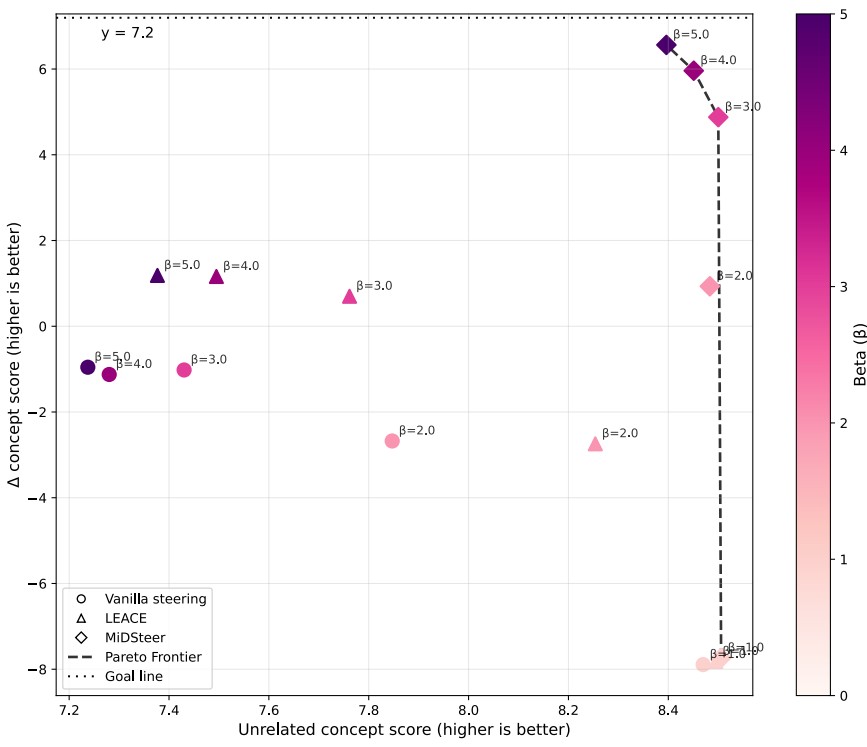

Figure 10: Pareto plot for concept flip on model qwen-14b

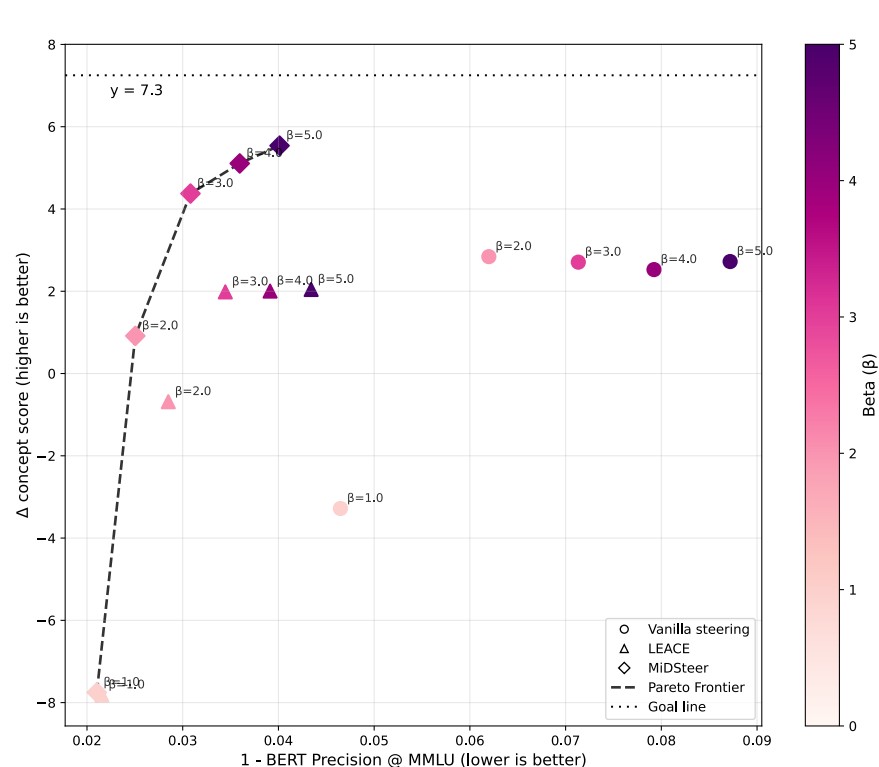

Figure 11: Pareto plot for concept flip on model qwen-7b

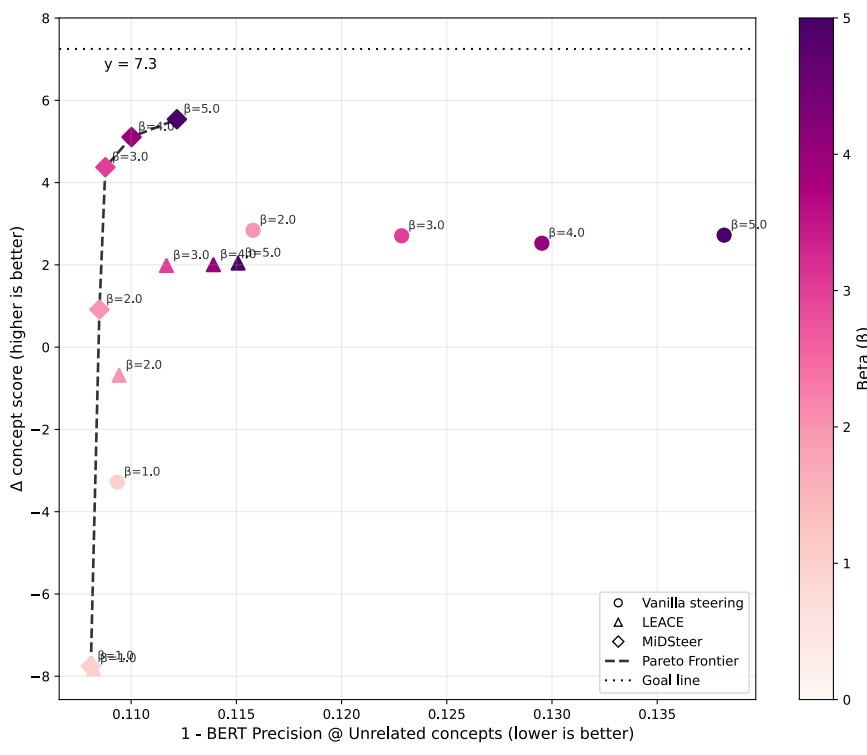

Figure 12: Pareto plot for concept flip on model qwen-7b

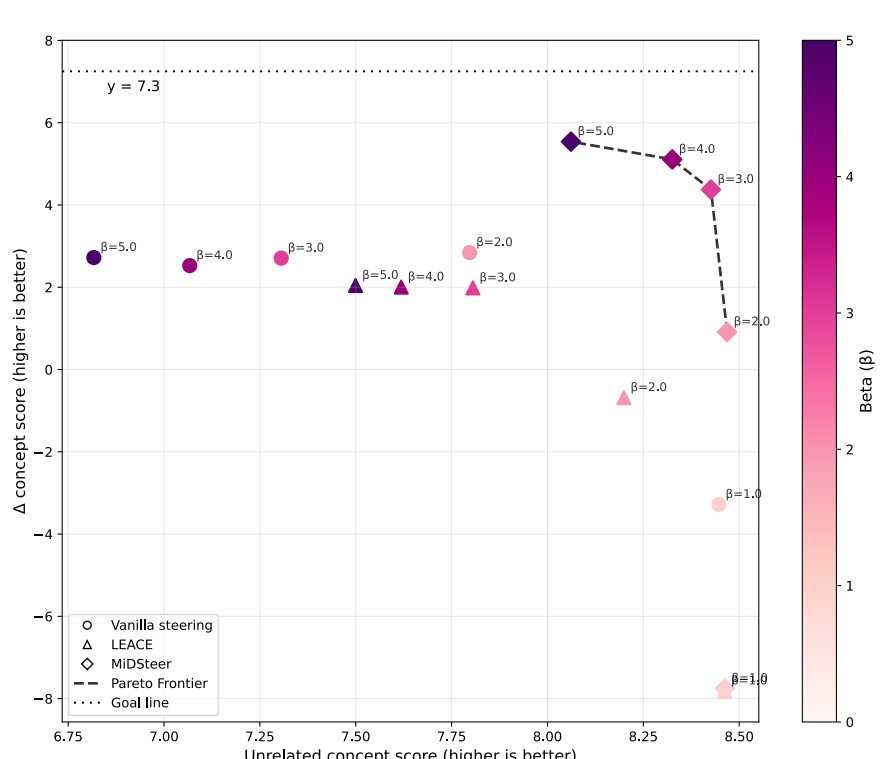

Figure 13: Pareto plot for concept flip on model qwen-7b

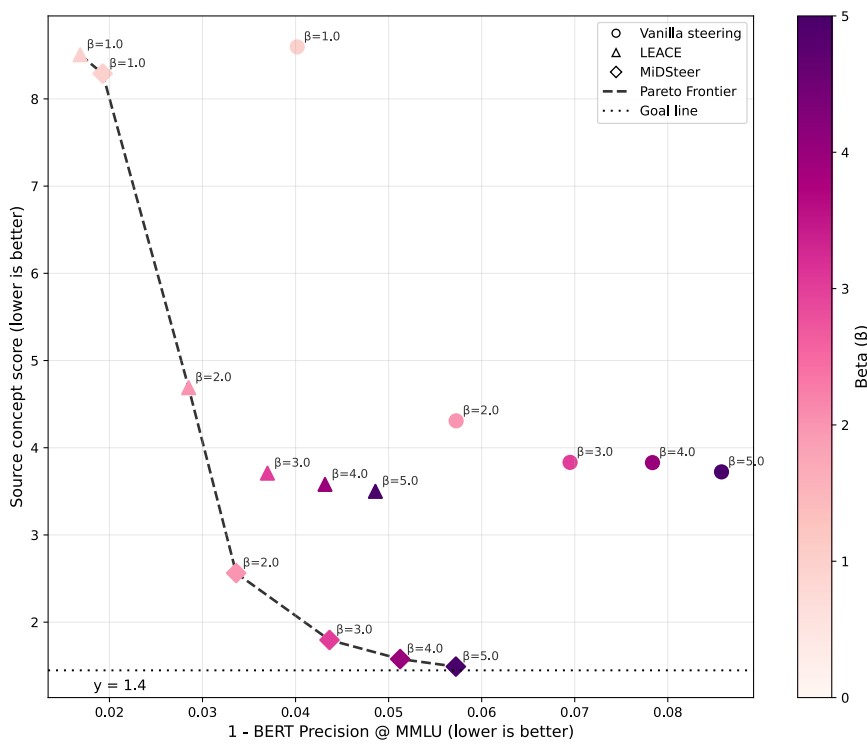

Figure 14: Pareto plot for concept flip on model llama2-7b

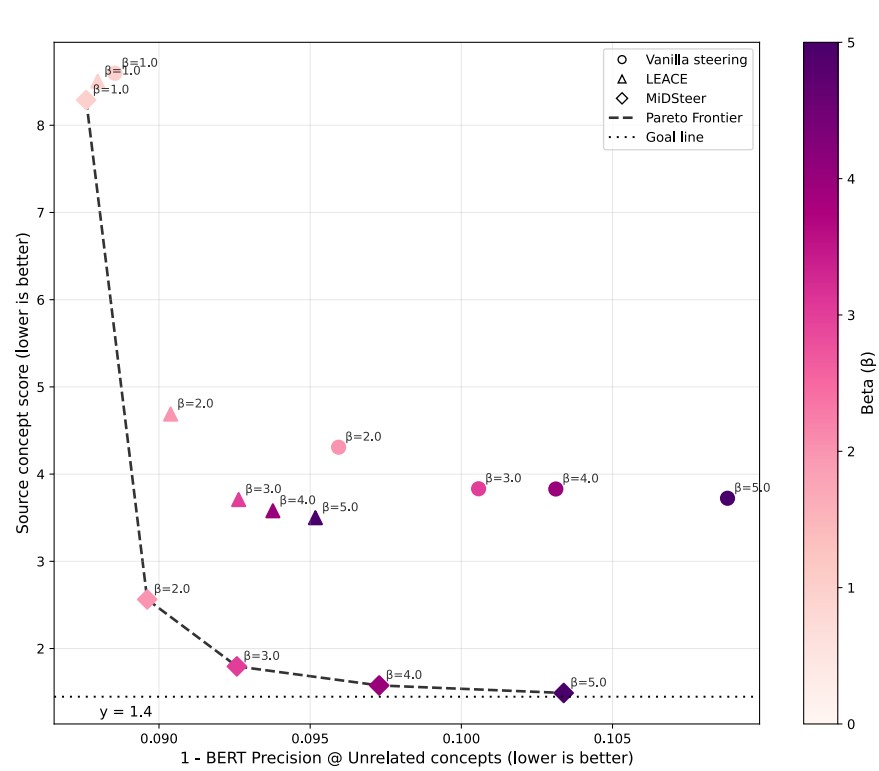

Figure 15: Pareto plot for concept flip on model llama2-7b

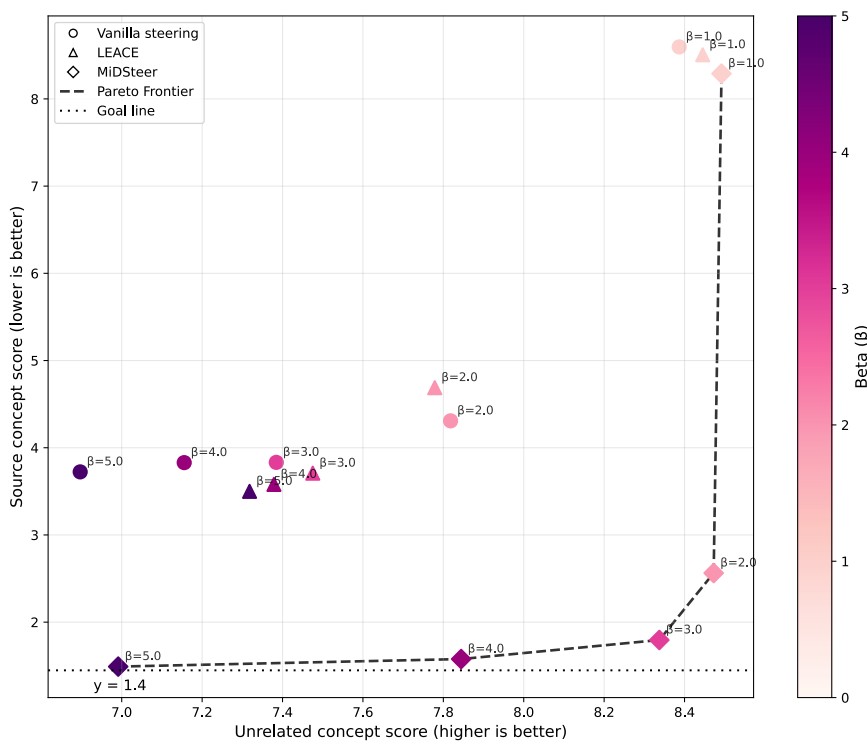

Figure 16: Pareto plot for concept flip on model llama2-7b

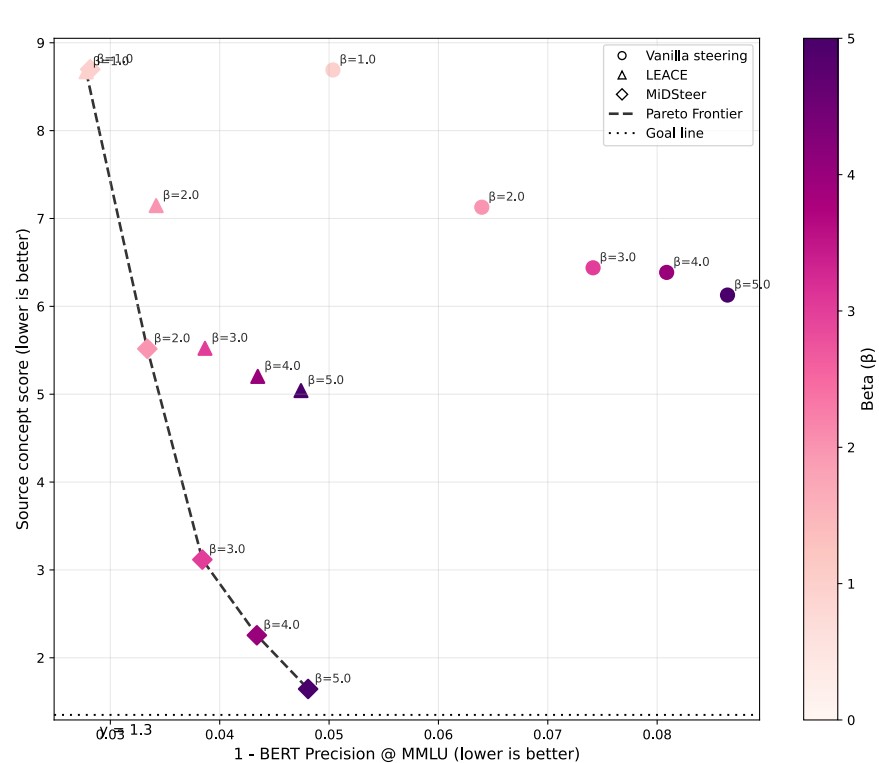

Figure 17: Pareto plot for concept flip on model qwen-14b

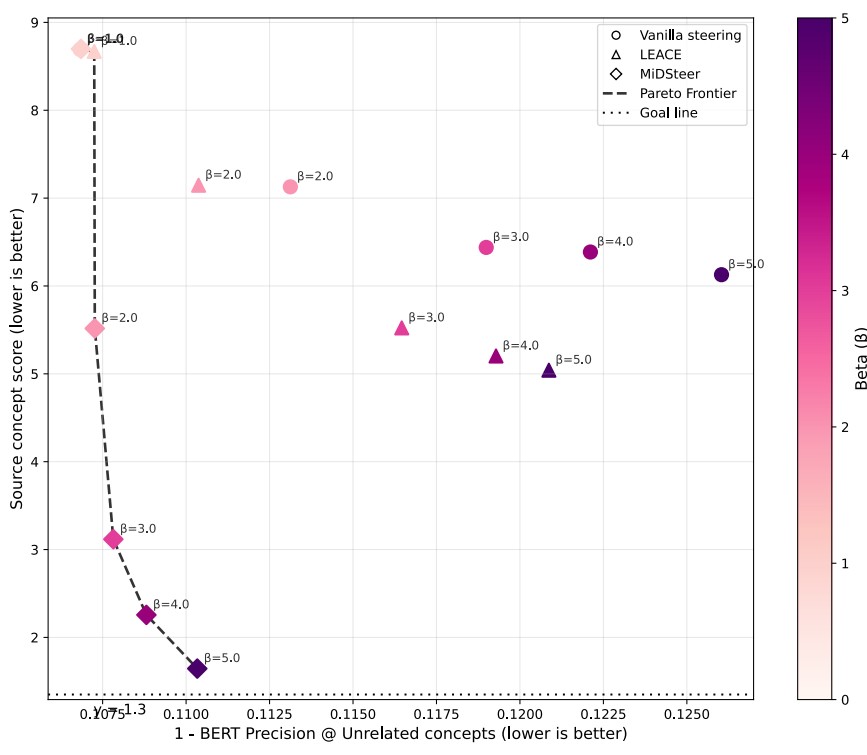

Figure 18: Pareto plot for concept flip on model qwen-14b

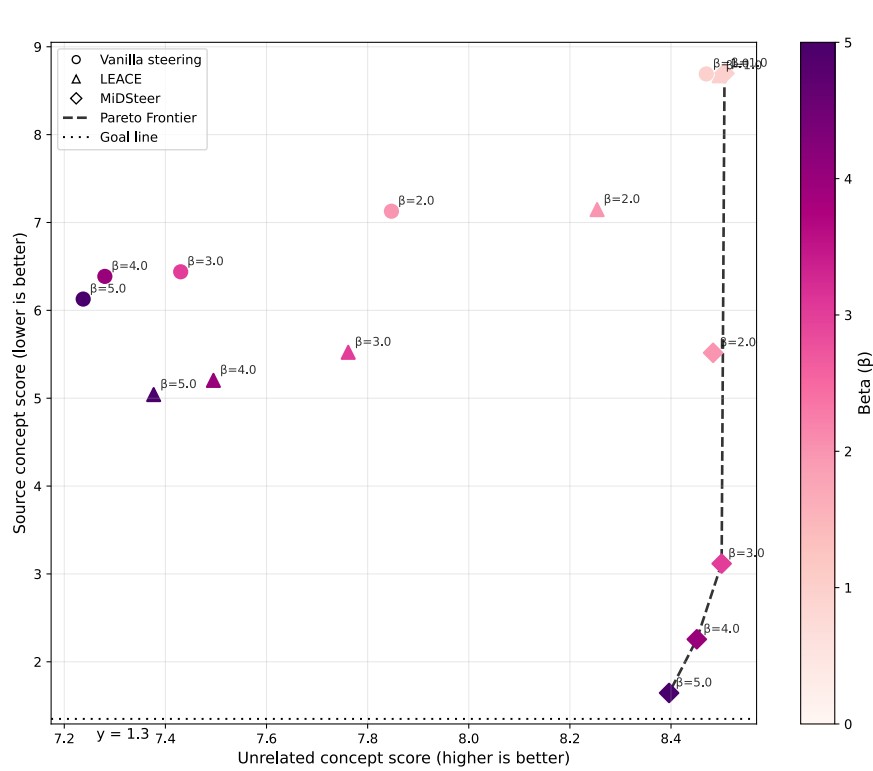

Figure 19: Pareto plot for concept flip on model qwen-14b

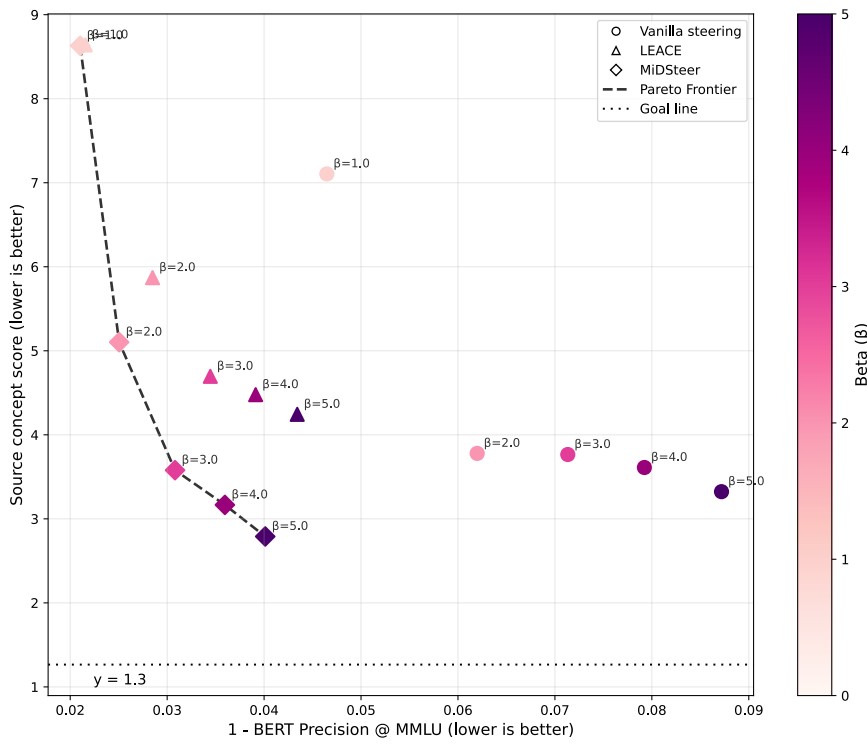

Figure 20: Pareto plot for concept flip on model qwen-7b

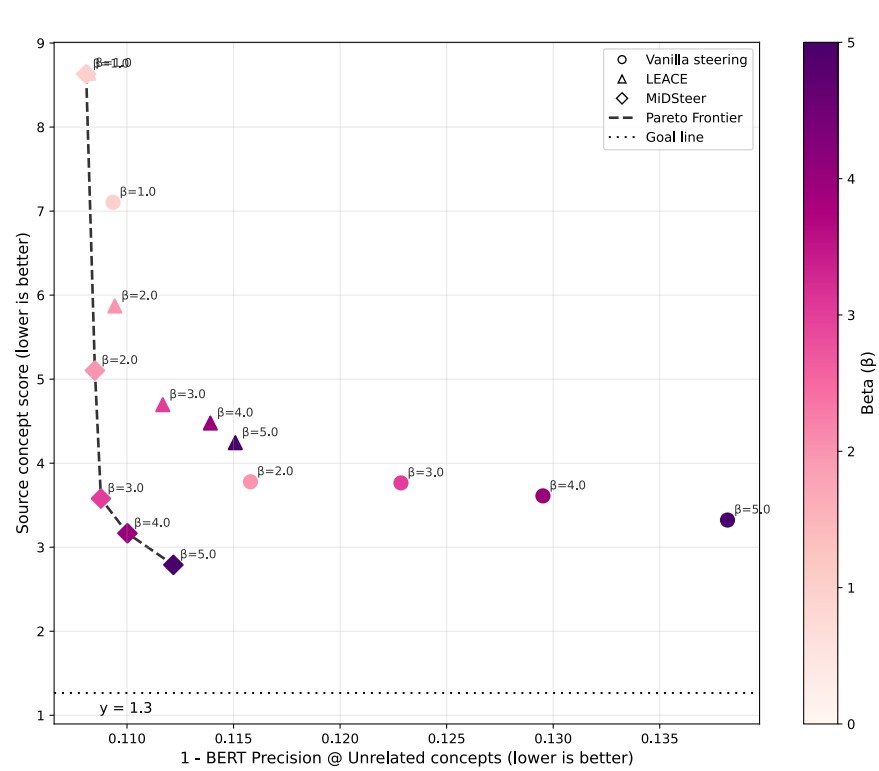

Figure 21: Pareto plot for concept flip on model qwen-7b

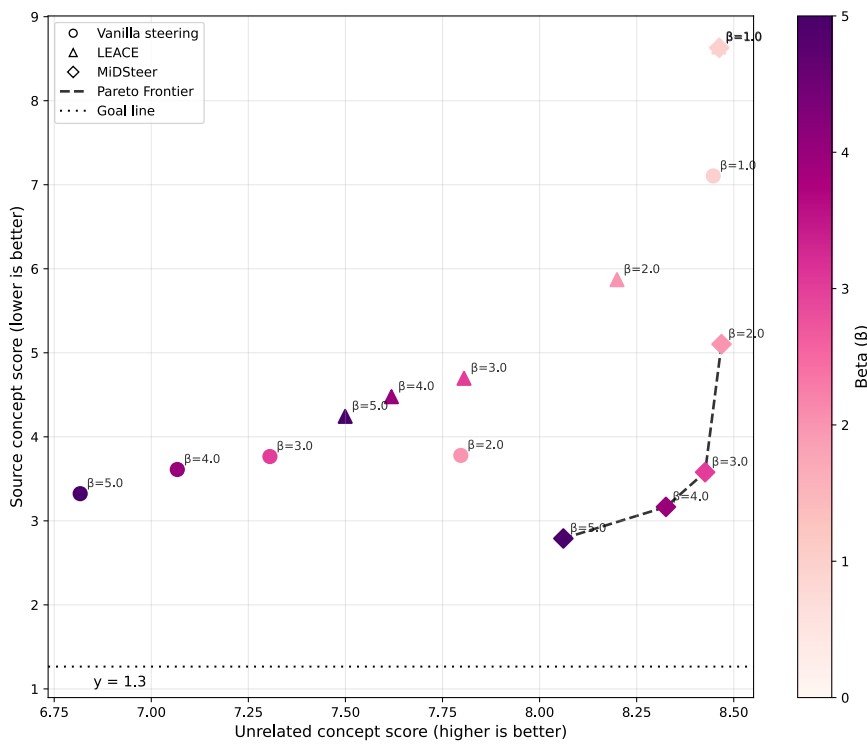

Figure 22: Pareto plot for concept flip on model qwen-7b

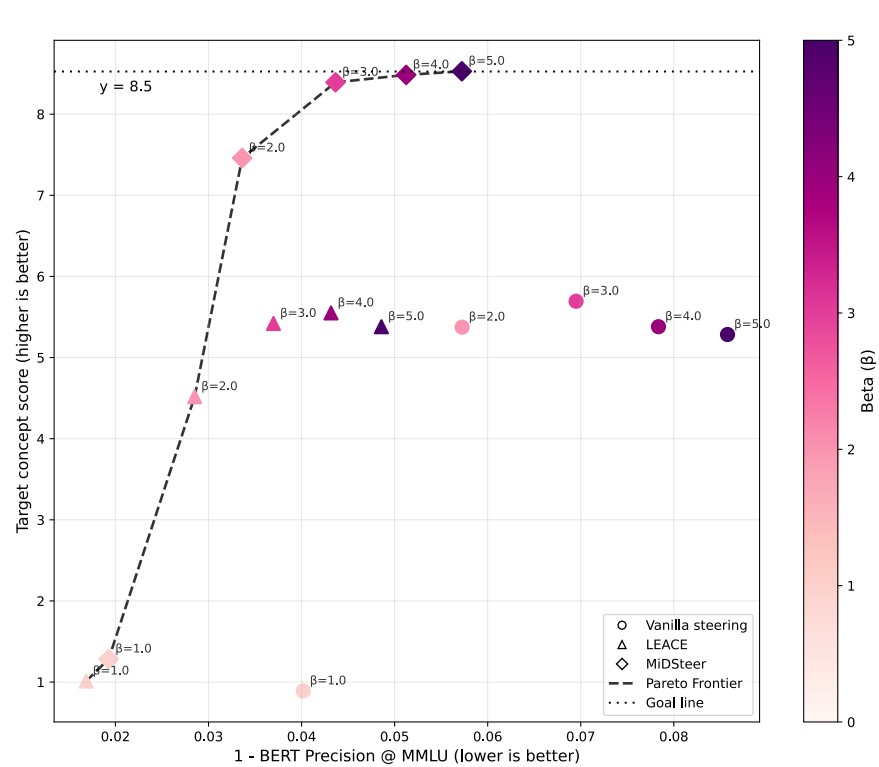

Figure 23: Pareto plot for concept flip on model llama2-7b

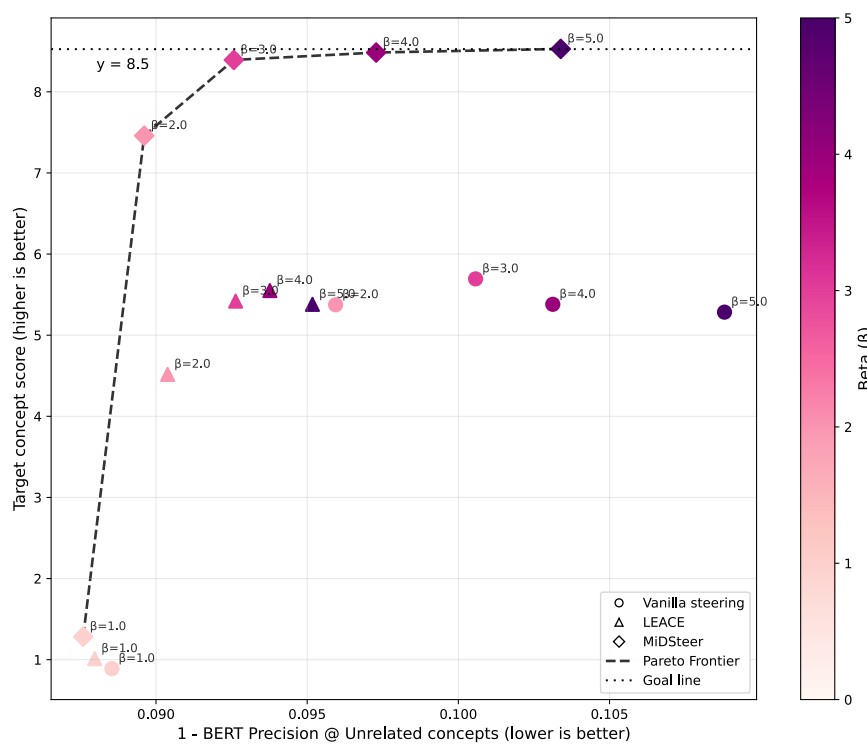

Figure 24: Pareto plot for concept flip on model llama2-7b

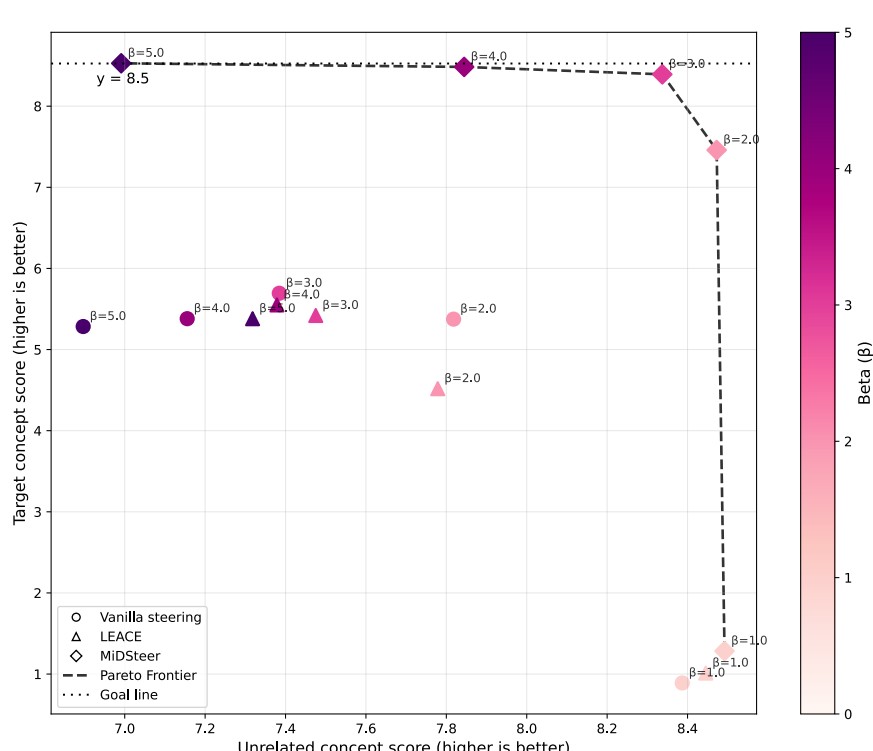

Figure 25: Pareto plot for concept flip on model llama2-7b

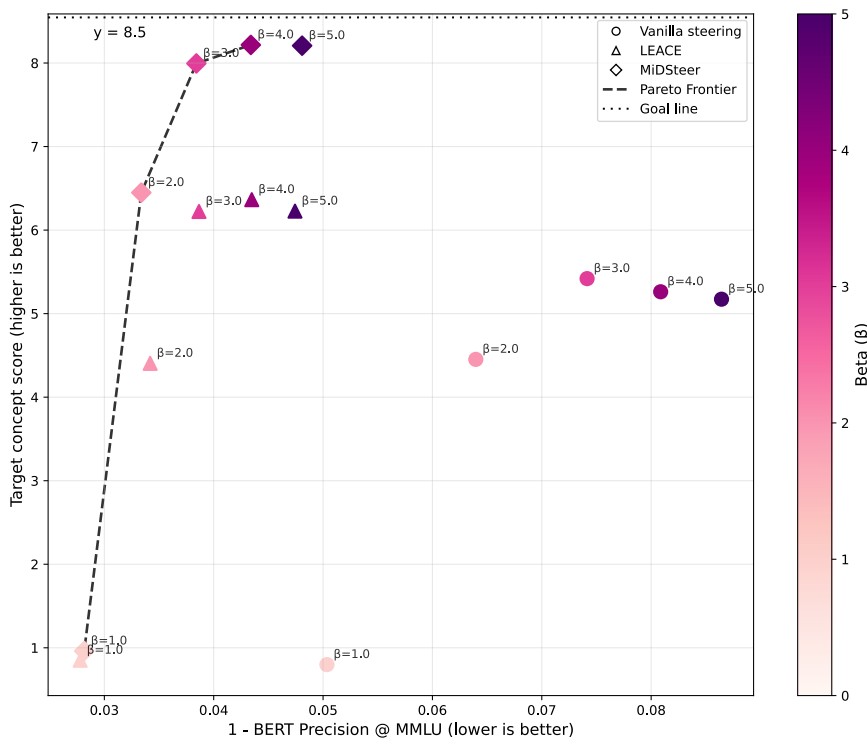

Figure 26: Pareto plot for concept flip on model qwen-14b

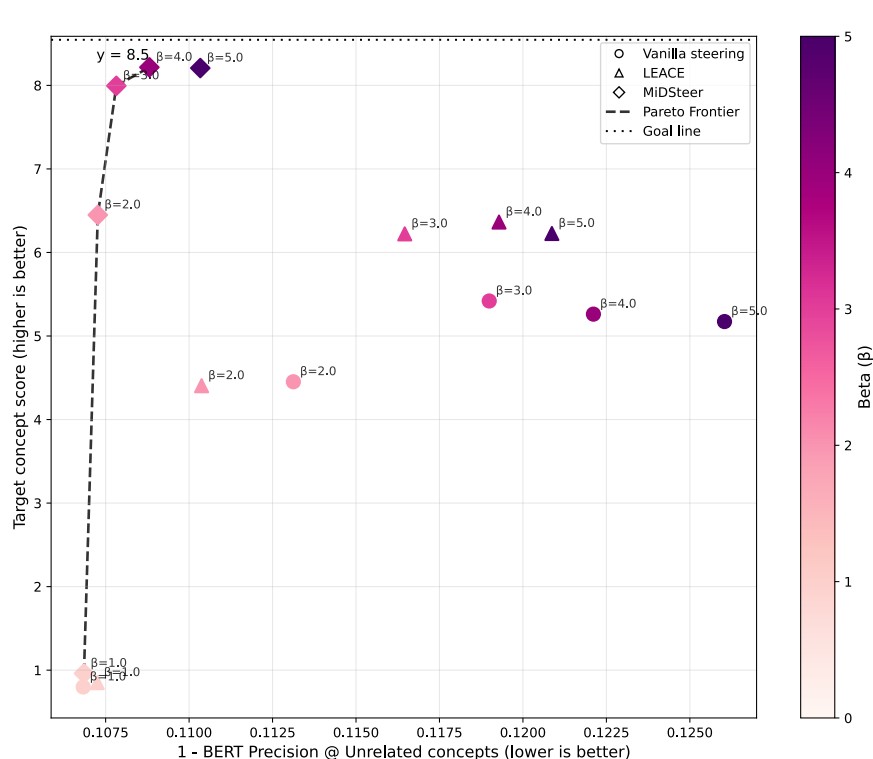

Figure 27: Pareto plot for concept flip on model qwen-14b

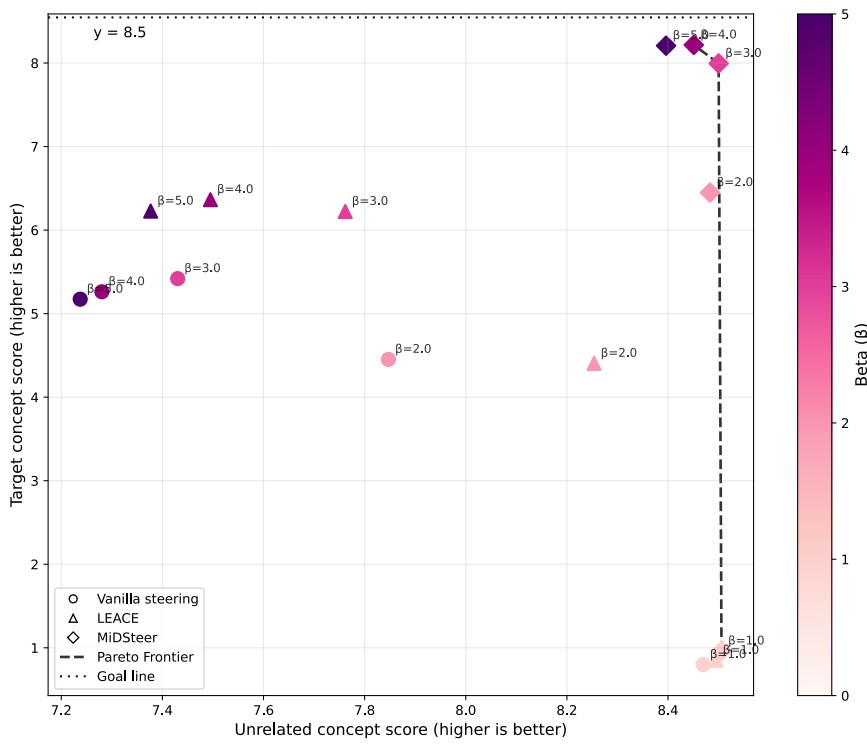

Figure 28: Pareto plot for concept flip on model qwen-14b

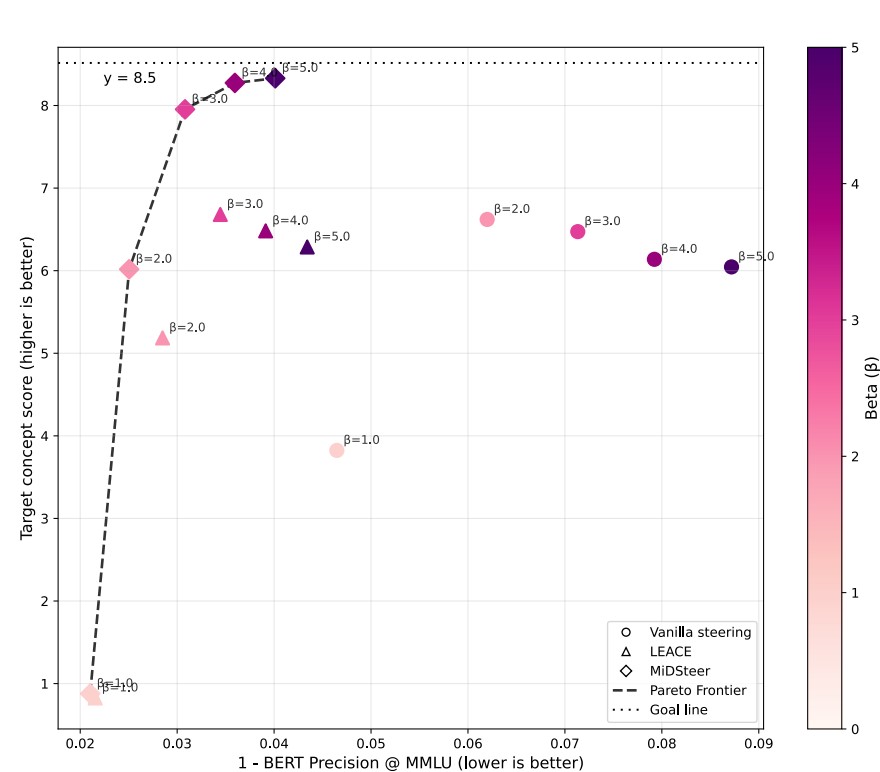

Figure 29: Pareto plot for concept flip on model qwen-7b

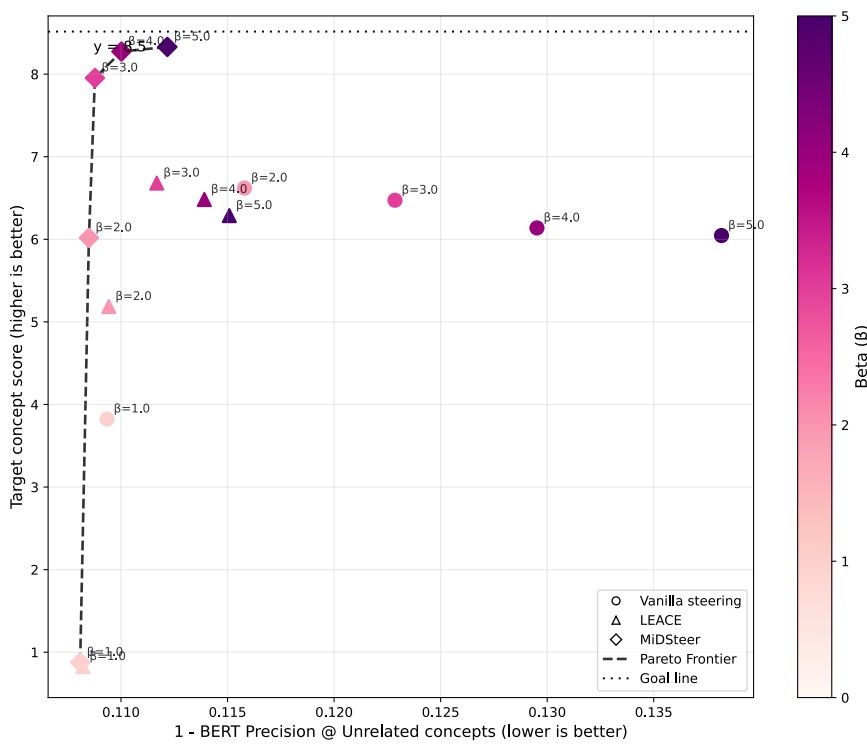

Figure 30: Pareto plot for concept flip on model qwen-7b

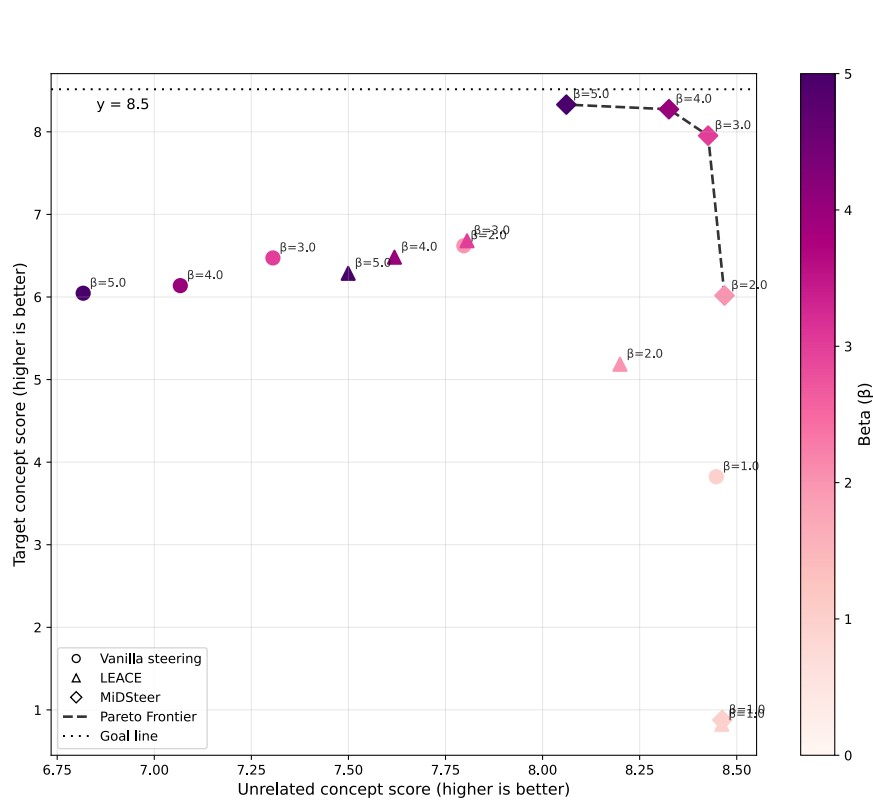

Figure 31: Pareto plot for concept flip on model qwen-7b

### 7.9.2 PARETO CHARTS FOR IMAGE DIFFUSION CONCEPT SWITCHING

In this section, we provide more Pareto charts for Diffusion Models concept switching. When switching concept $c_s$ to $c_t$, for each LLM model we provide 9 types of Pareto plots:

- FID score for unrelated concepts (horizontal axis) vs $\Delta$ Concept Score (CS) for the target $c_t$ and source $c_s$ concepts (fig. 33, 35)
- Average Concept Score (CS) for unrelated concepts $c_i$ (horizontal axis) vs $\Delta$ Concept Score (CS) for the target $c_t$ and source $c_s$ concepts (vertical axis) (fig. 32, 34)
- FID score for unrelated concepts (horizontal axis) vs Concept Score (CS) for the **source** $c_s$ concept (vertical axis) (fig. 37, 39)
- Average Concept Score (CS) for unrelated concepts $c_i$ (horizontal axis) vs Concept Score (CS) for the **source** $c_s$ concept (vertical axis) (fig.36, 38)
- FID score for unrelated concepts (horizontal axis) vs Concept Score (CS) for the **target** $c_s$ concept (vertical axis) (fig. 41, 43)
- Average Concept Score (CS) for unrelated concepts $c_i$ (horizontal axis) vs Concept Score (CS) for the **target** $c_s$ concept (vertical axis) (fig.40, 42)

In each case, we see clear superiority of MidSteer over other steering approaches.

We additionally provide detailed breakdown of scores for all $\beta$ values and all concepts $c_s, c_t, c_i$ in the tables in sec. 7.9.4

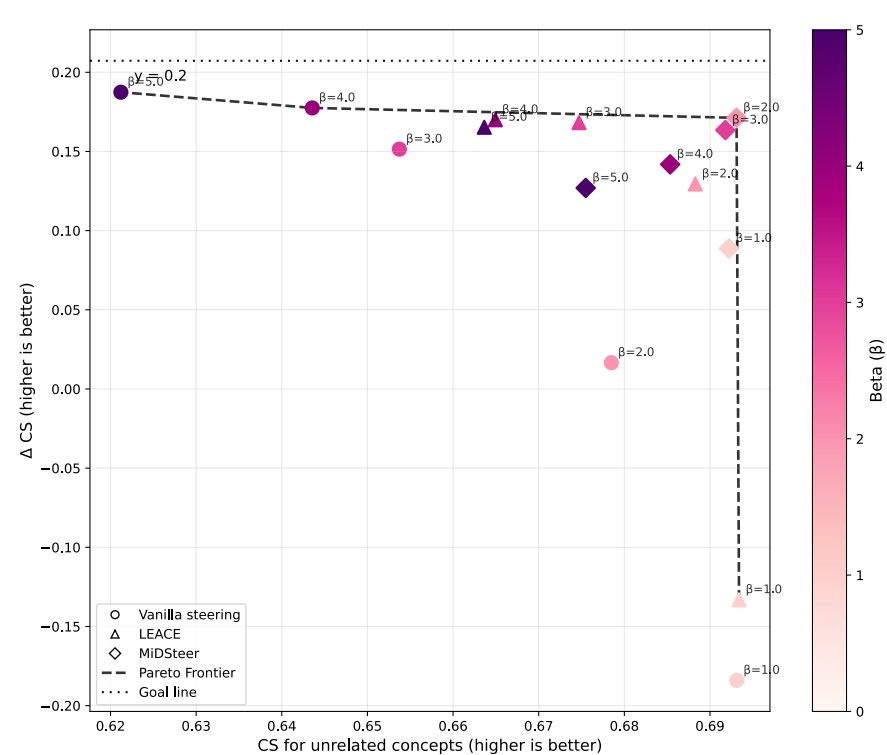

Figure 32: Pareto plot for concept flip on model SANA

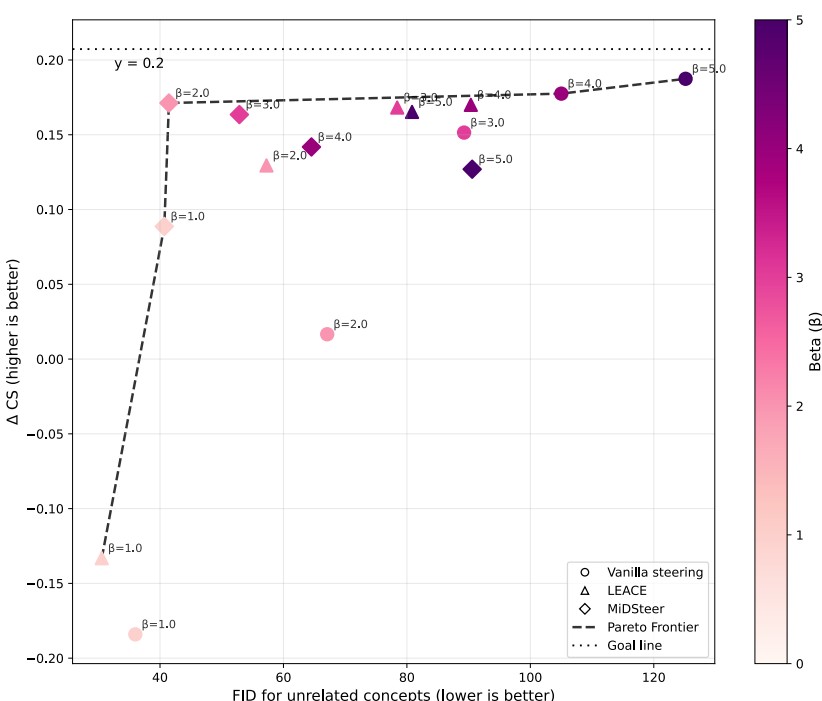

Figure 33: Pareto plot for concept flip on model SANA

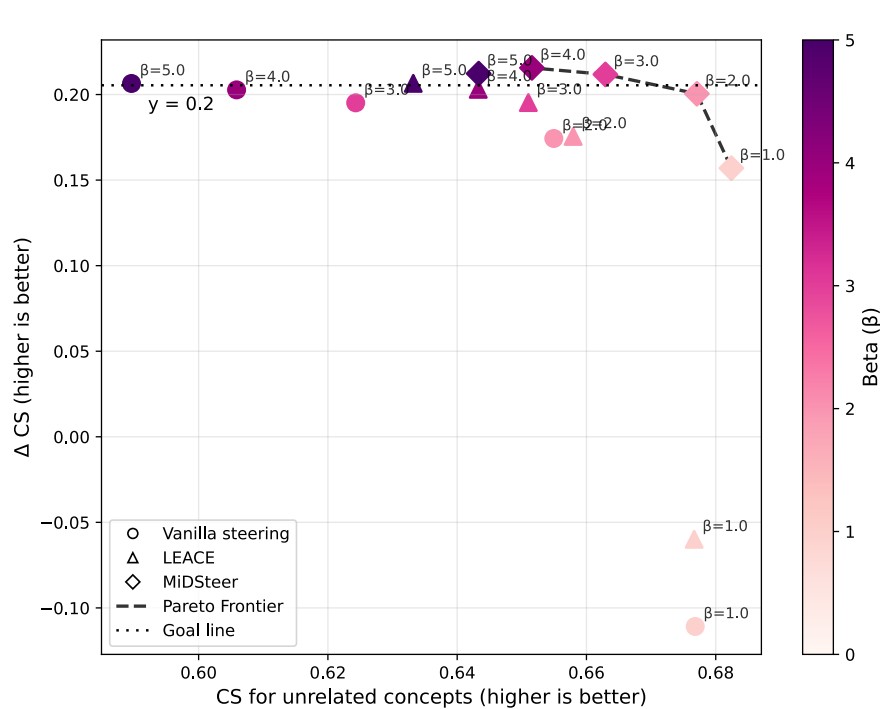

Figure 34: Pareto plot for concept flip on model SDXL

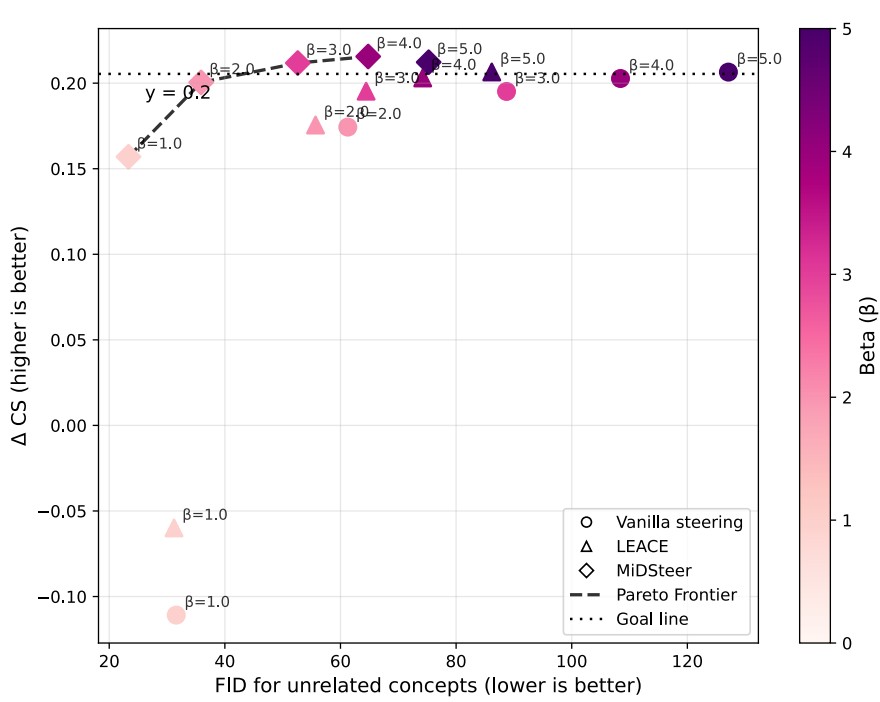

Figure 35: Pareto plot for concept flip on model SDXL

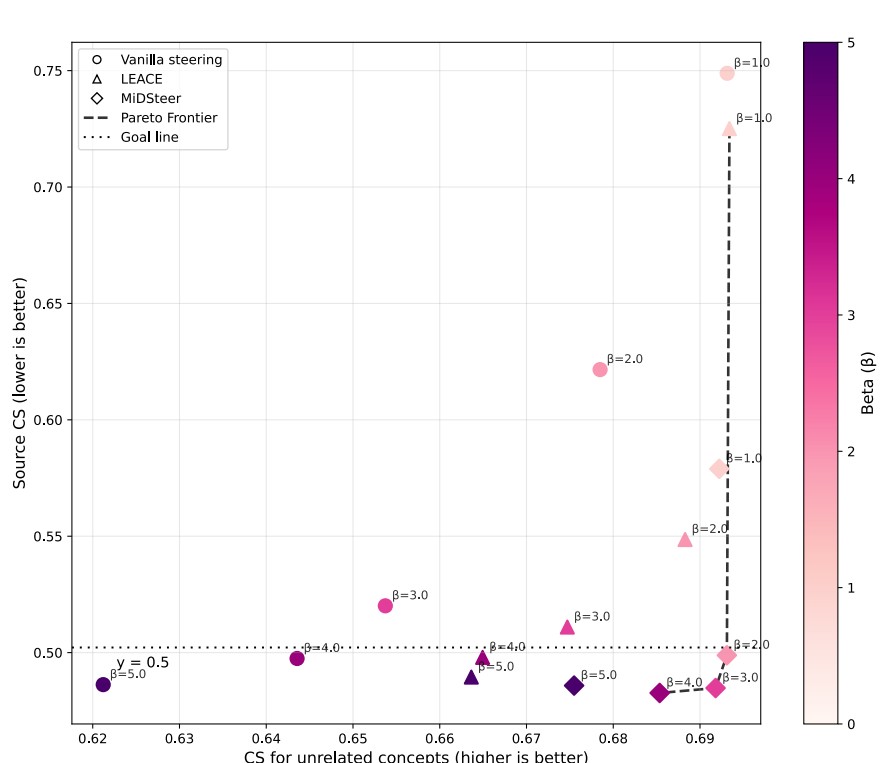

Figure 36: Pareto plot for concept flip on model SANA

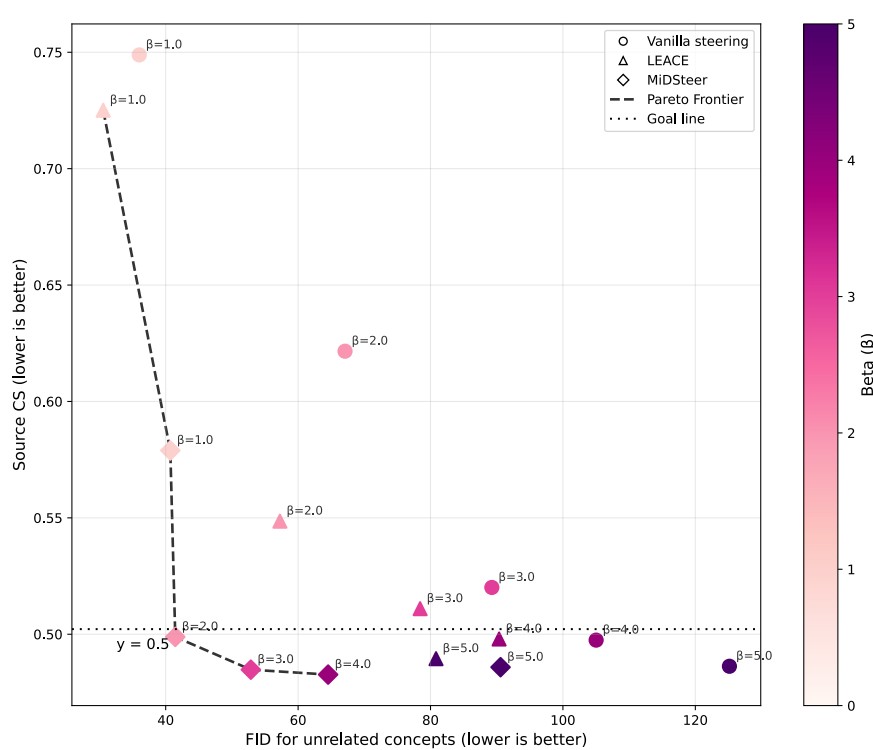

Figure 37: Pareto plot for concept flip on model SANA

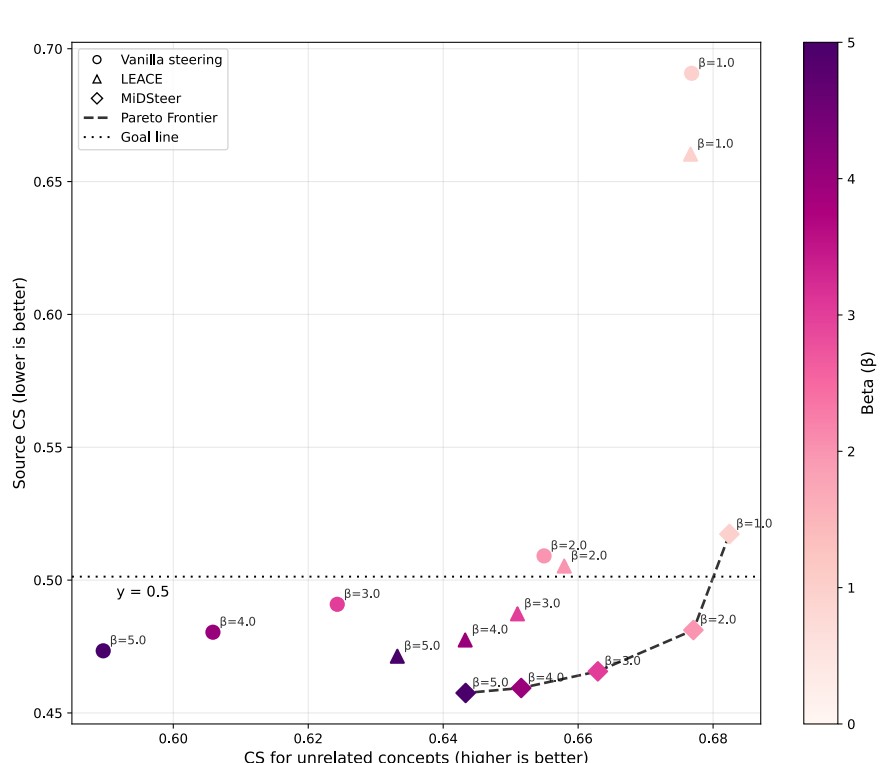

Figure 38: Pareto plot for concept flip on model SDXL

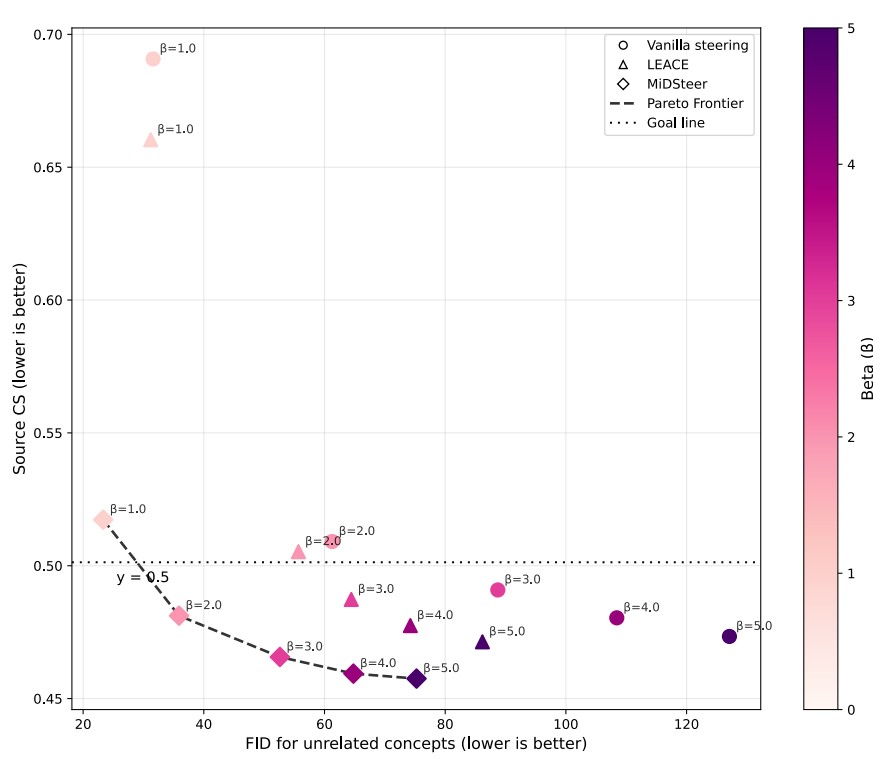

Figure 39: Pareto plot for concept flip on model SDXL

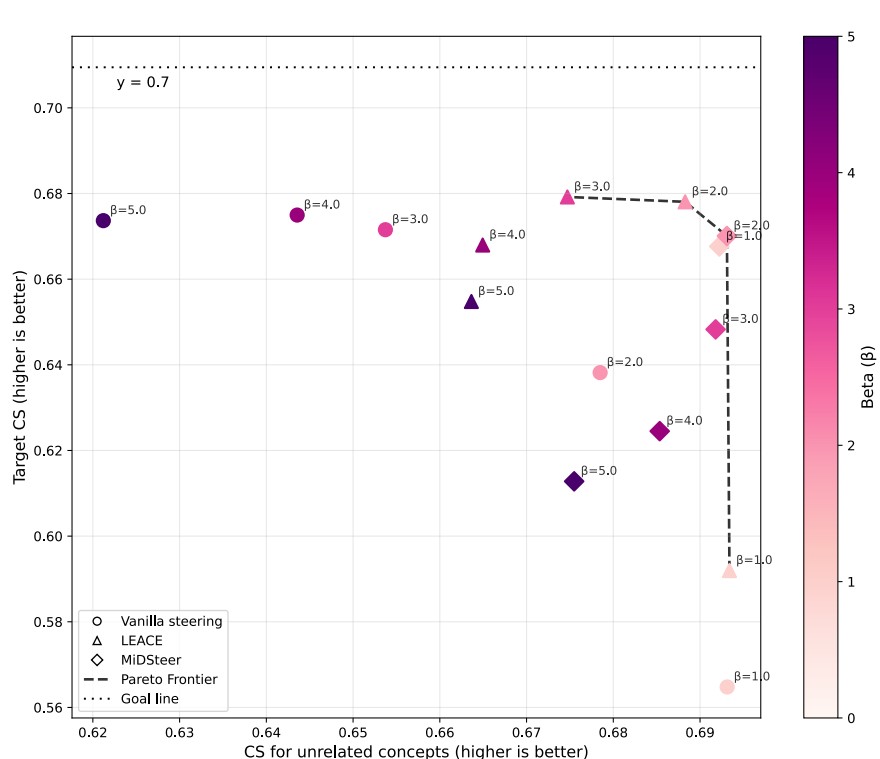

Figure 40: Pareto plot for concept flip on model SANA

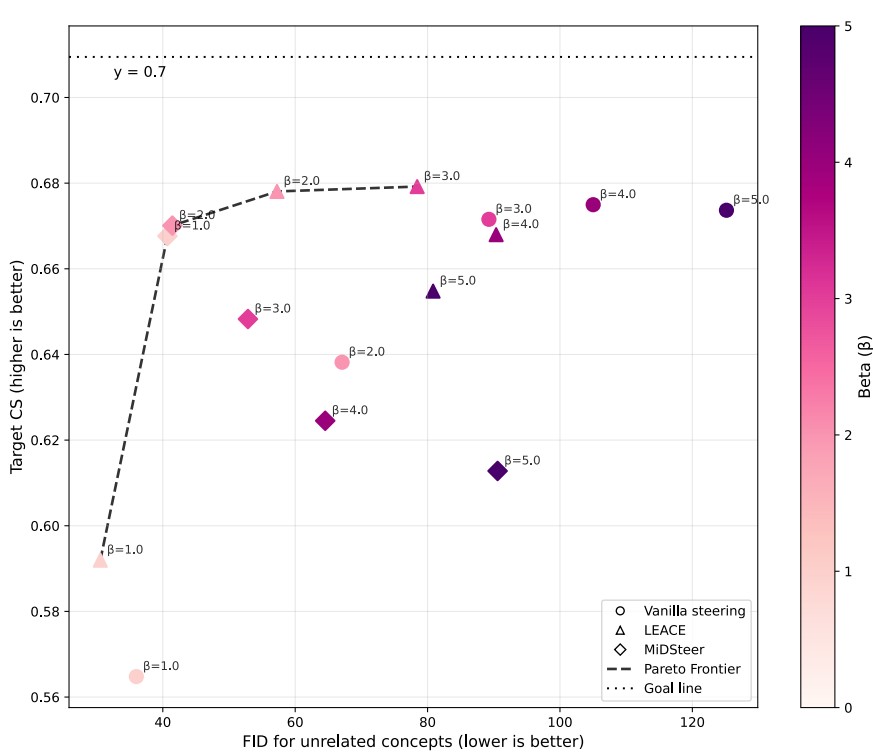

Figure 41: Pareto plot for concept flip on model SANA

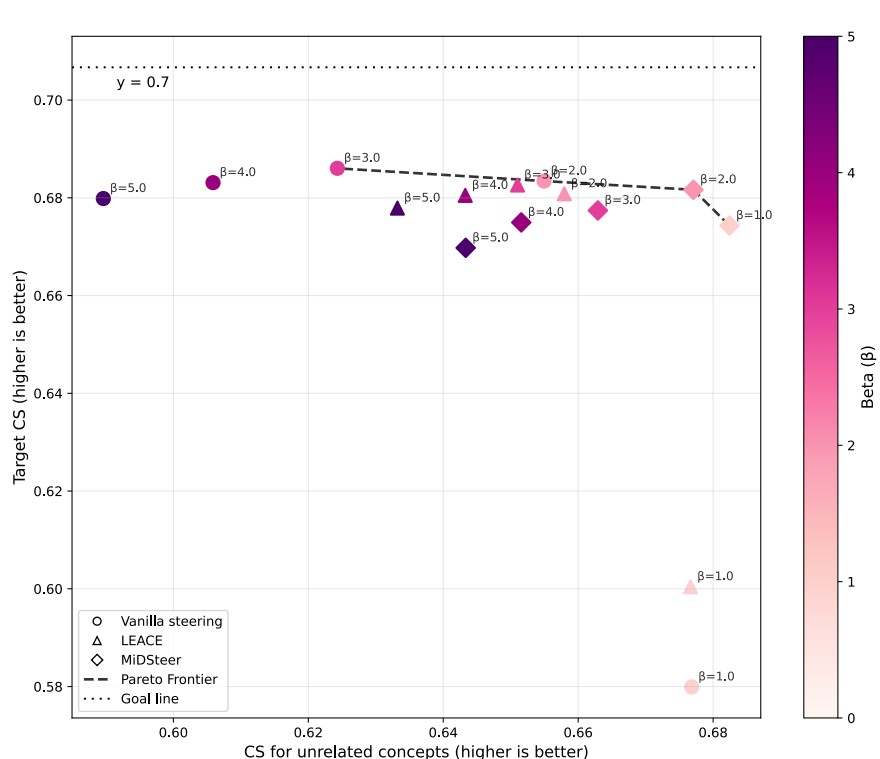

Figure 42: Pareto plot for concept flip on model SDXL

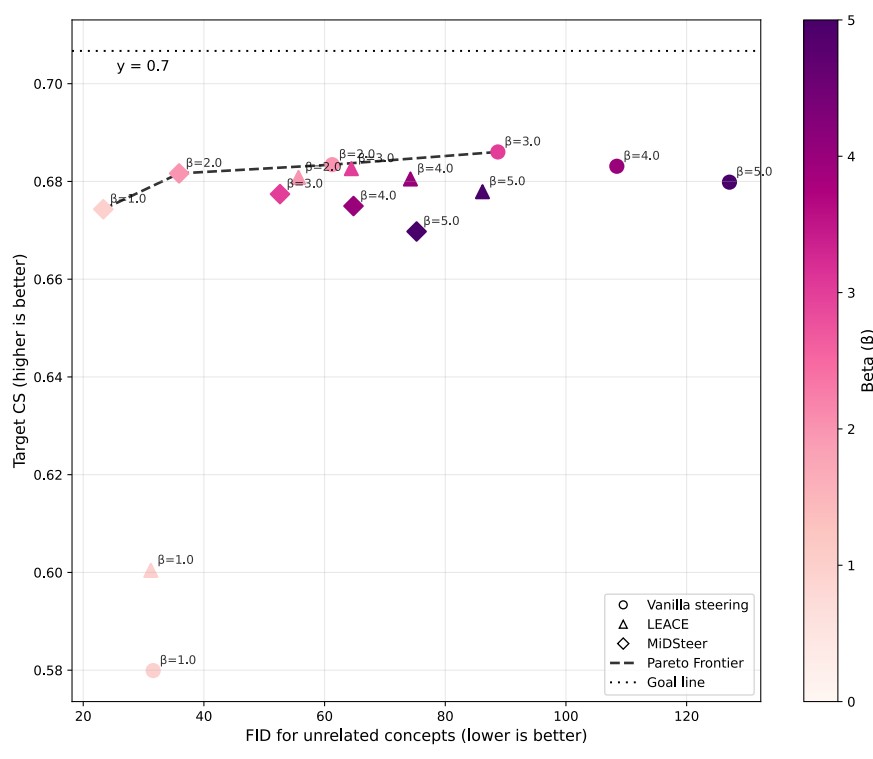

Figure 43: Pareto plot for concept flip on model SDXL

### 7.9.3 DETAILED RESULTS FOR LLM CONCEPT SWITCHING

In this section, in tab. 2,3,6,7,4,5 we provide detailed breakdown of scores for all $\beta$ values and all concepts $c_s, c_t, c_i$. Pareto plots in sec. 7.9.1 were created based on the scores provided in these tables.

Table 2: Model LLama2-7b, flipping from dogs to cats

| method | strength | dogs src-cs | tgt-cs | cats src-cs | tgt-cs | bertp | wolves cs | src-cs | tgt-cs | bertp | pigs cs | src-cs | tgt-cs | bertp | cows cs | src-cs | tgt-cs | bertp | legislators cs | src-cs | tgt-cs | bertp |
|---|---|---|---|---|---|---|---|---|---|---|---|---|---|---|---|---|---|---|---|---|---|---|
| CASteer | 1.0 | 8.6 | 1.1 | 2.9 | 7.5 | 0.91 | 8.5 | 7.8 | 1.5 | 0.91 | 8.5 | 1.5 | 0.4 | 0.91 | 8.4 | 0.7 | 0.3 | 0.91 | 8.5 | 0.3 | 0.2 | 0.90 |
| | 1.5 | 8.1 | 1.9 | 6.4 | 3.8 | 0.90 | 8.5 | 7.8 | 1.5 | 0.90 | 8.5 | 1.5 | 0.4 | 0.91 | 8.4 | 0.6 | 0.3 | 0.90 | 8.5 | 0.4 | 0.2 | 0.90 |
| | 2.0 | 5.9 | 3.9 | 7.1 | 2.8 | 0.89 | 8.4 | 7.8 | 1.4 | 0.90 | 8.5 | 1.5 | 0.4 | 0.90 | 8.4 | 0.7 | 0.3 | 0.90 | 8.5 | 0.4 | 0.2 | 0.90 |
| | 2.5 | 5.0 | 4.4 | 7.0 | 3.0 | 0.89 | 8.5 | 7.7 | 1.4 | 0.90 | 8.4 | 1.4 | 0.4 | 0.90 | 8.4 | 0.7 | 0.3 | 0.90 | 8.4 | 0.4 | 0.2 | 0.89 |
| | 3.0 | 4.8 | 4.6 | 6.6 | 2.8 | 0.89 | 8.5 | 7.7 | 1.4 | 0.90 | 8.4 | 1.4 | 0.4 | 0.90 | 8.4 | 0.7 | 0.3 | 0.90 | 8.5 | 0.3 | 0.2 | 0.89 |
| | 3.5 | 4.6 | 4.7 | 6.5 | 3.0 | 0.89 | 8.4 | 7.7 | 1.4 | 0.90 | 8.5 | 1.4 | 0.3 | 0.90 | 8.3 | 0.9 | 0.3 | 0.90 | 8.4 | 0.4 | 0.2 | 0.90 |
| | 4.0 | 4.6 | 4.7 | 6.5 | 2.9 | 0.89 | 8.4 | 7.7 | 1.5 | 0.90 | 8.4 | 1.4 | 0.4 | 0.90 | 8.2 | 0.9 | 0.3 | 0.90 | 8.4 | 0.4 | 0.2 | 0.89 |
| | 4.5 | 4.5 | 4.9 | 6.5 | 2.9 | 0.89 | 8.3 | 7.6 | 1.5 | 0.90 | 8.3 | 1.5 | 0.5 | 0.90 | 8.2 | 1.0 | 0.4 | 0.90 | 8.4 | 0.4 | 0.2 | 0.89 |
| | 5.0 | 4.4 | 4.9 | 6.5 | 2.7 | 0.88 | 8.3 | 7.6 | 1.5 | 0.89 | 8.2 | 1.5 | 0.6 | 0.89 | 7.9 | 1.2 | 0.5 | 0.89 | 8.2 | 0.4 | 0.2 | 0.89 |
| LEACE | 1.0 | 8.5 | 1.4 | 2.1 | 8.2 | 0.91 | 8.4 | 7.8 | 1.5 | 0.91 | 8.5 | 1.5 | 0.5 | 0.91 | 8.4 | 0.7 | 0.3 | 0.91 | 8.5 | 0.3 | 0.2 | 0.90 |
| | 1.5 | 5.4 | 4.5 | 3.4 | 6.6 | 0.91 | 8.4 | 7.8 | 1.5 | 0.91 | 8.5 | 1.4 | 0.4 | 0.91 | 8.4 | 0.7 | 0.3 | 0.91 | 8.5 | 0.3 | 0.1 | 0.90 |
| | 2.0 | 4.0 | 5.5 | 4.9 | 4.8 | 0.91 | 8.4 | 7.7 | 1.7 | 0.91 | 8.5 | 1.4 | 0.4 | 0.91 | 8.4 | 0.6 | 0.3 | 0.91 | 8.5 | 0.3 | 0.2 | 0.90 |
| | 2.5 | 3.8 | 5.6 | 5.3 | 4.1 | 0.90 | 8.4 | 7.7 | 1.5 | 0.91 | 8.5 | 1.5 | 0.5 | 0.91 | 8.4 | 0.7 | 0.3 | 0.91 | 8.5 | 0.3 | 0.2 | 0.90 |
| | 3.0 | 3.7 | 5.6 | 5.6 | 4.1 | 0.90 | 8.4 | 7.7 | 1.6 | 0.91 | 8.5 | 1.5 | 0.4 | 0.91 | 8.4 | 0.7 | 0.3 | 0.91 | 8.5 | 0.3 | 0.2 | 0.90 |
| | 3.5 | 3.7 | 5.7 | 5.5 | 3.9 | 0.90 | 8.4 | 7.7 | 1.5 | 0.91 | 8.4 | 1.5 | 0.5 | 0.91 | 8.4 | 0.7 | 0.3 | 0.91 | 8.5 | 0.3 | 0.2 | 0.90 |
| | 4.0 | 3.6 | 5.6 | 5.5 | 3.9 | 0.90 | 8.4 | 7.6 | 1.6 | 0.91 | 8.5 | 1.5 | 0.4 | 0.91 | 8.4 | 0.7 | 0.3 | 0.91 | 8.5 | 0.3 | 0.2 | 0.90 |
| | 4.5 | 3.6 | 5.6 | 5.5 | 3.9 | 0.90 | 8.4 | 7.7 | 1.6 | 0.91 | 8.5 | 1.4 | 0.4 | 0.91 | 8.4 | 0.7 | 0.3 | 0.91 | 8.5 | 0.3 | 0.2 | 0.90 |
| | 5.0 | 3.6 | 5.5 | 5.4 | 3.9 | 0.90 | 8.4 | 7.7 | 1.6 | 0.91 | 8.4 | 1.5 | 0.5 | 0.91 | 8.3 | 0.7 | 0.3 | 0.91 | 8.5 | 0.3 | 0.1 | 0.90 |
| mean_matching | 1.0 | 8.0 | 1.8 | 1.6 | 8.6 | 0.92 | 8.4 | 7.8 | 1.6 | 0.91 | 8.5 | 1.4 | 0.4 | 0.91 | 8.4 | 0.7 | 0.3 | 0.91 | 8.5 | 0.3 | 0.1 | 0.91 |
| | 1.5 | 4.1 | 6.1 | 1.5 | 8.6 | 0.91 | 8.5 | 7.8 | 1.6 | 0.91 | 8.4 | 1.5 | 0.5 | 0.91 | 8.4 | 0.6 | 0.3 | 0.91 | 8.5 | 0.4 | 0.2 | 0.90 |
| | 2.0 | 2.6 | 7.7 | 1.5 | 8.6 | 0.91 | 8.4 | 7.7 | 1.7 | 0.91 | 8.5 | 1.5 | 0.5 | 0.91 | 8.4 | 0.6 | 0.3 | 0.91 | 8.5 | 0.3 | 0.2 | 0.90 |
| | 2.5 | 2.1 | 8.2 | 1.5 | 8.6 | 0.91 | 8.3 | 7.5 | 1.8 | 0.91 | 8.4 | 1.4 | 0.5 | 0.91 | 8.2 | 0.7 | 0.4 | 0.91 | 8.4 | 0.3 | 0.2 | 0.90 |
| | 3.0 | 1.9 | 8.4 | 1.4 | 8.7 | 0.91 | 7.9 | 7.0 | 2.2 | 0.91 | 8.3 | 1.5 | 0.8 | 0.91 | 8.0 | 0.7 | 0.8 | 0.91 | 8.5 | 0.3 | 0.2 | 0.90 |
| | 3.5 | 1.7 | 8.5 | 1.3 | 8.7 | 0.91 | 7.4 | 6.5 | 3.2 | 0.90 | 7.8 | 1.6 | 1.5 | 0.90 | 7.7 | 0.9 | 1.4 | 0.90 | 8.4 | 0.3 | 0.2 | 0.90 |
| | 4.0 | 1.6 | 8.5 | 1.4 | 8.6 | 0.90 | 6.3 | 5.6 | 4.2 | 0.90 | 7.3 | 1.7 | 2.3 | 0.90 | 6.8 | 1.1 | 2.3 | 0.90 | 8.4 | 0.4 | 0.4 | 0.90 |
| | 4.5 | 1.6 | 8.6 | 1.3 | 8.7 | 0.90 | 5.4 | 4.8 | 5.1 | 0.89 | 6.5 | 1.8 | 3.2 | 0.90 | 5.8 | 1.2 | 3.8 | 0.90 | 8.3 | 0.5 | 0.9 | 0.90 |
| | 5.0 | 1.6 | 8.6 | 1.4 | 8.7 | 0.90 | 4.6 | 4.1 | 5.8 | 0.89 | 5.5 | 2.0 | 4.4 | 0.89 | 4.6 | 1.2 | 4.8 | 0.89 | 8.1 | 0.6 | 1.4 | 0.90 |

Table 3: Model LLama2-7b, flipping from horses to motorcycles

| method | strength | horses src-cs | tgt-cs | motorcycles src-cs | tgt-cs | bertp | cows cs | src-cs | tgt-cs | bertp | pigs cs | src-cs | tgt-cs | bertp | dogs cs | src-cs | tgt-cs | bertp | legislators cs | src-cs | tgt-cs | bertp |
|---|---|---|---|---|---|---|---|---|---|---|---|---|---|---|---|---|---|---|---|---|---|---|
| CASteer | 1.0 | 8.6 | 0.7 | 1.3 | 8.5 | 0.92 | 8.4 | 1.7 | 0.1 | 0.91 | 8.5 | 1.2 | 0.1 | 0.91 | 8.6 | 1.5 | 0.2 | 0.92 | 8.5 | 0.3 | 0.3 | 0.90 |
| | 1.5 | 6.5 | 2.9 | 1.4 | 8.5 | 0.92 | 8.4 | 1.7 | 0.1 | 0.91 | 8.4 | 1.2 | 0.1 | 0.91 | 8.6 | 1.5 | 0.2 | 0.91 | 8.5 | 0.3 | 0.3 | 0.90 |
| | 2.0 | 2.7 | 6.9 | 2.1 | 7.9 | 0.91 | 8.3 | 1.6 | 0.1 | 0.91 | 8.4 | 1.2 | 0.1 | 0.91 | 8.6 | 1.5 | 0.2 | 0.91 | 8.5 | 0.3 | 0.3 | 0.90 |
| | 2.5 | 2.6 | 7.2 | 3.7 | 5.5 | 0.91 | 8.1 | 1.4 | 0.2 | 0.90 | 8.3 | 1.2 | 0.1 | 0.91 | 8.5 | 1.5 | 0.2 | 0.91 | 8.5 | 0.3 | 0.3 | 0.90 |
| | 3.0 | 2.9 | 6.8 | 4.4 | 4.2 | 0.90 | 7.8 | 1.3 | 0.3 | 0.90 | 8.2 | 1.2 | 0.1 | 0.90 | 8.6 | 1.4 | 0.2 | 0.91 | 8.5 | 0.3 | 0.2 | 0.90 |
| | 3.5 | 3.0 | 6.5 | 4.7 | 3.6 | 0.90 | 7.2 | 1.3 | 0.3 | 0.90 | 8.1 | 1.1 | 0.1 | 0.90 | 8.5 | 1.3 | 0.2 | 0.91 | 8.4 | 0.3 | 0.3 | 0.90 |
| | 4.0 | 3.0 | 6.1 | 4.6 | 3.4 | 0.89 | 6.9 | 1.1 | 0.4 | 0.90 | 8.1 | 1.1 | 0.2 | 0.90 | 8.3 | 1.4 | 0.3 | 0.90 | 8.5 | 0.3 | 0.3 | 0.90 |
| | 4.5 | 3.1 | 5.8 | 5.0 | 3.2 | 0.89 | 6.5 | 1.1 | 0.4 | 0.89 | 7.9 | 1.1 | 0.2 | 0.90 | 8.2 | 1.4 | 0.3 | 0.90 | 8.5 | 0.4 | 0.4 | 0.90 |
| | 5.0 | 3.0 | 5.7 | 4.8 | 3.1 | 0.89 | 6.4 | 1.1 | 0.5 | 0.89 | 7.7 | 1.0 | 0.3 | 0.89 | 8.0 | 1.3 | 0.3 | 0.90 | 8.4 | 0.4 | 0.4 | 0.90 |
| LEACE | 1.0 | 8.5 | 0.7 | 1.2 | 8.5 | 0.92 | 8.4 | 1.7 | 0.1 | 0.91 | 8.4 | 1.2 | 0.1 | 0.91 | 8.6 | 1.5 | 0.1 | 0.92 | 8.5 | 0.3 | 0.3 | 0.90 |
| | 1.5 | 8.2 | 1.0 | 1.7 | 8.2 | 0.92 | 8.4 | 1.6 | 0.2 | 0.91 | 8.4 | 1.3 | 0.1 | 0.91 | 8.6 | 1.5 | 0.2 | 0.92 | 8.5 | 0.4 | 0.3 | 0.90 |
| | 2.0 | 5.4 | 3.5 | 4.0 | 5.4 | 0.91 | 8.3 | 1.7 | 0.2 | 0.91 | 8.4 | 1.3 | 0.1 | 0.91 | 8.6 | 1.5 | 0.2 | 0.92 | 8.5 | 0.3 | 0.2 | 0.90 |
| | 2.5 | 4.0 | 4.9 | 4.9 | 3.9 | 0.90 | 8.3 | 1.6 | 0.2 | 0.91 | 8.4 | 1.2 | 0.1 | 0.91 | 8.6 | 1.5 | 0.2 | 0.91 | 8.5 | 0.3 | 0.2 | 0.91 |
| | 3.0 | 3.7 | 5.2 | 5.5 | 3.2 | 0.90 | 8.2 | 1.5 | 0.2 | 0.91 | 8.4 | 1.2 | 0.1 | 0.91 | 8.6 | 1.4 | 0.2 | 0.91 | 8.5 | 0.3 | 0.3 | 0.90 |
| | 3.5 | 3.6 | 5.3 | 5.4 | 3.0 | 0.90 | 8.1 | 1.4 | 0.2 | 0.91 | 8.3 | 1.2 | 0.1 | 0.91 | 8.6 | 1.4 | 0.2 | 0.91 | 8.5 | 0.3 | 0.2 | 0.90 |
| | 4.0 | 3.5 | 5.5 | 5.5 | 2.9 | 0.90 | 8.0 | 1.3 | 0.2 | 0.90 | 8.3 | 1.1 | 0.1 | 0.91 | 8.6 | 1.5 | 0.2 | 0.91 | 8.5 | 0.4 | 0.3 | 0.90 |
| | 4.5 | 3.4 | 5.2 | 5.5 | 2.8 | 0.90 | 7.7 | 1.2 | 0.2 | 0.90 | 8.2 | 1.2 | 0.1 | 0.91 | 8.5 | 1.5 | 0.2 | 0.91 | 8.5 | 0.3 | 0.3 | 0.90 |
| | 5.0 | 3.4 | 5.3 | 5.6 | 2.9 | 0.90 | 7.6 | 1.3 | 0.3 | 0.90 | 8.2 | 1.1 | 0.1 | 0.91 | 8.5 | 1.5 | 0.2 | 0.91 | 8.5 | 0.4 | 0.3 | 0.90 |
| mean_matching | 1.0 | 8.6 | 0.7 | 1.2 | 8.5 | 0.92 | 8.4 | 1.6 | 0.1 | 0.91 | 8.5 | 1.2 | 0.1 | 0.91 | 8.6 | 1.4 | 0.2 | 0.92 | 8.5 | 0.3 | 0.3 | 0.90 |
| | 1.5 | 5.9 | 3.5 | 1.2 | 8.5 | 0.92 | 8.4 | 1.6 | 0.1 | 0.91 | 8.4 | 1.2 | 0.1 | 0.91 | 8.6 | 1.6 | 0.2 | 0.92 | 8.4 | 0.3 | 0.3 | 0.90 |
| | 2.0 | 2.6 | 7.2 | 1.2 | 8.5 | 0.92 | 8.3 | 1.5 | 0.1 | 0.91 | 8.4 | 1.2 | 0.1 | 0.91 | 8.6 | 1.5 | 0.2 | 0.91 | 8.4 | 0.3 | 0.3 | 0.90 |
| | 2.5 | 1.8 | 8.2 | 1.2 | 8.5 | 0.92 | 8.3 | 1.6 | 0.2 | 0.91 | 8.4 | 1.2 | 0.1 | 0.91 | 8.6 | 1.5 | 0.2 | 0.91 | 8.5 | 0.3 | 0.3 | 0.90 |
| | 3.0 | 1.7 | 8.4 | 1.2 | 8.5 | 0.92 | 8.3 | 1.6 | 0.2 | 0.91 | 8.2 | 1.1 | 0.1 | 0.91 | 8.6 | 1.6 | 0.2 | 0.91 | 8.5 | 0.3 | 0.3 | 0.90 |
| | 3.5 | 1.6 | 8.4 | 1.2 | 8.5 | 0.92 | 8.0 | 1.4 | 0.4 | 0.90 | 8.2 | 1.0 | 0.2 | 0.90 | 8.5 | 1.5 | 0.2 | 0.91 | 8.4 | 0.4 | 0.3 | 0.90 |
| | 4.0 | 1.6 | 8.4 | 1.2 | 8.5 | 0.91 | 7.6 | 1.3 | 0.7 | 0.90 | 8.0 | 1.1 | 0.4 | 0.90 | 8.4 | 1.4 | 0.2 | 0.91 | 8.4 | 0.3 | 0.3 | 0.90 |
| | 4.5 | 1.6 | 8.5 | 1.2 | 8.5 | 0.91 | 7.0 | 1.3 | 1.4 | 0.90 | 7.6 | 1.0 | 0.6 | 0.90 | 8.3 | 1.4 | 0.4 | 0.91 | 8.5 | 0.3 | 0.3 | 0.90 |
| | 5.0 | 1.4 | 8.5 | 1.2 | 8.5 | 0.91 | 6.1 | 1.2 | 2.2 | 0.89 | 7.3 | 0.9 | 0.9 | 0.90 | 8.2 | 1.4 | 0.5 | 0.90 | 8.4 | 0.3 | 0.3 | 0.90 |

Table 4: Model Qwen2.5-7b, flipping from dogs to cats

| method | strength | dogs src-cs | tgt-cs | cats src-cs | tgt-cs | bertp | wolves cs | src-cs | tgt-cs | bertp | pigs cs | src-cs | tgt-cs | bertp | cows cs | src-cs | tgt-cs | bertp | legislators cs | src-cs | tgt-cs | bertp |
|---|---|---|---|---|---|---|---|---|---|---|---|---|---|---|---|---|---|---|---|---|---|---|
| CASteer | 1.0 | 5.6 | 6.9 | 1.9 | 8.4 | 0.89 | 8.3 | 7.6 | 2.0 | 0.89 | 8.4 | 1.6 | 0.6 | 0.89 | 8.4 | 0.8 | 0.4 | 0.89 | 8.5 | 0.3 | 0.2 | 0.89 |
| | 2.0 | 3.2 | 7.5 | 2.2 | 7.8 | 0.88 | 7.6 | 6.4 | 3.7 | 0.88 | 8.1 | 1.6 | 1.9 | 0.88 | 7.9 | 1.1 | 1.8 | 0.89 | 8.4 | 0.5 | 0.9 | 0.89 |
| | 3.0 | 3.4 | 7.5 | 2.3 | 7.6 | 0.87 | 7.3 | 6.2 | 3.9 | 0.87 | 7.2 | 1.7 | 3.0 | 0.88 | 7.0 | 1.4 | 3.0 | 0.88 | 8.4 | 0.7 | 2.0 | 0.88 |
| | 4.0 | 3.0 | 7.1 | 2.4 | 7.2 | 0.86 | 7.2 | 6.2 | 3.7 | 0.87 | 7.4 | 1.6 | 2.2 | 0.87 | 7.1 | 1.3 | 2.8 | 0.87 | 8.3 | 0.6 | 1.6 | 0.88 |
| | 5.0 | 2.9 | 6.8 | 2.4 | 6.6 | 0.85 | 5.9 | 5.2 | 4.3 | 0.85 | 7.3 | 1.6 | 1.7 | 0.86 | 7.5 | 1.2 | 1.8 | 0.87 | 8.3 | 0.5 | 1.0 | 0.87 |
| LEACE | 1.0 | 8.7 | 1.0 | 1.8 | 8.5 | 0.89 | 8.3 | 7.7 | 1.6 | 0.89 | 8.4 | 1.6 | 0.4 | 0.89 | 8.4 | 0.9 | 0.3 | 0.89 | 8.5 | 0.3 | 0.2 | 0.89 |
| | 2.0 | 7.4 | 4.2 | 6.0 | 6.7 | 0.88 | 8.3 | 7.6 | 1.6 | 0.89 | 8.4 | 1.4 | 0.4 | 0.89 | 8.4 | 0.9 | 0.3 | 0.89 | 8.5 | 0.3 | 0.2 | 0.89 |
| | 3.0 | 6.3 | 6.7 | 7.1 | 5.5 | 0.88 | 8.3 | 7.6 | 1.7 | 0.89 | 8.4 | 1.6 | 0.4 | 0.89 | 8.4 | 0.9 | 0.3 | 0.89 | 8.5 | 0.3 | 0.2 | 0.89 |
| | 4.0 | 5.8 | 6.6 | 6.9 | 4.7 | 0.87 | 8.3 | 7.5 | 1.7 | 0.89 | 8.4 | 1.5 | 0.4 | 0.89 | 8.4 | 0.8 | 0.3 | 0.89 | 8.5 | 0.3 | 0.2 | 0.89 |
| | 5.0 | 5.3 | 6.5 | 6.7 | 4.3 | 0.87 | 8.3 | 7.6 | 1.8 | 0.89 | 8.4 | 1.5 | 0.4 | 0.89 | 8.4 | 0.8 | 0.3 | 0.89 | 8.5 | 0.4 | 0.2 | 0.89 |
| mean_matching | 1.0 | 8.7 | 1.0 | 1.7 | 8.5 | 0.89 | 8.3 | 7.6 | 1.6 | 0.89 | 8.5 | 1.5 | 0.5 | 0.89 | 8.4 | 0.8 | 0.3 | 0.89 | 8.5 | 0.4 | 0.2 | 0.89 |
| | 2.0 | 7.3 | 5.0 | 1.7 | 8.5 | 0.89 | 8.4 | 7.6 | 1.7 | 0.89 | 8.5 | 1.5 | 0.4 | 0.89 | 8.4 | 0.8 | 0.3 | 0.89 | 8.5 | 0.3 | 0.2 | 0.89 |
| | 3.0 | 5.7 | 7.7 | 1.7 | 8.5 | 0.89 | 8.3 | 7.5 | 2.0 | 0.89 | 8.4 | 1.4 | 0.6 | 0.89 | 8.4 | 0.8 | 0.5 | 0.89 | 8.5 | 0.3 | 0.2 | 0.89 |
| | 4.0 | 5.0 | 8.2 | 1.7 | 8.5 | 0.89 | 8.1 | 7.4 | 2.5 | 0.89 | 8.3 | 1.6 | 1.0 | 0.89 | 8.3 | 1.0 | 0.7 | 0.89 | 8.5 | 0.4 | 0.3 | 0.89 |
| | 5.0 | 4.3 | 8.3 | 1.7 | 8.5 | 0.89 | 7.8 | 6.9 | 3.2 | 0.89 | 8.3 | 1.7 | 1.1 | 0.89 | 8.2 | 1.1 | 1.0 | 0.89 | 8.4 | 0.4 | 0.3 | 0.89 |

Table 5: Model Qwen2.5-7b, flipping from horses to motorcycles

| method | strength | horses | | motorcycles | | | cows | | | | pigs | | | | dogs | | | | legislators | | | |
|---|---|---|---|---|---|---|---|---|---|---|---|---|---|---|---|---|---|---|---|---|---|---|
| | | src-cs | tgt-cs | src-cs | tgt-cs | bertp | cs | src-cs | tgt-cs | bertp | cs | src-cs | tgt-cs | bertp | cs | src-cs | tgt-cs | bertp | cs | src-cs | tgt-cs | bertp |
| CASteer | 1.0 | 8.6 | 0.8 | 1.2 | 8.4 | 0.90 | 8.4 | 1.8 | 0.2 | 0.89 | 8.4 | 1.1 | 0.1 | 0.89 | 8.6 | 1.4 | 0.3 | 0.89 | 8.5 | 0.4 | 0.4 | 0.89 |
| | 2.0 | 4.3 | 5.7 | 5.7 | 4.4 | 0.87 | 8.2 | 1.6 | 0.3 | 0.89 | 8.3 | 0.9 | 0.1 | 0.88 | 8.6 | 1.3 | 0.2 | 0.89 | 8.5 | 0.4 | 0.3 | 0.89 |
| | 3.0 | 4.1 | 5.5 | 6.1 | 3.2 | 0.86 | 7.4 | 1.3 | 0.8 | 0.88 | 8.0 | 0.8 | 0.3 | 0.87 | 8.5 | 1.3 | 0.3 | 0.88 | 8.5 | 0.3 | 0.3 | 0.89 |
| | 4.0 | 4.2 | 5.2 | 6.2 | 2.8 | 0.86 | 6.6 | 1.5 | 1.3 | 0.86 | 7.5 | 0.9 | 0.4 | 0.87 | 8.1 | 1.3 | 0.4 | 0.88 | 8.4 | 0.3 | 0.3 | 0.89 |
| | 5.0 | 3.8 | 5.3 | 6.4 | 2.9 | 0.85 | 6.3 | 1.4 | 1.2 | 0.86 | 7.1 | 0.9 | 0.6 | 0.86 | 7.8 | 1.3 | 0.5 | 0.87 | 8.4 | 0.3 | 0.3 | 0.88 |
| LEACE | 1.0 | 8.6 | 0.7 | 1.0 | 8.4 | 0.90 | 8.4 | 1.9 | 0.2 | 0.89 | 8.4 | 1.2 | 0.1 | 0.89 | 8.7 | 1.5 | 0.2 | 0.89 | 8.5 | 0.4 | 0.3 | 0.89 |
| | 2.0 | 4.4 | 6.2 | 2.8 | 7.7 | 0.89 | 8.4 | 1.8 | 0.2 | 0.89 | 8.4 | 1.1 | 0.1 | 0.89 | 8.7 | 1.5 | 0.2 | 0.89 | 8.6 | 0.4 | 0.3 | 0.89 |
| | 3.0 | 3.1 | 6.7 | 5.3 | 5.0 | 0.88 | 8.3 | 1.7 | 0.2 | 0.89 | 8.4 | 1.0 | 0.1 | 0.89 | 8.7 | 1.4 | 0.2 | 0.89 | 8.5 | 0.4 | 0.3 | 0.89 |
| | 4.0 | 3.2 | 6.3 | 5.5 | 4.1 | 0.87 | 8.2 | 1.6 | 0.4 | 0.89 | 8.3 | 1.0 | 0.1 | 0.89 | 8.6 | 1.4 | 0.2 | 0.89 | 8.5 | 0.4 | 0.4 | 0.89 |
| | 5.0 | 3.2 | 6.1 | 5.6 | 3.8 | 0.87 | 7.9 | 1.5 | 0.7 | 0.88 | 8.3 | 0.9 | 0.2 | 0.88 | 8.6 | 1.3 | 0.2 | 0.89 | 8.5 | 0.4 | 0.4 | 0.89 |
| mean_matching | 1.0 | 8.6 | 0.8 | 0.9 | 8.5 | 0.90 | 8.3 | 1.9 | 0.2 | 0.89 | 8.4 | 1.1 | 0.1 | 0.89 | 8.7 | 1.5 | 0.2 | 0.89 | 8.5 | 0.3 | 0.3 | 0.89 |
| | 2.0 | 2.9 | 7.1 | 0.8 | 8.5 | 0.90 | 8.4 | 1.8 | 0.2 | 0.89 | 8.4 | 1.1 | 0.1 | 0.89 | 8.7 | 1.4 | 0.2 | 0.89 | 8.5 | 0.3 | 0.3 | 0.89 |
| | 3.0 | 1.4 | 8.2 | 0.9 | 8.5 | 0.90 | 8.2 | 1.7 | 0.3 | 0.89 | 8.4 | 1.0 | 0.2 | 0.89 | 8.6 | 1.4 | 0.3 | 0.89 | 8.5 | 0.3 | 0.3 | 0.89 |
| | 4.0 | 1.3 | 8.4 | 0.8 | 8.5 | 0.89 | 7.7 | 1.7 | 1.0 | 0.89 | 8.2 | 0.9 | 0.3 | 0.88 | 8.6 | 1.2 | 0.3 | 0.89 | 8.4 | 0.3 | 0.4 | 0.89 |
| | 5.0 | 1.2 | 8.4 | 0.9 | 8.6 | 0.90 | 6.3 | 1.5 | 2.8 | 0.88 | 7.8 | 1.0 | 1.0 | 0.88 | 8.2 | 1.4 | 0.8 | 0.89 | 8.5 | 0.3 | 0.4 | 0.89 |

Table 6: Model Qwen2.5-14b, flipping from dogs to cats

| method | strength | dogs | | cats | | | wolves | | | | pigs | | | | cows | | | | legislators | | | |
|---|---|---|---|---|---|---|---|---|---|---|---|---|---|---|---|---|---|---|---|---|---|---|
| | | src-cs | tgt-cs | src-cs | tgt-cs | bertp | cs | src-cs | tgt-cs | bertp | cs | src-cs | tgt-cs | bertp | cs | src-cs | tgt-cs | bertp | cs | src-cs | tgt-cs | bertp |
| CASteer | 1.0 | 8.7 | 0.9 | 2.7 | 8.2 | 0.89 | 8.4 | 7.7 | 1.3 | 0.89 | 8.5 | 1.4 | 0.4 | 0.89 | 8.4 | 0.8 | 0.3 | 0.90 | 8.5 | 0.3 | 0.2 | 0.89 |
| | 2.0 | 7.8 | 4.5 | 7.3 | 4.9 | 0.86 | 8.4 | 7.7 | 1.4 | 0.89 | 8.5 | 1.4 | 0.4 | 0.89 | 8.4 | 0.8 | 0.2 | 0.89 | 8.5 | 0.3 | 0.2 | 0.89 |
| | 3.0 | 7.0 | 5.9 | 7.3 | 4.3 | 0.86 | 8.4 | 7.6 | 1.3 | 0.89 | 8.5 | 1.3 | 0.3 | 0.89 | 8.4 | 0.8 | 0.3 | 0.89 | 8.5 | 0.4 | 0.2 | 0.89 |
| | 4.0 | 6.8 | 6.2 | 7.2 | 4.6 | 0.86 | 8.4 | 7.6 | 1.3 | 0.89 | 8.5 | 1.3 | 0.4 | 0.89 | 8.4 | 0.8 | 0.3 | 0.89 | 8.5 | 0.4 | 0.2 | 0.89 |
| | 5.0 | 6.8 | 6.1 | 7.3 | 5.1 | 0.86 | 8.3 | 7.5 | 1.3 | 0.89 | 8.5 | 1.3 | 0.4 | 0.89 | 8.3 | 0.8 | 0.2 | 0.89 | 8.5 | 0.4 | 0.2 | 0.89 |
| LEACE | 1.0 | 8.7 | 1.0 | 1.9 | 8.4 | 0.89 | 8.4 | 7.7 | 1.3 | 0.89 | 8.5 | 1.5 | 0.4 | 0.89 | 8.4 | 0.8 | 0.3 | 0.89 | 8.5 | 0.3 | 0.2 | 0.89 |
| | 2.0 | 7.9 | 4.2 | 6.0 | 7.3 | 0.88 | 8.4 | 7.7 | 1.3 | 0.89 | 8.5 | 1.4 | 0.4 | 0.89 | 8.4 | 0.8 | 0.3 | 0.89 | 8.5 | 0.3 | 0.2 | 0.89 |
| | 3.0 | 6.9 | 6.5 | 7.0 | 5.4 | 0.86 | 8.4 | 7.6 | 1.4 | 0.89 | 8.5 | 1.3 | 0.3 | 0.89 | 8.4 | 0.8 | 0.3 | 0.90 | 8.5 | 0.3 | 0.1 | 0.89 |
| | 4.0 | 6.2 | 6.7 | 6.9 | 4.2 | 0.86 | 8.4 | 7.7 | 1.4 | 0.89 | 8.5 | 1.4 | 0.4 | 0.89 | 8.4 | 0.8 | 0.3 | 0.90 | 8.5 | 0.3 | 0.2 | 0.89 |
| | 5.0 | 6.0 | 6.6 | 6.9 | 3.7 | 0.85 | 8.4 | 7.7 | 1.4 | 0.89 | 8.4 | 1.4 | 0.4 | 0.89 | 8.4 | 0.8 | 0.2 | 0.89 | 8.5 | 0.3 | 0.2 | 0.89 |
| mean_matching | 1.0 | 8.7 | 1.2 | 1.8 | 8.5 | 0.89 | 8.4 | 7.7 | 1.4 | 0.89 | 8.5 | 1.5 | 0.4 | 0.89 | 8.5 | 0.8 | 0.3 | 0.89 | 8.5 | 0.3 | 0.1 | 0.89 |
| | 2.0 | 7.1 | 6.4 | 1.6 | 8.5 | 0.89 | 8.4 | 7.7 | 1.5 | 0.89 | 8.5 | 1.4 | 0.4 | 0.89 | 8.4 | 0.8 | 0.3 | 0.89 | 8.5 | 0.4 | 0.2 | 0.89 |
| | 3.0 | 4.8 | 7.8 | 1.6 | 8.6 | 0.89 | 8.4 | 7.6 | 1.8 | 0.89 | 8.4 | 1.5 | 0.5 | 0.89 | 8.4 | 0.7 | 0.3 | 0.89 | 8.5 | 0.3 | 0.2 | 0.89 |
| | 4.0 | 3.3 | 8.2 | 1.6 | 8.5 | 0.89 | 8.3 | 7.5 | 2.3 | 0.89 | 8.4 | 1.6 | 0.8 | 0.89 | 8.4 | 0.9 | 0.5 | 0.89 | 8.5 | 0.4 | 0.2 | 0.89 |
| | 5.0 | 2.1 | 8.3 | 1.6 | 8.6 | 0.89 | 8.1 | 7.0 | 2.9 | 0.89 | 8.3 | 1.8 | 1.2 | 0.89 | 8.3 | 0.9 | 0.7 | 0.89 | 8.5 | 0.4 | 0.3 | 0.89 |

Table 7: Model Qwen2.5-14b, flipping from horses to motorcycles

| method | strength | horses | | motorcycles | | | cows | | | | pigs | | | | dogs | | | | legislators | | | |
|---|---|---|---|---|---|---|---|---|---|---|---|---|---|---|---|---|---|---|---|---|---|---|
| | | src-cs | tgt-cs | src-cs | tgt-cs | bertp | cs | src-cs | tgt-cs | bertp | cs | src-cs | tgt-cs | bertp | cs | src-cs | tgt-cs | bertp | cs | src-cs | tgt-cs | bertp |
| CASteer | 1.0 | 8.7 | 0.7 | 1.2 | 8.6 | 0.90 | 8.4 | 1.7 | 0.2 | 0.89 | 8.4 | 1.1 | 0.1 | 0.89 | 8.7 | 1.4 | 0.2 | 0.90 | 8.5 | 0.3 | 0.3 | 0.89 |
| | 2.0 | 6.4 | 4.4 | 4.9 | 6.0 | 0.87 | 8.3 | 1.5 | 0.3 | 0.89 | 8.4 | 1.0 | 0.1 | 0.89 | 8.6 | 1.4 | 0.2 | 0.89 | 8.4 | 0.3 | 0.3 | 0.89 |
| | 3.0 | 5.9 | 5.0 | 6.9 | 2.9 | 0.84 | 7.9 | 1.4 | 0.5 | 0.88 | 8.4 | 0.9 | 0.2 | 0.88 | 8.6 | 1.3 | 0.2 | 0.89 | 8.5 | 0.3 | 0.3 | 0.89 |
| | 4.0 | 6.0 | 4.4 | 7.3 | 1.8 | 0.84 | 7.6 | 1.4 | 0.6 | 0.87 | 8.1 | 0.9 | 0.3 | 0.87 | 8.5 | 1.4 | 0.3 | 0.88 | 8.5 | 0.3 | 0.3 | 0.89 |
| | 5.0 | 5.5 | 4.3 | 7.4 | 1.6 | 0.83 | 7.5 | 1.3 | 0.5 | 0.86 | 7.9 | 0.9 | 0.3 | 0.86 | 8.3 | 1.4 | 0.3 | 0.88 | 8.5 | 0.3 | 0.3 | 0.89 |
| LEACE | 1.0 | 8.6 | 0.7 | 1.1 | 8.6 | 0.90 | 8.4 | 1.8 | 0.2 | 0.90 | 8.5 | 1.1 | 0.1 | 0.89 | 8.7 | 1.4 | 0.2 | 0.90 | 8.5 | 0.4 | 0.3 | 0.89 |
| | 2.0 | 6.4 | 4.6 | 3.8 | 7.5 | 0.88 | 8.4 | 1.6 | 0.2 | 0.89 | 8.4 | 1.0 | 0.1 | 0.89 | 8.7 | 1.3 | 0.2 | 0.90 | 8.5 | 0.4 | 0.4 | 0.89 |
| | 3.0 | 4.2 | 6.0 | 6.1 | 4.6 | 0.85 | 8.2 | 1.5 | 0.4 | 0.89 | 8.4 | 1.0 | 0.1 | 0.89 | 8.7 | 1.4 | 0.2 | 0.89 | 8.5 | 0.3 | 0.3 | 0.89 |
| | 4.0 | 4.2 | 6.1 | 6.6 | 3.5 | 0.84 | 7.9 | 1.5 | 0.6 | 0.88 | 8.3 | 1.0 | 0.2 | 0.88 | 8.7 | 1.4 | 0.3 | 0.89 | 8.5 | 0.4 | 0.4 | 0.89 |
| | 5.0 | 4.1 | 5.9 | 6.8 | 3.1 | 0.83 | 7.7 | 1.4 | 0.7 | 0.88 | 8.3 | 1.0 | 0.2 | 0.88 | 8.6 | 1.3 | 0.4 | 0.89 | 8.5 | 0.4 | 0.3 | 0.89 |
| mean_matching | 1.0 | 8.7 | 0.7 | 0.9 | 8.6 | 0.90 | 8.4 | 1.8 | 0.1 | 0.89 | 8.5 | 1.1 | 0.1 | 0.89 | 8.7 | 1.4 | 0.2 | 0.90 | 8.5 | 0.4 | 0.3 | 0.89 |
| | 2.0 | 3.9 | 6.5 | 0.9 | 8.6 | 0.90 | 8.3 | 1.6 | 0.2 | 0.89 | 8.4 | 1.1 | 0.1 | 0.89 | 8.7 | 1.3 | 0.2 | 0.90 | 8.5 | 0.3 | 0.3 | 0.89 |
| | 3.0 | 1.4 | 8.2 | 0.9 | 8.6 | 0.90 | 8.3 | 1.5 | 0.2 | 0.89 | 8.4 | 1.0 | 0.1 | 0.89 | 8.7 | 1.4 | 0.3 | 0.89 | 8.5 | 0.3 | 0.3 | 0.89 |
| | 4.0 | 1.2 | 8.2 | 0.9 | 8.6 | 0.90 | 8.3 | 1.5 | 0.2 | 0.89 | 8.4 | 1.0 | 0.2 | 0.89 | 8.7 | 1.3 | 0.3 | 0.89 | 8.4 | 0.4 | 0.4 | 0.89 |
| | 5.0 | 1.2 | 8.2 | 0.8 | 8.6 | 0.90 | 8.1 | 1.6 | 0.7 | 0.89 | 8.3 | 0.9 | 0.4 | 0.89 | 8.6 | 1.3 | 0.3 | 0.89 | 8.5 | 0.3 | 0.3 | 0.89 |

### 7.9.4  DETAILED RESULTS FOR FOR DIFFUSION MODELS CONCEPT SWITCHING

In this section, in tab. 11,12,8,9 we provide detailed breakdown of scores for all $\beta$ values and all concepts $c_s, c_t, c_i$. Pareto plots in sec. 7.9.1 were created based on the scores provided in these tables.

Table 8: Model SDXL, flipping from horse to motorcycle

| method | strength | horse src-cs | tgt-cs | motorcycle src-cs | tgt-cs | fid | cow cs | src-cs | tgt-cs | fid | pig cs | src-cs | tgt-cs | fid | dog cs | src-cs | tgt-cs | fid | legislator cs | src-cs | tgt-cs | fid |
|---|---|---|---|---|---|---|---|---|---|---|---|---|---|---|---|---|---|---|---|---|---|---|
| No Steering | - | 71.0 | 49.1 | 51.8 | 70.7 | - | 72.7 | 54.6 | 41.5 | - | 71.8 | 49.5 | 43.6 | - | 66.3 | 52.4 | 44.9 | - | 60.8 | 44.8 | 42.4 | - |
| CASteer | 1.0 | 70.0 | 50.8 | 52.6 | 71.3 | 27.8 | 72.3 | 54.4 | 42.4 | 21.9 | 71.8 | 49.0 | 43.8 | 13.3 | 66.2 | 52.0 | 45.0 | 20.0 | 60.9 | 44.8 | 42.5 | 16.3 |
| | 1.5 | 53.4 | 68.3 | 60.1 | 63.7 | 121.3 | 72.0 | 54.4 | 43.0 | 28.7 | 71.9 | 48.9 | 44.1 | 16.2 | 66.1 | 51.9 | 45.1 | 24.7 | 60.9 | 44.8 | 42.5 | 21.0 |
| | 2.0 | 52.1 | 69.5 | 68.3 | 52.9 | 212.4 | 70.9 | 54.4 | 44.9 | 42.7 | 71.9 | 48.9 | 44.4 | 18.9 | 66.1 | 51.8 | 45.4 | 28.6 | 60.9 | 44.8 | 42.6 | 24.6 |
| | 2.5 | 51.7 | 69.4 | 69.4 | 51.7 | 213.0 | 62.5 | 54.2 | 55.8 | 105.2 | 72.0 | 48.7 | 44.6 | 22.0 | 66.1 | 51.7 | 45.6 | 33.0 | 60.9 | 44.9 | 42.7 | 27.0 |
| | 3.0 | 51.4 | 69.1 | 69.9 | 50.9 | 210.0 | 52.4 | 53.2 | 66.2 | 186.9 | 72.0 | 48.5 | 44.9 | 25.8 | 66.0 | 51.6 | 45.7 | 37.2 | 60.9 | 44.9 | 42.7 | 29.7 |
| | 4.0 | 51.0 | 68.7 | 70.6 | 49.9 | 207.9 | 48.6 | 52.3 | 69.5 | 222.6 | 72.0 | 48.5 | 46.5 | 37.2 | 65.7 | 51.4 | 46.7 | 46.5 | 60.9 | 44.9 | 42.8 | 35.0 |
| | 5.0 | 50.7 | 68.5 | 70.9 | 49.5 | 207.4 | 47.8 | 51.8 | 69.6 | 231.6 | 70.5 | 49.1 | 50.6 | 63.7 | 64.3 | 51.4 | 49.0 | 61.6 | 61.0 | 45.1 | 43.0 | 39.0 |
| LEACE | 1.0 | 65.0 | 56.5 | 52.7 | 71.2 | 28.6 | 72.5 | 54.4 | 42.0 | 17.0 | 71.7 | 49.3 | 43.6 | 8.9 | 66.2 | 52.3 | 44.8 | 14.1 | 60.7 | 44.8 | 42.5 | 21.1 |
| | 1.5 | 52.1 | 68.6 | 57.1 | 67.0 | 84.6 | 72.2 | 54.4 | 42.3 | 21.3 | 71.7 | 49.3 | 43.7 | 10.9 | 66.2 | 52.3 | 44.8 | 17.7 | 60.7 | 44.8 | 42.5 | 24.9 |
| | 2.0 | 51.2 | 68.8 | 67.6 | 53.3 | 207.6 | 72.2 | 54.5 | 42.7 | 25.2 | 71.7 | 49.3 | 43.7 | 12.6 | 66.1 | 52.3 | 44.8 | 20.8 | 60.6 | 44.8 | 42.4 | 28.2 |
| | 2.5 | 50.8 | 68.5 | 69.0 | 51.5 | 213.3 | 71.9 | 54.5 | 43.1 | 30.0 | 71.7 | 49.4 | 43.7 | 14.0 | 66.1 | 52.2 | 44.8 | 22.6 | 60.6 | 44.9 | 42.5 | 31.0 |
| | 3.0 | 50.5 | 68.2 | 69.6 | 50.5 | 210.6 | 71.4 | 54.4 | 43.8 | 37.0 | 71.7 | 49.4 | 43.8 | 15.1 | 66.0 | 52.2 | 44.8 | 24.5 | 60.6 | 44.8 | 42.5 | 33.2 |
| | 4.0 | 50.2 | 68.0 | 70.4 | 49.6 | 206.7 | 64.6 | 54.2 | 52.6 | 85.7 | 71.8 | 49.4 | 43.9 | 17.2 | 66.0 | 52.2 | 44.7 | 28.1 | 60.6 | 44.8 | 42.6 | 37.3 |
| | 5.0 | 49.9 | 67.7 | 70.8 | 49.1 | 204.4 | 54.4 | 53.2 | 63.6 | 166.2 | 71.8 | 49.4 | 44.0 | 19.4 | 65.9 | 52.2 | 44.7 | 31.2 | 60.4 | 44.8 | 42.7 | 42.0 |
| mean_matching | 1.0 | 51.2 | 68.7 | 51.9 | 70.7 | 12.7 | 72.2 | 54.5 | 42.5 | 23.9 | 71.8 | 49.2 | 43.9 | 12.4 | 66.1 | 52.3 | 44.8 | 20.7 | 60.7 | 45.0 | 42.3 | 27.2 |
| | 1.5 | 50.4 | 68.1 | 51.9 | 70.7 | 15.1 | 71.6 | 54.6 | 43.5 | 34.5 | 71.9 | 49.1 | 44.1 | 15.2 | 66.1 | 52.2 | 44.8 | 25.3 | 60.5 | 45.1 | 42.3 | 32.3 |
| | 2.0 | 50.0 | 67.8 | 52.0 | 70.8 | 17.2 | 65.9 | 54.3 | 50.9 | 77.3 | 71.9 | 49.0 | 44.3 | 17.6 | 66.1 | 52.1 | 44.8 | 28.8 | 60.6 | 45.2 | 42.3 | 35.9 |
| | 2.5 | 49.5 | 67.4 | 52.0 | 70.7 | 18.6 | 55.1 | 53.3 | 62.6 | 162.3 | 71.9 | 49.0 | 44.5 | 20.2 | 66.0 | 52.1 | 44.8 | 32.2 | 60.5 | 45.5 | 42.4 | 39.8 |
| | 3.0 | 49.2 | 67.2 | 52.0 | 70.6 | 20.0 | 51.0 | 52.7 | 66.7 | 199.3 | 72.0 | 49.0 | 44.8 | 23.5 | 66.0 | 52.0 | 44.8 | 35.3 | 60.2 | 45.5 | 42.4 | 43.1 |
| | 4.0 | 48.8 | 67.2 | 52.0 | 70.5 | 22.9 | 48.4 | 51.9 | 68.7 | 222.5 | 72.1 | 49.0 | 45.5 | 29.5 | 65.8 | 51.9 | 44.9 | 41.4 | 59.7 | 46.0 | 42.7 | 49.9 |
| | 5.0 | 48.3 | 66.9 | 52.0 | 70.4 | 28.0 | 47.9 | 51.5 | 68.7 | 229.8 | 71.9 | 49.2 | 46.8 | 40.0 | 65.5 | 51.9 | 45.2 | 48.7 | 59.1 | 46.7 | 43.2 | 58.2 |

Table 9: Model SDXL, flipping from chihuahua to muffin

| method | strength | chihuahua src-cs | tgt-cs | muffin src-cs | tgt-cs | fid | dog cs | src-cs | tgt-cs | fid | wolf cs | src-cs | tgt-cs | fid | cat cs | src-cs | tgt-cs | fid | legislator cs | src-cs | tgt-cs | fid |
|---|---|---|---|---|---|---|---|---|---|---|---|---|---|---|---|---|---|---|---|---|---|---|
| No Steering | - | 75.9 | 54.6 | 42.6 | 68.2 | - | 66.3 | 57.9 | 52.5 | - | 71.8 | 52.7 | 45.6 | - | 67.5 | 53.4 | 54.2 | - | 60.8 | 42.6 | 40.1 | - |
| CASteer | 1.0 | 71.7 | 54.7 | 59.3 | 61.5 | 145.9 | 65.7 | 54.4 | 52.4 | 37.4 | 71.9 | 51.7 | 44.7 | 19.1 | 67.2 | 52.8 | 53.9 | 26.2 | 60.8 | 42.4 | 39.9 | 19.2 |
| | 1.5 | 47.0 | 61.0 | 66.5 | 57.3 | 211.5 | 65.2 | 53.1 | 52.4 | 53.3 | 71.8 | 51.2 | 44.3 | 24.4 | 67.1 | 52.4 | 53.8 | 33.8 | 60.8 | 42.3 | 39.6 | 23.2 |
| | 2.0 | 43.7 | 63.2 | 69.5 | 56.7 | 226.4 | 63.2 | 51.7 | 53.0 | 79.7 | 71.8 | 51.0 | 43.9 | 30.4 | 66.8 | 52.0 | 53.6 | 41.7 | 60.8 | 42.3 | 39.6 | 26.7 |
| | 2.5 | 42.3 | 63.8 | 71.5 | 56.3 | 241.3 | 55.2 | 48.9 | 55.5 | 140.5 | 71.5 | 50.6 | 43.6 | 36.8 | 66.3 | 51.7 | 53.6 | 53.8 | 60.9 | 42.3 | 39.6 | 29.8 |
| | 3.0 | 41.4 | 64.0 | 72.9 | 56.3 | 253.8 | 48.1 | 45.8 | 58.2 | 192.4 | 70.8 | 50.3 | 43.6 | 45.7 | 63.6 | 50.7 | 54.3 | 82.4 | 60.9 | 42.4 | 39.5 | 32.7 |
| | 4.0 | 40.1 | 64.3 | 74.8 | 56.2 | 276.0 | 44.2 | 43.5 | 60.3 | 226.9 | 65.6 | 49.0 | 45.6 | 96.6 | 52.5 | 47.2 | 57.0 | 168.7 | 60.9 | 42.4 | 39.4 | 38.0 |
| | 5.0 | 39.3 | 64.1 | 75.6 | 56.1 | 291.6 | 42.9 | 42.5 | 60.8 | 243.3 | 53.5 | 47.0 | 51.5 | 206.1 | 46.9 | 45.2 | 58.0 | 213.0 | 60.8 | 42.5 | 39.3 | 43.9 |
| LEACE | 1.0 | 67.1 | 55.7 | 60.4 | 60.6 | 160.6 | 66.0 | 55.8 | 52.4 | 25.4 | 72.0 | 52.1 | 45.3 | 11.9 | 67.5 | 53.2 | 54.1 | 17.5 | 60.8 | 42.5 | 40.2 | 20.0 |
| | 1.5 | 46.2 | 61.5 | 66.4 | 57.3 | 212.1 | 65.9 | 54.9 | 52.3 | 33.1 | 72.1 | 52.0 | 45.1 | 14.7 | 67.5 | 53.1 | 54.0 | 21.1 | 60.9 | 42.4 | 40.3 | 24.6 |
| | 2.0 | 42.8 | 63.3 | 69.3 | 56.9 | 227.7 | 65.8 | 54.2 | 52.3 | 40.3 | 72.1 | 51.8 | 44.9 | 16.9 | 67.6 | 53.0 | 53.9 | 24.6 | 60.9 | 42.3 | 40.3 | 28.1 |
| | 2.5 | 41.4 | 63.6 | 71.0 | 56.7 | 240.7 | 65.5 | 53.4 | 52.2 | 48.6 | 72.1 | 51.5 | 44.7 | 19.0 | 67.6 | 52.9 | 53.9 | 27.8 | 61.0 | 42.3 | 40.4 | 31.3 |
| | 3.0 | 40.6 | 63.9 | 72.3 | 56.7 | 250.8 | 65.1 | 52.5 | 52.3 | 57.1 | 72.2 | 51.4 | 44.6 | 21.5 | 67.6 | 52.7 | 53.8 | 30.4 | 61.0 | 42.5 | 40.5 | 34.0 |
| | 4.0 | 39.3 | 63.8 | 73.7 | 56.6 | 269.4 | 63.8 | 51.3 | 52.6 | 76.7 | 72.2 | 51.0 | 44.2 | 25.7 | 67.5 | 52.6 | 53.8 | 35.4 | 61.0 | 42.7 | 40.6 | 39.4 |
| | 5.0 | 38.5 | 63.7 | 74.6 | 56.5 | 282.5 | 60.3 | 49.7 | 53.7 | 104.1 | 72.2 | 50.7 | 44.0 | 29.8 | 67.4 | 52.4 | 53.7 | 41.1 | 61.0 | 43.0 | 41.1 | 43.4 |
| mean_matching | 1.0 | 44.3 | 62.5 | 42.0 | 68.3 | 23.9 | 66.9 | 54.7 | 52.4 | 35.1 | 72.2 | 51.7 | 44.9 | 16.9 | 67.6 | 53.0 | 54.1 | 22.7 | 60.8 | 42.5 | 40.1 | 23.9 |
| | 1.5 | 41.5 | 63.9 | 41.9 | 68.2 | 28.3 | 65.5 | 53.5 | 52.4 | 48.8 | 72.2 | 51.2 | 44.6 | 21.2 | 67.7 | 52.8 | 54.1 | 27.7 | 60.8 | 42.4 | 40.2 | 29.0 |
| | 2.0 | 39.8 | 64.0 | 41.7 | 68.2 | 32.2 | 64.5 | 52.2 | 52.7 | 65.6 | 72.2 | 50.8 | 44.1 | 25.4 | 67.7 | 52.6 | 54.1 | 31.6 | 60.9 | 42.5 | 40.4 | 32.5 |
| | 2.5 | 38.9 | 64.3 | 41.6 | 68.1 | 35.7 | 63.2 | 51.3 | 53.1 | 83.0 | 72.2 | 50.6 | 43.9 | 29.0 | 67.6 | 52.4 | 54.0 | 35.7 | 61.0 | 42.4 | 40.5 | 36.3 |
| | 3.0 | 38.0 | 64.2 | 41.5 | 68.0 | 39.5 | 59.4 | 49.7 | 54.4 | 109.8 | 72.2 | 50.3 | 43.8 | 33.4 | 67.5 | 52.2 | 54.0 | 40.1 | 60.9 | 42.4 | 40.7 | 39.4 |
| | 4.0 | 37.5 | 64.0 | 41.3 | 67.9 | 48.2 | 48.3 | 44.8 | 58.7 | 178.3 | 71.6 | 50.3 | 44.2 | 43.9 | 67.0 | 51.8 | 54.3 | 50.7 | 60.6 | 42.4 | 41.1 | 45.1 |
| | 5.0 | 37.9 | 63.6 | 41.3 | 67.5 | 58.9 | 43.4 | 42.2 | 61.1 | 209.8 | 70.2 | 50.4 | 44.8 | 63.1 | 64.5 | 50.7 | 54.5 | 69.6 | 60.4 | 42.5 | 41.5 | 51.4 |

Table 10: Model SDXL, flipping from snoopy to mickey

| method | strength | snoopy src-cs | tgt-cs | mickey src-cs | tgt-cs | fid | pikachu cs | src-cs | tgt-cs | fid | spongebob cs | src-cs | tgt-cs | fid | dog cs | src-cs | tgt-cs | fid | legislator cs | src-cs | tgt-cs | fid |
|---|---|---|---|---|---|---|---|---|---|---|---|---|---|---|---|---|---|---|---|---|---|---|
| No Steering | - | 74.3 | 58.7 | 56.0 | 73.1 | - | 72.6 | 41.3 | 51.2 | - | 75.1 | 49.0 | 52.5 | - | 66.3 | 56.1 | 52.0 | - | 60.8 | 41.6 | 45.0 | - |
| CASteer | 1.0 | 65.6 | 68.4 | 58.9 | 70.8 | 47.5 | 72.7 | 41.4 | 51.2 | 13.8 | 75.1 | 48.9 | 52.5 | 25.0 | 66.4 | 55.0 | 52.1 | 25.4 | 60.8 | 41.6 | 45.0 | 14.9 |
| | 1.5 | 59.8 | 71.2 | 61.7 | 68.6 | 59.6 | 72.7 | 41.4 | 51.2 | 17.1 | 75.1 | 48.9 | 52.6 | 29.7 | 66.3 | 54.4 | 52.1 | 33.9 | 60.7 | 41.6 | 45.0 | 18.4 |
| | 2.0 | 57.0 | 72.4 | 66.8 | 65.3 | 72.3 | 72.7 | 41.5 | 51.1 | 19.4 | 75.1 | 48.9 | 52.6 | 33.5 | 66.4 | 53.9 | 52.2 | 40.6 | 60.8 | 41.5 | 45.0 | 21.1 |
| | 2.5 | 55.4 | 72.7 | 69.6 | 61.3 | 83.9 | 72.7 | 41.6 | 51.2 | 21.2 | 75.0 | 48.8 | 52.5 | 36.5 | 66.4 | 53.3 | 52.2 | 48.5 | 60.9 | 41.6 | 45.0 | 23.3 |
| | 3.0 | 54.5 | 72.7 | 70.3 | 59.4 | 91.8 | 72.7 | 41.6 | 51.2 | 22.9 | 75.1 | 48.9 | 52.6 | 39.1 | 66.4 | 52.8 | 52.3 | 54.8 | 60.8 | 41.6 | 45.0 | 25.6 |
| | 4.0 | 53.0 | 71.9 | 70.8 | 57.4 | 106.3 | 72.7 | 41.7 | 51.2 | 25.9 | 75.2 | 48.7 | 52.6 | 43.3 | 66.4 | 52.1 | 52.6 | 66.5 | 60.7 | 41.6 | 45.0 | 28.9 |
| | 5.0 | 52.0 | 71.3 | 71.2 | 56.4 | 117.5 | 72.7 | 41.7 | 51.2 | 28.9 | 75.0 | 48.7 | 52.7 | 47.7 | 66.3 | 51.3 | 52.8 | 79.9 | 60.7 | 41.6 | 45.0 | 31.5 |
| LEACE | 1.0 | 65.9 | 67.9 | 58.8 | 70.6 | 50.0 | 72.8 | 41.4 | 51.1 | 18.5 | 75.0 | 48.9 | 52.5 | 33.5 | 66.3 | 55.8 | 52.0 | 18.6 | 60.9 | 41.7 | 45.1 | 22.2 |
| | 1.5 | 60.1 | 71.0 | 61.0 | 68.1 | 62.6 | 72.8 | 41.5 | 51.1 | 22.0 | 75.0 | 48.8 | 52.5 | 39.0 | 66.3 | 55.8 | 52.1 | 23.9 | 60.9 | 41.7 | 45.0 | 26.4 |
| | 2.0 | 57.5 | 72.2 | 64.3 | 64.6 | 77.0 | 72.9 | 41.5 | 51.1 | 24.5 | 75.1 | 48.9 | 52.5 | 43.6 | 66.3 | 55.5 | 52.1 | 28.4 | 60.9 | 41.9 | 45.2 | 29.7 |
| | 2.5 | 56.0 | 72.7 | 67.3 | 60.7 | 89.9 | 72.9 | 41.6 | 51.1 | 26.7 | 75.2 | 48.9 | 52.5 | 47.7 | 66.3 | 55.5 | 52.2 | 32.0 | 61.0 | 41.9 | 45.1 | 31.9 |
| | 3.0 | 55.1 | 72.7 | 68.2 | 58.2 | 101.5 | 73.0 | 41.7 | 51.2 | 29.1 | 75.2 | 48.8 | 52.5 | 51.3 | 66.4 | 55.4 | 52.3 | 35.7 | 61.0 | 41.9 | 45.1 | 34.8 |
| | 4.0 | 53.7 | 72.3 | 68.7 | 55.9 | 118.6 | 73.2 | 41.9 | 51.2 | 33.4 | 75.0 | 49.0 | 52.6 | 57.7 | 66.5 | 55.3 | 52.5 | 43.4 | 60.9 | 42.1 | 45.2 | 38.9 |
| | 5.0 | 53.0 | 72.0 | 69.5 | 55.2 | 134.6 | 73.4 | 42.4 | 51.4 | 37.6 | 74.8 | 49.1 | 52.7 | 64.0 | 66.5 | 55.2 | 52.7 | 50.0 | 60.9 | 42.3 | 45.3 | 42.2 |
| mean_matching | 1.0 | 59.7 | 71.0 | 55.5 | 72.8 | 30.1 | 72.8 | 41.3 | 51.3 | 19.0 | 74.7 | 48.6 | 52.6 | 19.3 | 66.2 | 55.9 | 52.1 | 20.4 | 60.9 | 41.7 | 45.0 | 24.5 |
| | 1.5 | 56.0 | 72.5 | 55.3 | 72.7 | 34.8 | 72.9 | 41.2 | 51.2 | 22.3 | 74.6 | 48.6 | 52.6 | 42.7 | 66.2 | 55.8 | 52.1 | 25.9 | 60.8 | 41.8 | 45.1 | 29.1 |
| | 2.0 | 54.5 | 72.7 | 55.0 | 72.4 | 38.9 | 72.9 | 41.3 | 51.3 | 25.3 | 74.5 | 48.6 | 52.6 | 46.8 | 66.3 | 55.6 | 52.1 | 30.6 | 60.7 | 42.0 | 45.1 | 32.6 |
| | 2.5 | 53.4 | 72.4 | 54.8 | 72.4 | 42.2 | 73.0 | 41.2 | 51.3 | 27.3 | 74.3 | 48.4 | 52.6 | 50.8 | 66.3 | 55.6 | 52.3 | 34.4 | 60.7 | 42.0 | 45.1 | 35.2 |
| | 3.0 | 52.5 | 71.8 | 54.7 | 72.2 | 44.7 | 73.1 | 41.3 | 51.3 | 30.0 | 74.2 | 48.5 | 52.7 | 55.2 | 66.4 | 55.5 | 52.5 | 38.3 | 60.7 | 42.1 | 45.3 | 37.6 |
| | 4.0 | 51.5 | 71.2 | 54.6 | 71.9 | 49.6 | 73.2 | 41.4 | 51.4 | 34.5 | 73.4 | 48.4 | 53.0 | 64.7 | 66.4 | 55.3 | 52.7 | 47.2 | 60.4 | 42.3 | 45.4 | 43.3 |
| | 5.0 | 51.0 | 70.4 | 54.4 | 71.5 | 55.1 | 73.3 | 41.3 | 51.4 | 39.1 | 72.7 | 48.6 | 53.4 | 72.9 | 66.5 | 55.1 | 52.9 | 55.3 | 60.3 | 42.5 | 45.6 | 48.8 |

Table 11: Model SANA, flipping from horse to motorcycle

| method | strength | horse | | motorcycle | | | cow | | | | pig | | | | dog | | | | legislator | | | |
|---|---|---|---|---|---|---|---|---|---|---|---|---|---|---|---|---|---|---|---|---|---|---|
| | | src-cs | tgt-cs | src-cs | tgt-cs | fid | cs | src-cs | tgt-cs | fid | cs | src-cs | tgt-cs | fid | cs | src-cs | tgt-cs | fid | cs | src-cs | tgt-cs | fid |
| No Steering | - | 72.1 | 50.6 | 50.9 | 70.5 | - | 73.8 | 55.3 | 42.5 | - | 73.5 | 48.3 | 44.5 | - | 68.1 | 51.7 | 46.1 | - | 60.4 | 45.8 | 43.9 | - |
| CASteer | 1.0 | 71.0 | 52.0 | 52.2 | 71.3 | 48.1 | 73.6 | 54.8 | 43.0 | 22.1 | 73.8 | 48.0 | 44.8 | 13.8 | 68.1 | 51.6 | 46.2 | 15.7 | 60.2 | 46.0 | 44.3 | 15.9 |
| | 2.0 | 65.5 | 60.4 | 67.5 | 56.0 | 229.0 | 73.1 | 54.2 | 43.7 | 37.9 | 74.0 | 47.7 | 45.0 | 22.6 | 68.1 | 51.4 | 46.3 | 24.1 | 60.0 | 46.0 | 44.5 | 21.5 |
| | 3.0 | 54.9 | 68.7 | 69.5 | 51.8 | 234.6 | 72.3 | 54.0 | 45.0 | 55.8 | 74.2 | 47.4 | 45.3 | 30.5 | 68.1 | 51.3 | 46.5 | 32.2 | 59.7 | 46.3 | 44.9 | 25.9 |
| | 4.0 | 52.6 | 70.3 | 70.2 | 50.7 | 228.7 | 69.3 | 55.1 | 53.0 | 107.1 | 74.5 | 47.4 | 45.9 | 39.9 | 68.2 | 51.2 | 46.7 | 40.4 | 59.5 | 46.5 | 45.3 | 31.1 |
| | 5.0 | 51.8 | 71.0 | 70.5 | 50.3 | 224.9 | 62.3 | 54.8 | 61.7 | 167.0 | 74.7 | 47.5 | 46.9 | 48.9 | 68.2 | 51.2 | 47.0 | 51.3 | 59.3 | 46.7 | 45.8 | 36.8 |
| LEACE | 1.0 | 68.7 | 56.2 | 51.6 | 71.6 | 36.3 | 73.7 | 55.2 | 42.6 | 10.2 | 73.6 | 48.2 | 44.7 | 8.0 | 68.1 | 51.8 | 46.0 | 8.1 | 60.4 | 45.7 | 43.9 | 11.6 |
| | 2.0 | 53.1 | 70.7 | 58.7 | 69.4 | 125.0 | 73.7 | 55.2 | 42.8 | 16.2 | 73.7 | 48.1 | 44.7 | 12.6 | 68.1 | 51.8 | 46.0 | 13.1 | 60.5 | 45.7 | 43.8 | 17.0 |
| | 3.0 | 51.9 | 71.3 | 65.7 | 58.2 | 229.3 | 73.7 | 55.1 | 43.1 | 21.0 | 69.8 | 57.6 | 52.1 | 220.6 | 68.0 | 51.8 | 46.0 | 17.7 | 60.7 | 45.8 | 43.8 | 21.6 |
| | 4.0 | 51.9 | 71.1 | 67.4 | 53.7 | 242.8 | 73.7 | 55.0 | 43.3 | 25.9 | 73.9 | 48.0 | 44.9 | 20.4 | 68.0 | 51.8 | 46.0 | 21.5 | 55.1 | 52.2 | 51.9 | 294.6 |
| | 5.0 | 51.7 | 70.8 | 68.4 | 52.6 | 236.2 | 73.7 | 54.9 | 43.6 | 31.8 | 74.0 | 48.0 | 45.0 | 24.0 | 68.1 | 51.8 | 45.9 | 25.0 | 60.8 | 45.8 | 43.8 | 30.1 |
| mean_matching | 1.0 | 55.5 | 68.9 | 50.9 | 70.4 | 8.7 | 73.6 | 55.1 | 42.7 | 14.7 | 73.7 | 48.2 | 44.7 | 10.4 | 68.0 | 51.8 | 46.1 | 9.3 | 60.5 | 45.8 | 44.0 | 13.0 |
| | 2.0 | 51.7 | 71.1 | 50.9 | 70.4 | 11.7 | 73.6 | 55.0 | 43.1 | 23.2 | 73.9 | 48.1 | 44.8 | 16.5 | 68.0 | 51.7 | 46.1 | 15.0 | 60.6 | 45.8 | 43.9 | 19.1 |
| | 3.0 | 51.4 | 70.7 | 50.9 | 70.4 | 13.9 | 73.4 | 54.7 | 43.4 | 32.5 | 74.1 | 48.0 | 45.0 | 22.2 | 67.9 | 51.7 | 46.1 | 19.8 | 60.8 | 45.9 | 44.0 | 23.9 |
| | 4.0 | 51.4 | 70.9 | 50.9 | 70.4 | 15.9 | 73.1 | 54.5 | 43.9 | 43.8 | 74.3 | 48.1 | 45.3 | 27.4 | 67.9 | 51.6 | 46.1 | 24.3 | 60.8 | 45.9 | 44.1 | 28.9 |
| | 5.0 | 51.5 | 71.4 | 51.0 | 70.4 | 18.0 | 69.8 | 55.2 | 50.0 | 78.2 | 74.5 | 48.1 | 45.7 | 34.3 | 67.9 | 51.6 | 46.2 | 28.4 | 61.0 | 46.0 | 44.2 | 34.3 |

Table 12: Model SANA, flipping from chihuahua to muffin

| method | strength | chihuahua | | muffin | | | dog | | | | wolf | | | | cat | | | | legislator | | | |
|---|---|---|---|---|---|---|---|---|---|---|---|---|---|---|---|---|---|---|---|---|---|---|
| | | src-cs | tgt-cs | src-cs | tgt-cs | fid | cs | src-cs | tgt-cs | fid | cs | src-cs | tgt-cs | fid | cs | src-cs | tgt-cs | fid | cs | src-cs | tgt-cs | fid |
| No Steering | - | 76.4 | 55.0 | 43.4 | 66.3 | - | 68.1 | 62.0 | 52.7 | - | 73.2 | 52.8 | 46.1 | - | 68.5 | 53.4 | 53.0 | - | 60.4 | 42.7 | 40.8 | - |
| CASteer | 1.0 | 75.8 | 55.8 | 45.3 | 65.6 | 46.6 | 67.7 | 58.4 | 52.9 | 39.4 | 73.1 | 52.7 | 45.8 | 16.3 | 68.3 | 52.8 | 53.5 | 24.5 | 60.2 | 42.8 | 41.0 | 13.8 |
| | 2.0 | 55.3 | 60.1 | 62.0 | 60.4 | 200.7 | 67.1 | 56.3 | 54.1 | 78.8 | 72.9 | 52.4 | 45.6 | 28.4 | 68.3 | 52.1 | 54.3 | 39.2 | 60.1 | 42.7 | 41.1 | 18.9 |
| | 3.0 | 45.4 | 60.5 | 60.0 | 47.6 | 278.5 | 50.6 | 49.0 | 59.2 | 198.9 | 72.4 | 52.4 | 45.8 | 46.2 | 67.9 | 51.0 | 55.0 | 61.7 | 59.9 | 42.8 | 41.3 | 23.4 |
| | 4.0 | 44.7 | 60.7 | 70.6 | 54.9 | 250.7 | 45.1 | 46.0 | 60.8 | 223.3 | 68.6 | 52.0 | 46.6 | 113.8 | 65.5 | 50.1 | 55.5 | 109.2 | 59.9 | 42.7 | 41.4 | 27.6 |
| | 5.0 | 44.1 | 60.7 | 72.1 | 54.2 | 260.7 | 43.4 | 45.3 | 61.1 | 229.3 | 58.9 | 51.1 | 50.3 | 216.2 | 56.9 | 48.0 | 57.6 | 179.1 | 59.9 | 42.7 | 41.6 | 31.3 |
| LEACE | 1.0 | 72.3 | 58.3 | 46.1 | 65.0 | 57.7 | 67.9 | 59.9 | 52.7 | 21.0 | 73.2 | 52.9 | 46.1 | 5.8 | 68.4 | 53.2 | 52.9 | 8.9 | 60.4 | 42.7 | 40.8 | 9.1 |
| | 2.0 | 48.9 | 61.4 | 59.8 | 63.0 | 185.8 | 67.8 | 58.3 | 52.8 | 35.1 | 73.2 | 52.9 | 46.1 | 9.4 | 68.4 | 53.1 | 53.0 | 13.6 | 60.4 | 42.5 | 40.8 | 13.7 |
| | 3.0 | 45.5 | 60.7 | 65.7 | 57.2 | 227.5 | 67.7 | 57.1 | 52.8 | 48.4 | 73.2 | 52.9 | 46.0 | 12.5 | 68.5 | 53.1 | 53.0 | 17.5 | 60.4 | 42.7 | 40.7 | 16.9 |
| | 4.0 | 44.6 | 60.5 | 68.1 | 55.8 | 240.4 | 67.7 | 56.3 | 53.0 | 60.8 | 73.2 | 52.9 | 46.0 | 15.4 | 68.5 | 53.0 | 53.0 | 21.0 | 60.4 | 42.7 | 40.7 | 19.4 |
| | 5.0 | 44.2 | 60.1 | 69.4 | 55.1 | 246.3 | 67.4 | 55.9 | 53.6 | 76.7 | 73.1 | 52.9 | 46.0 | 18.1 | 68.6 | 52.9 | 53.0 | 24.0 | 60.4 | 42.9 | 40.6 | 21.5 |
| mean_matching | 1.0 | 53.3 | 60.5 | 43.4 | 66.2 | 7.8 | 66.2 | 62.5 | 48.3 | 216.4 | 73.2 | 52.8 | 45.9 | 7.1 | 68.4 | 53.2 | 53.1 | 12.1 | 60.4 | 42.6 | 40.8 | 10.7 |
| | 2.0 | 44.4 | 58.7 | 43.4 | 66.0 | 13.1 | 67.6 | 56.8 | 53.0 | 57.9 | 73.1 | 52.8 | 45.8 | 11.2 | 68.4 | 52.9 | 53.1 | 17.8 | 60.4 | 42.5 | 40.8 | 15.8 |
| | 3.0 | 44.1 | 56.4 | 43.5 | 65.8 | 17.7 | 65.8 | 55.4 | 54.4 | 96.5 | 73.0 | 52.8 | 45.7 | 14.5 | 68.5 | 52.7 | 53.2 | 23.0 | 60.3 | 42.4 | 40.8 | 19.8 |
| | 4.0 | 45.0 | 53.8 | 43.6 | 65.6 | 22.3 | 56.4 | 51.6 | 56.3 | 154.4 | 73.0 | 52.9 | 45.6 | 17.3 | 68.5 | 52.5 | 53.2 | 27.2 | 60.2 | 42.5 | 40.8 | 22.9 |
| | 5.0 | 46.1 | 51.8 | 43.6 | 65.5 | 27.3 | 49.3 | 48.5 | 57.2 | 193.2 | 73.0 | 52.9 | 45.5 | 19.9 | 68.5 | 52.2 | 53.4 | 31.3 | 60.1 | 42.4 | 40.6 | 25.9 |

Table 13: Model SANA, flipping from snoopy to mickey

| method | strength | snoopy | | mickey | | | pikachu | | | | spongebob | | | | dog | | | | legislator | | | |
|---|---|---|---|---|---|---|---|---|---|---|---|---|---|---|---|---|---|---|---|---|---|---|
| | | src-cs | tgt-cs | src-cs | tgt-cs | fid | cs | src-cs | tgt-cs | fid | cs | src-cs | tgt-cs | fid | cs | src-cs | tgt-cs | fid | cs | src-cs | tgt-cs | fid |
| No Steering | - | 79.7 | 58.0 | 56.3 | 76.1 | - | 74.0 | 41.5 | 50.9 | - | 79.0 | 50.7 | 53.8 | - | 67.3 | 57.0 | 57.1 | - | 60.4 | 42.9 | 46.6 | - |
| CASteer | 1.0 | 77.9 | 61.7 | 57.8 | 76.3 | 31.8 | 74.1 | 41.5 | 50.8 | 8.6 | 79.0 | 50.7 | 53.8 | 14.5 | 68.1 | 54.0 | 52.0 | 218.8 | 60.3 | 43.0 | 46.8 | 12.5 |
| | 2.0 | 65.6 | 71.0 | 61.0 | 76.5 | 53.8 | 74.1 | 41.6 | 50.8 | 13.9 | 78.9 | 50.8 | 53.8 | 19.8 | 68.2 | 53.5 | 52.0 | 218.3 | 60.2 | 43.0 | 46.9 | 17.7 |
| | 3.0 | 55.8 | 72.3 | 71.7 | 74.8 | 78.8 | 74.2 | 41.7 | 50.7 | 18.6 | 78.9 | 50.8 | 53.8 | 23.9 | 68.1 | 53.2 | 52.1 | 218.3 | 60.1 | 43.0 | 46.8 | 21.4 |
| | 4.0 | 52.0 | 71.4 | 78.5 | 67.6 | 102.9 | 74.3 | 41.8 | 50.7 | 22.7 | 78.9 | 50.8 | 53.8 | 27.9 | 68.2 | 52.8 | 52.1 | 218.5 | 60.1 | 43.0 | 47.0 | 25.4 |
| | 5.0 | 50.0 | 70.4 | 79.8 | 61.9 | 121.0 | 74.5 | 42.0 | 50.7 | 26.7 | 79.0 | 50.8 | 53.8 | 31.8 | 68.2 | 52.4 | 52.2 | 218.0 | 60.1 | 43.0 | 47.2 | 28.5 |
| LEACE | 1.0 | 76.6 | 63.0 | 58.2 | 76.4 | 36.0 | 74.1 | 41.6 | 51.0 | 3.2 | 79.0 | 50.7 | 53.7 | 10.8 | 68.1 | 54.2 | 52.0 | 219.3 | 60.2 | 42.9 | 46.6 | 5.9 |
| | 2.0 | 62.6 | 71.3 | 57.9 | 72.9 | 170.9 | 74.1 | 41.7 | 51.0 | 5.5 | 78.9 | 50.7 | 53.7 | 15.1 | 68.0 | 54.0 | 52.0 | 220.0 | 60.2 | 42.9 | 46.6 | 9.6 |
| | 3.0 | 55.9 | 71.7 | 75.1 | 73.4 | 88.9 | 74.2 | 41.7 | 51.0 | 7.5 | 78.9 | 50.7 | 53.7 | 18.2 | 68.0 | 53.7 | 52.1 | 218.7 | 60.1 | 42.9 | 46.7 | 12.2 |
| | 4.0 | 52.9 | 68.8 | 79.2 | 66.4 | 111.0 | 74.2 | 41.8 | 51.1 | 9.5 | 78.9 | 50.8 | 53.8 | 21.1 | 68.0 | 53.6 | 52.1 | 218.8 | 60.1 | 42.9 | 46.7 | 14.6 |
| | 5.0 | 51.0 | 65.5 | 79.6 | 60.8 | 130.5 | 74.3 | 41.9 | 51.1 | 11.2 | 78.9 | 50.8 | 53.8 | 24.1 | 68.0 | 53.3 | 52.2 | 219.7 | 59.9 | 42.9 | 46.7 | 16.4 |
| mean_matching | 1.0 | 64.9 | 70.8 | 56.0 | 76.2 | 16.0 | 74.1 | 41.5 | 51.0 | 5.8 | 79.1 | 50.6 | 53.8 | 12.3 | 68.1 | 54.0 | 52.0 | 219.2 | 60.2 | 42.9 | 46.7 | 8.3 |
| | 2.0 | 53.5 | 71.1 | 55.6 | 76.2 | 25.2 | 74.1 | 41.5 | 51.1 | 10.0 | 79.2 | 50.6 | 53.9 | 17.4 | 68.1 | 53.6 | 52.1 | 218.2 | 60.1 | 42.9 | 46.7 | 12.4 |
| | 3.0 | 50.0 | 67.4 | 55.2 | 76.2 | 34.8 | 74.4 | 41.5 | 51.2 | 13.5 | 79.2 | 50.6 | 54.0 | 21.6 | 68.0 | 53.2 | 52.2 | 218.6 | 59.9 | 42.8 | 46.7 | 15.6 |
| | 4.0 | 48.4 | 62.6 | 54.9 | 76.4 | 45.4 | 74.5 | 41.6 | 51.3 | 16.8 | 79.2 | 50.5 | 54.0 | 25.8 | 67.9 | 52.6 | 52.3 | 216.6 | 59.8 | 42.8 | 46.7 | 17.8 |
| | 5.0 | 48.1 | 60.7 | 54.5 | 76.6 | 55.3 | 74.6 | 41.5 | 51.3 | 20.0 | 79.2 | 50.3 | 54.1 | 29.6 | 68.0 | 52.3 | 52.4 | 215.1 | 55.0 | 47.5 | 52.8 | 271.2 |

## 7.10 CONCEPT ERASURE

In this section, we provide experiments for concept erasure. The reason for such is to show that unified MiDSteer framework, that also includes erasure in case $Z_2$ is constant, performs favourably on a variety of tasks. Note that in case of erasure, both MiDSteer and LEACE provide the same solutions, so we will label the results as LEACE / MiDSteer in charts.

For LLM, We test erasure of the concepts $c_s \in$ ("Horse", $c_2$ ="Dog"). Corresponding testing concepts are $t_i = \{c_i\}_{i=1}^5$: $t_1 =$ ("Motorcycle", "Cow", "Dog", "Pig", "Legislator"), $t_2 =$ ("Cat", "Cow", "Wolf", "Pig", "Legislator"). For Diffusion Models, We test erasure of the concepts $c_s \in$ ("Horse", $c_2$ ="Chichuahua"). Corresponding testing concepts are $t_i = \{c_i\}_{i=1}^5$: $t_1 =$ ("Motorcycle", "Cow", "Dog", "Pig", "Legislator"), $t_2 =$ ("Muffin", "Cat", "Dog", "Wolf", "Legislator"),

The setup for erasure is similar to those of main paper experiments (Sec. 5). The only difference is that there is no target concept (i.e. it is a dummy concept). We utilise similar pairs of prompts $(c_1, c_2)$ as in switching experiments, with the goal of removing $c_1$.

Fig. 44b,44d,44a,44c present Pareto plots for concept erasure similar to that for concept switching in sec. 5. More precisely, we compare results on concept switching by applying vanilla steering, LEACE (which is equivalent to MiDSteer in the case of erasure) with different values of $\beta$. We present results on LLama-2-7b and SDXL models aggregated for all three erase concepts. It can be clearly seen, that in each case, MiDSteer achieves much better balance between level of concept switch between c1 and c2 and preservation of other concepts across different values of $\beta$.

Next, in sec. 7.10.1, 7.10.2 we provide detailed Pareto plots for each model, and in sec. 7.10.3,7.10.4 we provide Tables with detailed breakdown of scores for all $\beta$ values and all erased concepts.

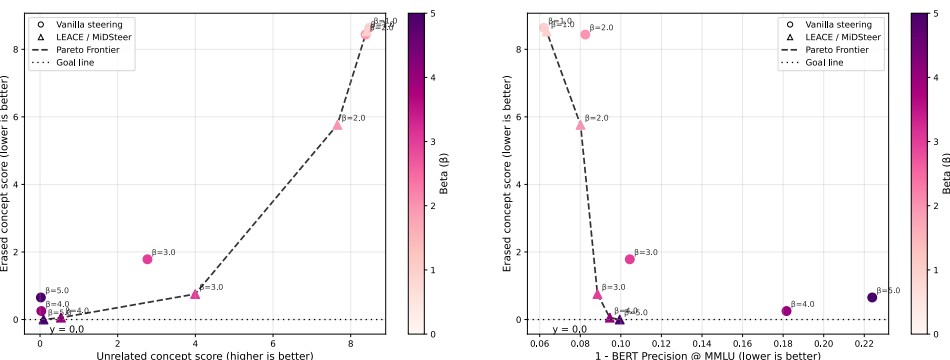

(a) Erased concept score vs CS of unrelated concepts on Llama-2-7b model.

(b) Erased concept score vs 1 - BERT Precision on MMLU on Llama-2-7b model.

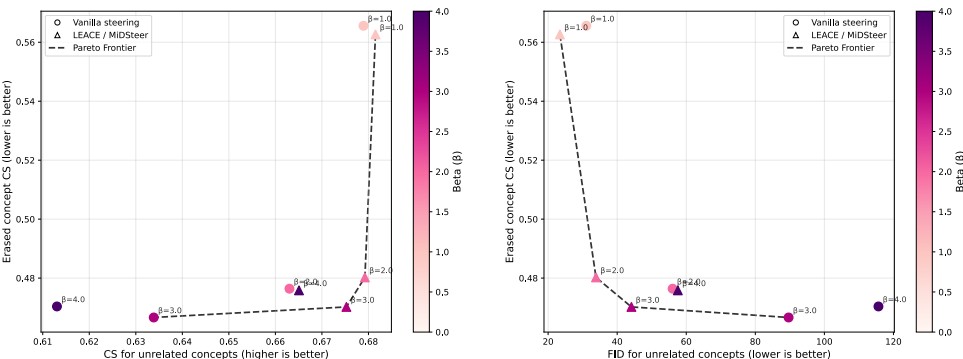

(c) Erased concept score vs CS of unrelated concepts on SDXL model.

(d) Erased concept score vs FID of unrelated concepts on SDXL model.

Figure 44: Pareto efficiency frontiers for concept *erasure* experiments with vanilla steering and LEACE / MiDSteer highlighting different values of $\beta$.

### 7.10.1 PARETO CHARTS FOR LLM CONCEPT ERASURE

In this section, we provide more Pareto charts for LLM concept erasure. When erasing concept $c_s$, for each LLM model we provide 3 types of Pareto plots:

- 1 - BERT Precision score for unrelated concepts (horizontal axis) vs Concept Score (CS) for the erased $c_s$ concept (vertical axis) (fig. 46,49,51)

- 1 - BERT Precision score for MMLU (horizontal axis) vs Concept Score (CS) for the erased $c_s$ concept (vertical axis) (fig. 45,48,51)

- Average Concept Score (CS) for unrelated concepts $c_i$ (horizontal axis) vs Concept Score (CS) for the erased $c_s$ concept (vertical axis) (fig. 47,50,53)

In each case, we see clear superiority of LEACE/MidSteer over other steering approaches.

We additionally provide detailed breakdown of scores for all $\beta$ values and all concepts $c_s, c_i$ in the tables in sec. 7.10.3

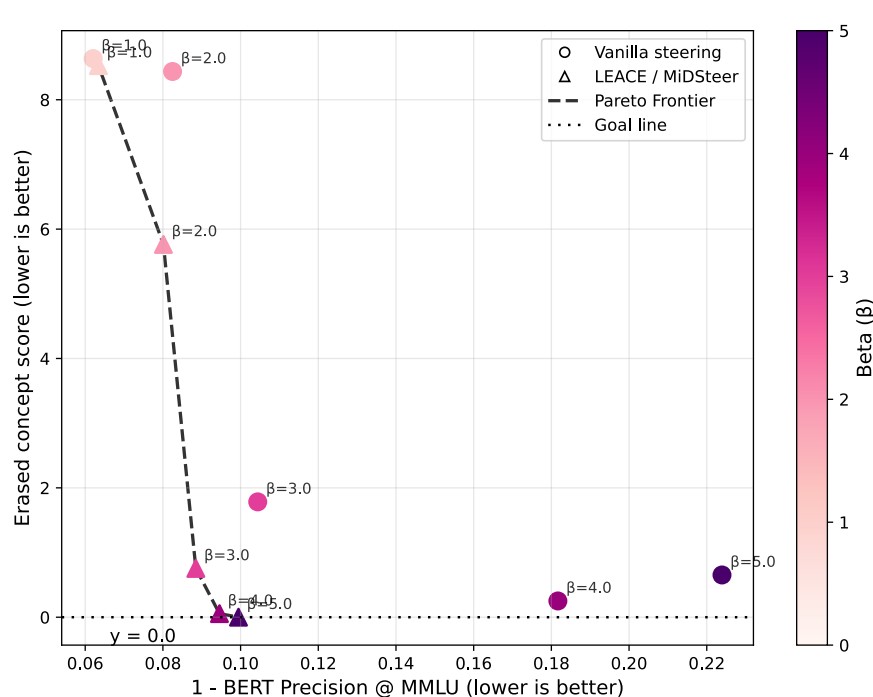

Figure 45: Pareto plot for concept erasure on model llama2-7b

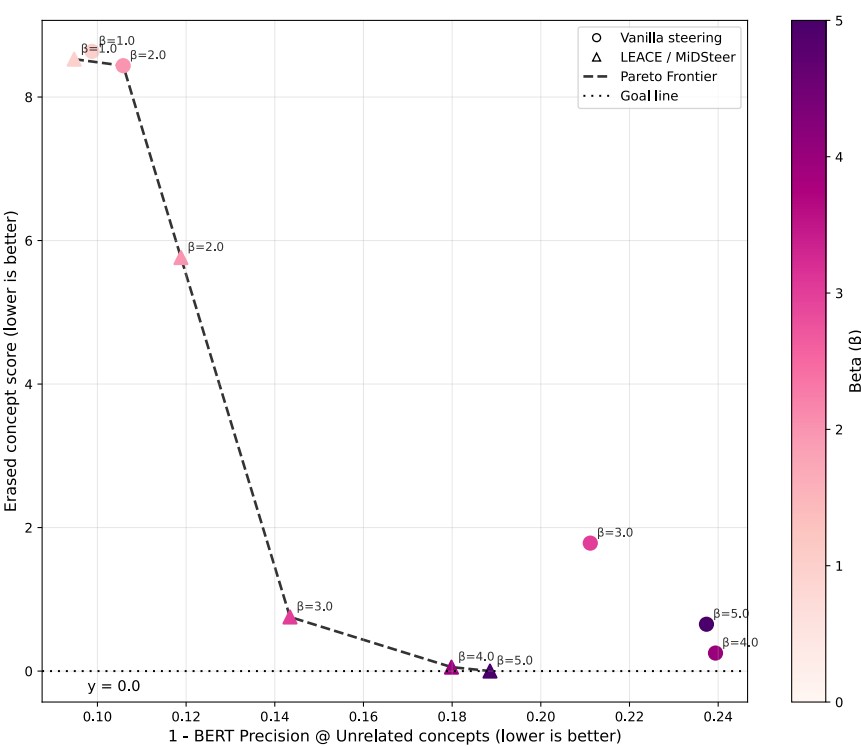

Figure 46: Pareto plot for concept erasure on model llama2-7b

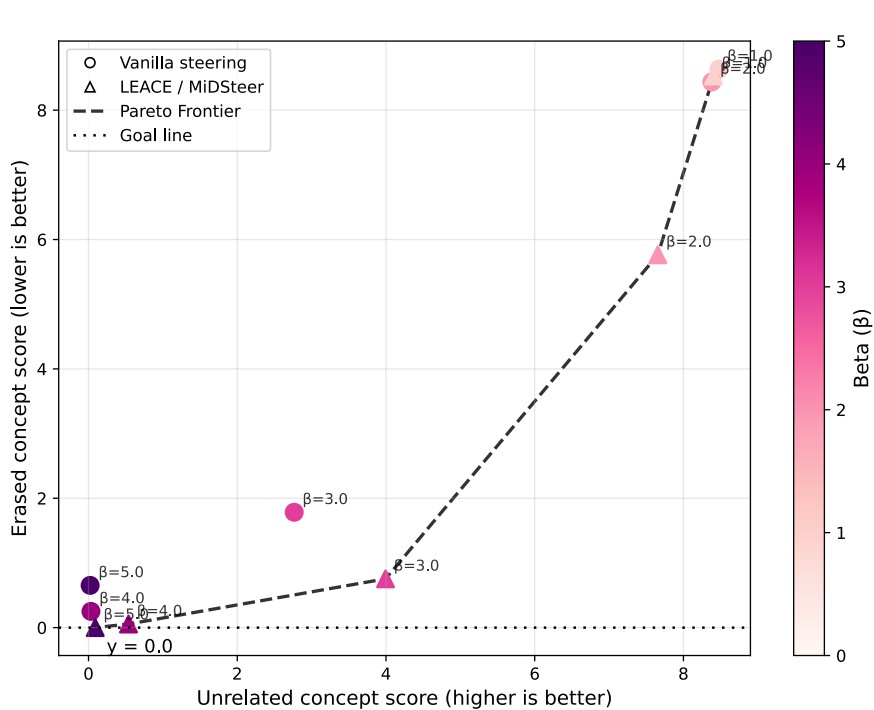

Figure 47: Pareto plot for concept erasure on model llama2-7b

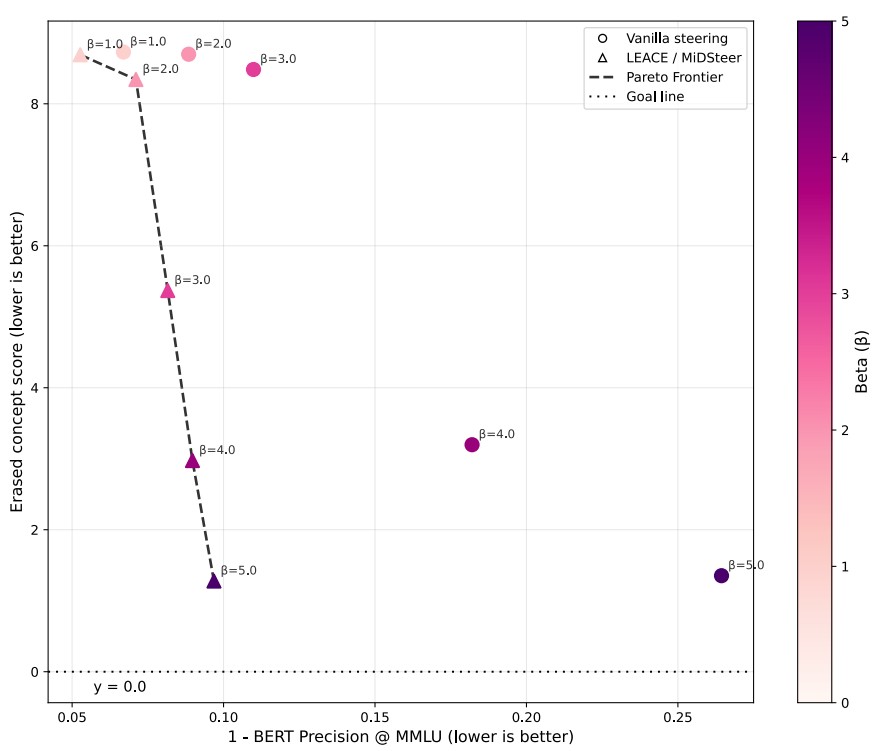

Figure 48: Pareto plot for concept erasure on model qwen-14b

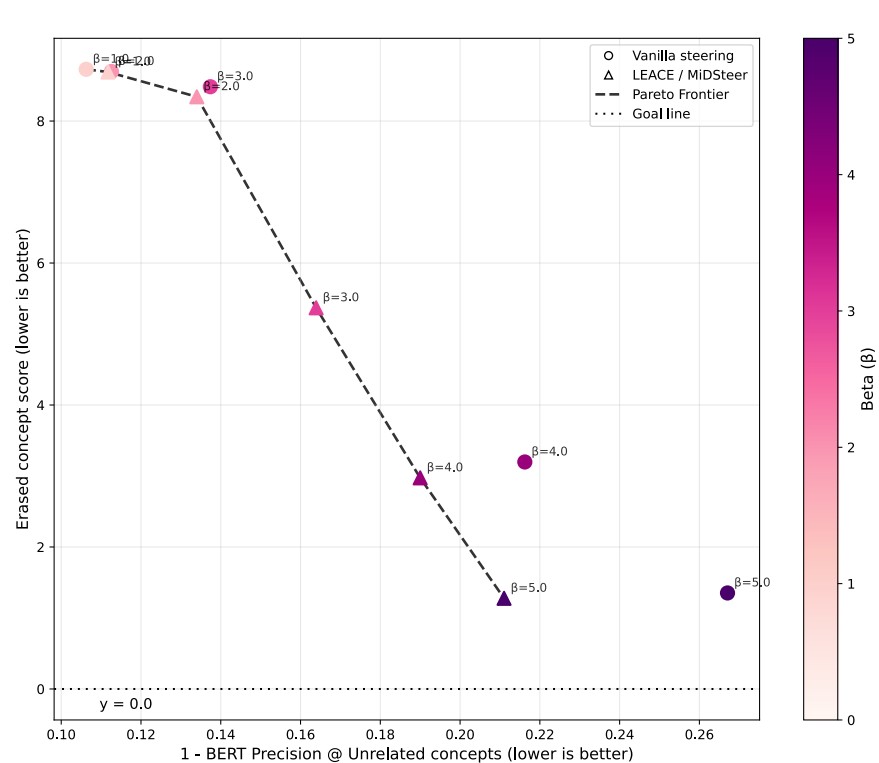

Figure 49: Pareto plot for concept erasure on model qwen-14b

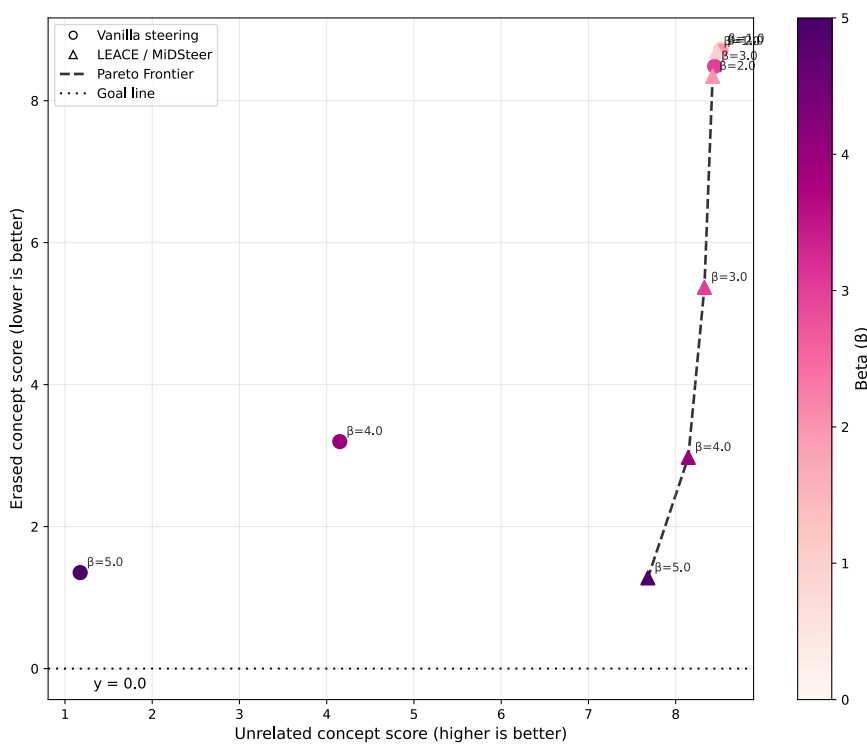

Figure 50: Pareto plot for concept erasure on model qwen-14b

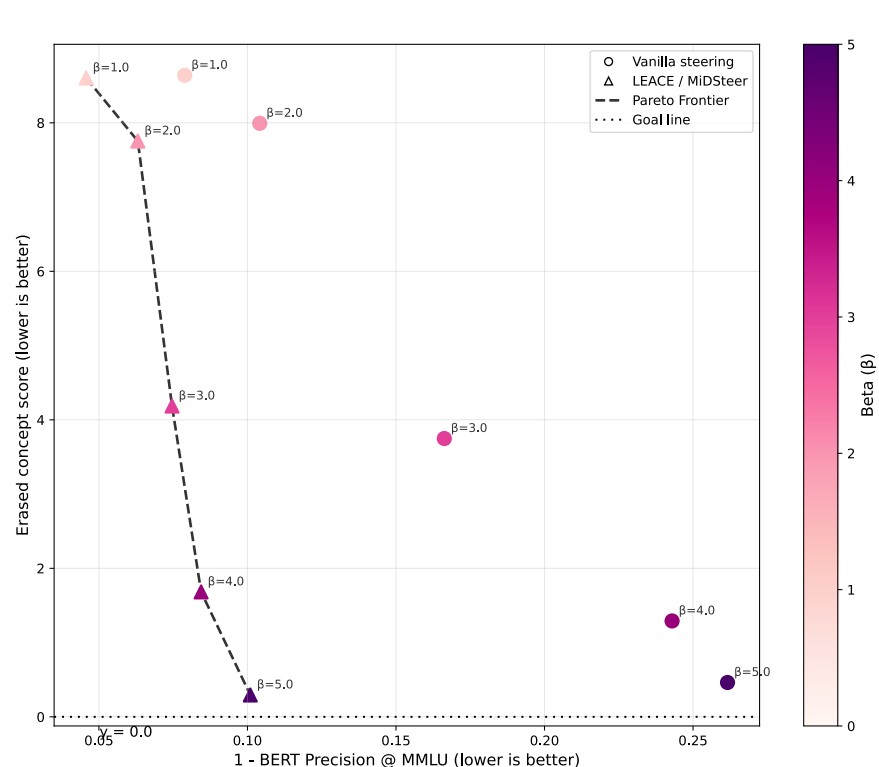

Figure 51: Pareto plot for concept erasure on model qwen-7b

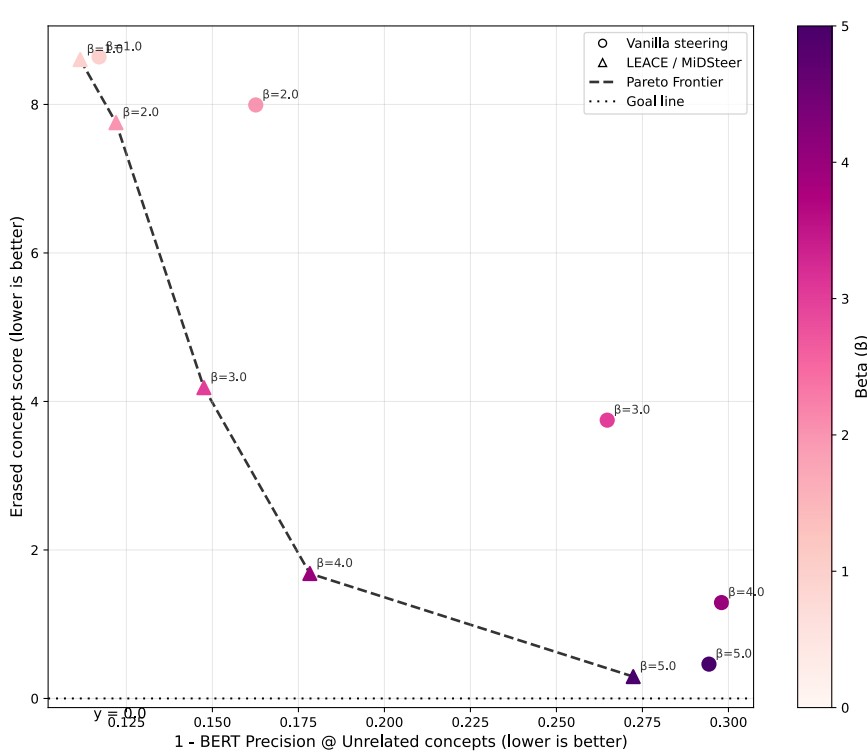

Figure 52: Pareto plot for concept erasure on model qwen-7b

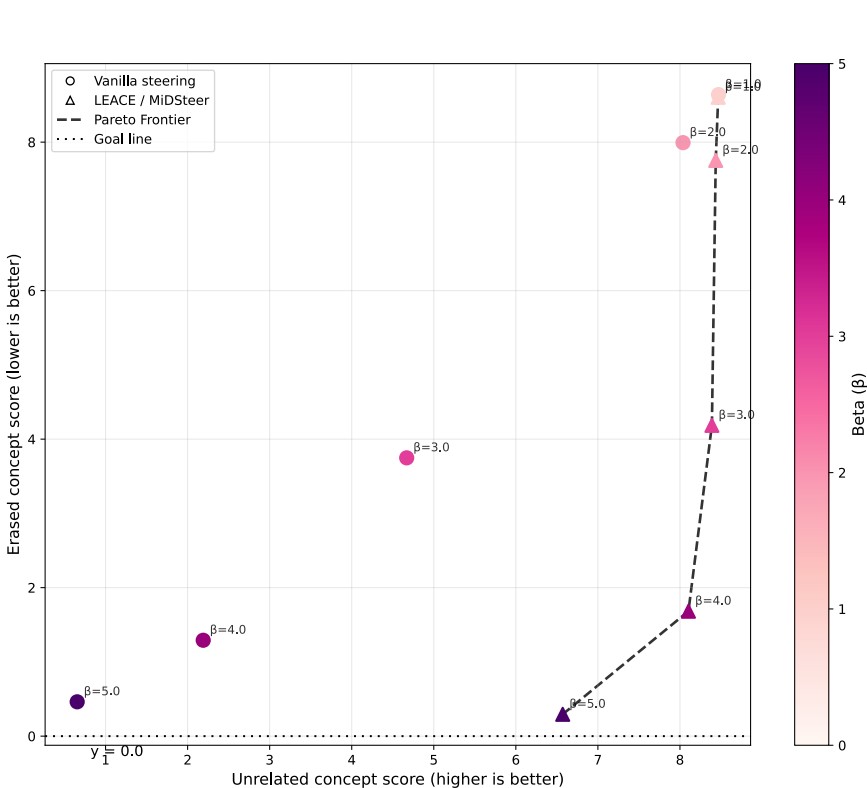

Figure 53: Pareto plot for concept erasure on model qwen-7b

### 7.10.2 PARETO CHARTS FOR IMAGE DIFFUSION CONCEPT ERASURE

In this section, we provide more Pareto charts for LLM concept erasure. When erasing concept $c_s$, for each Diffusion model we provide 2 types of Pareto plots:

- FID score for unrelated concepts (horizontal axis) vs Concept Score (CS) for the erased $c_s$ concept (vertical axis) (fig. 57,57)
- Average Concept Score (CS) for unrelated concepts $c_i$ (horizontal axis) vs Concept Score (CS) for the erased $c_s$ concept (vertical axis) (fig. 56,54)

In each case, we see clear superiority of LEACE/MidSteer over other steering approaches.

We additionally provide detailed breakdown of scores for all $\beta$ values and all concepts $c_s, c_i$ in the tables in sec. 7.10.4

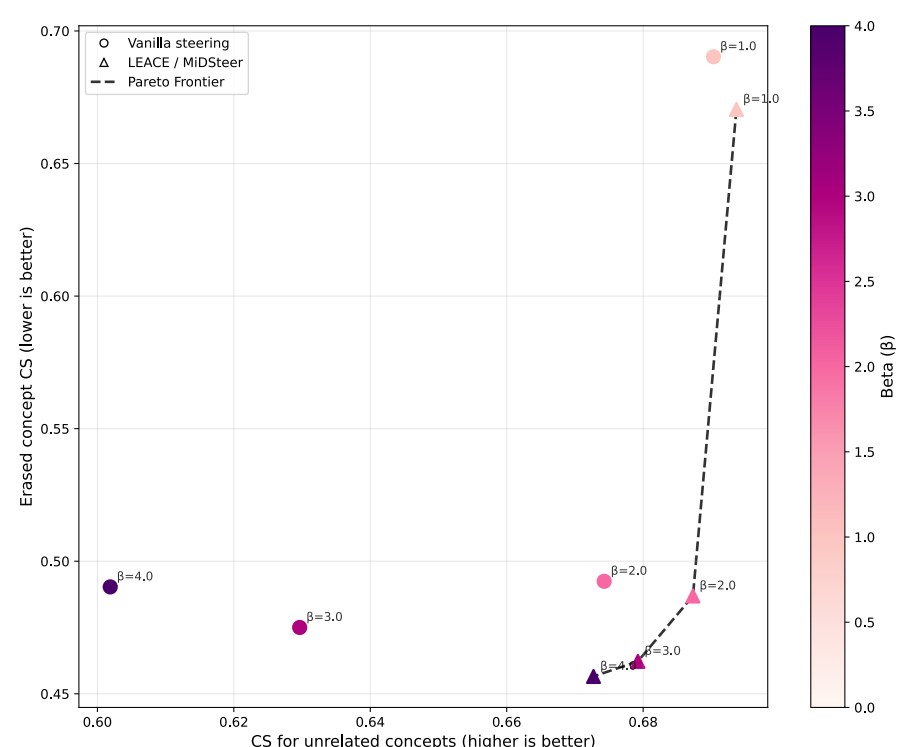

Figure 54: Pareto plot for concept erase on model sana

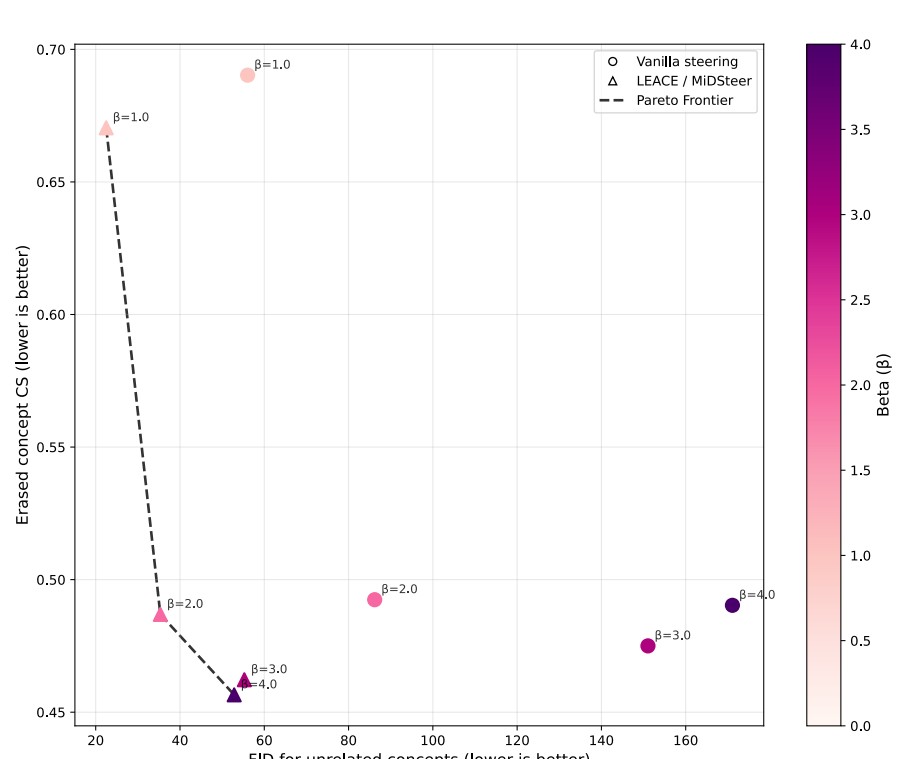

Figure 55: Pareto plot for concept erase on model sana

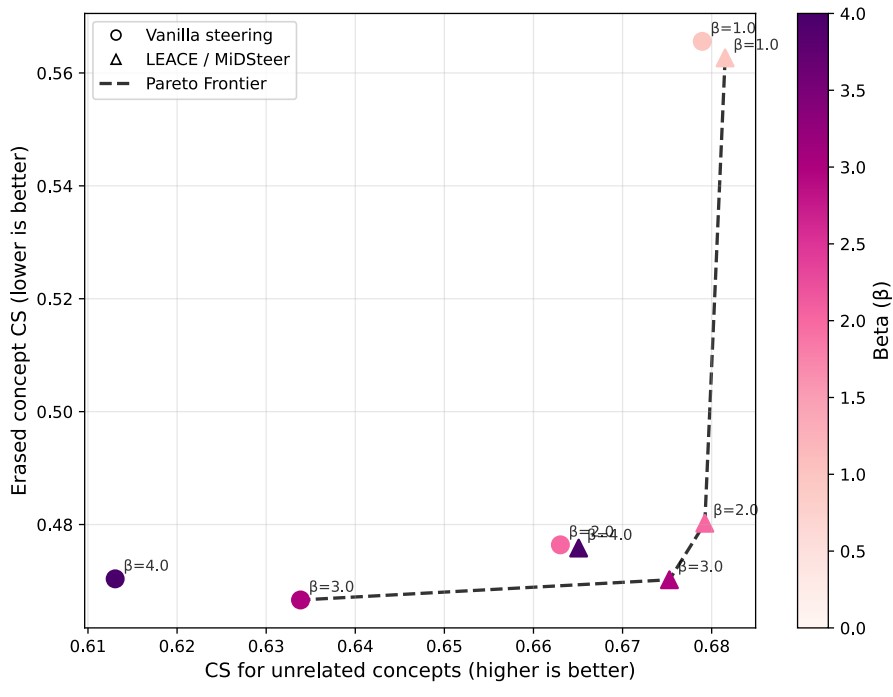

Figure 56: Pareto plot for concept erase on model sdxl

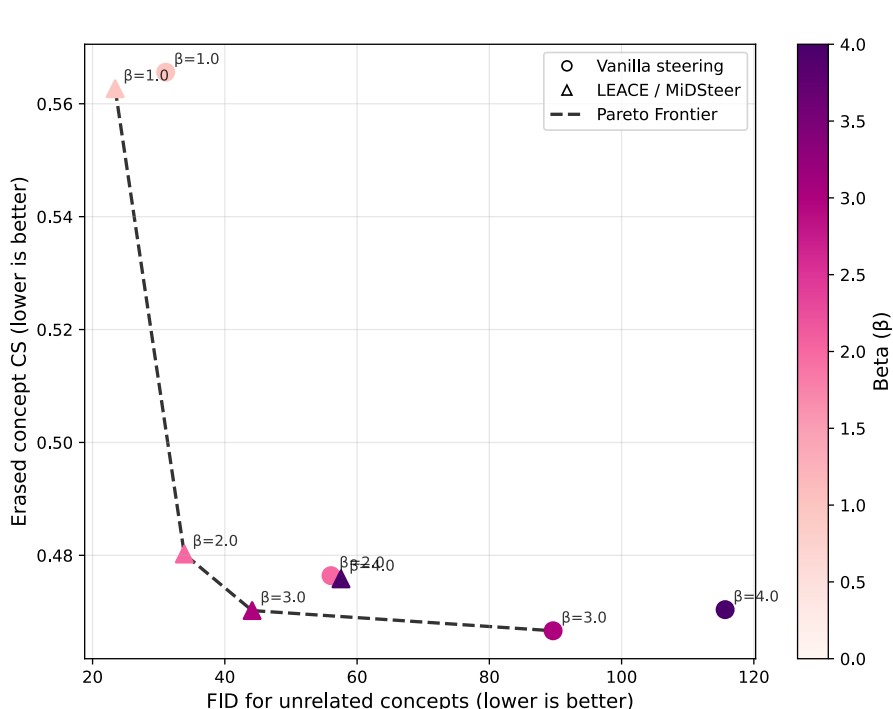

Figure 57: Pareto plot for concept erase on model sdxl

### 7.10.3 RESULTS FOR LLM CONCEPT ERASURE

In this section, in tab. 14,15,18,19,18,17we provide detailed breakdown of scores for all $\beta$ values and all concepts $c_s, c_i$. Pareto plots in sec. 7.10.1 were created based on the scores provided in these tables.

Table 14: Model LLama2-7b, removal of horses

| method | strength | horses cs | motorcycles cs | bertp | cows cs | bertp | pigs cs | bertp | dogs cs | bertp | legislators cs | bertp |
|---|---|---|---|---|---|---|---|---|---|---|---|---|
| No Steering | - | 8.6 | 8.5 | - | 8.4 | - | 8.5 | - | 8.7 | - | 8.4 | - |
| CASteer | 1.0 | 8.6 | 8.5 | 0.91 | 8.4 | 0.90 | 8.4 | 0.90 | 8.6 | 0.90 | 8.5 | 0.89 |
| | 2.0 | 8.4 | 8.5 | 0.90 | 8.4 | 0.89 | 8.3 | 0.89 | 8.5 | 0.90 | 8.3 | 0.89 |
| | 3.0 | 1.5 | 3.4 | 0.80 | 2.4 | 0.78 | 1.4 | 0.78 | 2.9 | 0.79 | 2.6 | 0.79 |
| | 4.0 | 0.4 | 0.0 | 0.76 | 0.1 | 0.76 | 0.0 | 0.77 | 0.0 | 0.77 | 0.0 | 0.77 |
| | 5.0 | 1.0 | 0.0 | 0.76 | 0.1 | 0.77 | 0.0 | 0.77 | 0.0 | 0.77 | 0.0 | 0.77 |
| LEACE | 1.0 | 8.5 | 8.5 | 0.91 | 8.3 | 0.91 | 8.4 | 0.90 | 8.6 | 0.91 | 8.3 | 0.90 |
| | 2.0 | 5.0 | 8.3 | 0.89 | 7.3 | 0.87 | 7.4 | 0.88 | 8.0 | 0.89 | 7.8 | 0.88 |
| | 3.0 | 0.2 | 4.6 | 0.86 | 3.3 | 0.85 | 3.7 | 0.85 | 4.0 | 0.86 | 4.6 | 0.86 |
| | 4.0 | 0.0 | 1.0 | 0.82 | 0.2 | 0.81 | 0.2 | 0.81 | 0.4 | 0.82 | 1.1 | 0.82 |
| | 5.0 | 0.0 | 0.1 | 0.81 | 0.0 | 0.81 | 0.0 | 0.81 | 0.1 | 0.81 | 0.2 | 0.81 |

Table 15: Model LLama2-7b, removal of dogs

| method | strength | dogs cs | cats cs | bertp | wolves cs | bertp | cows cs | bertp | pigs cs | bertp | legislators cs | bertp |
|---|---|---|---|---|---|---|---|---|---|---|---|---|
| No Steering | - | 8.6 | 8.6 | - | 8.5 | - | 8.4 | - | 8.4 | - | 8.5 | - |
| CASteer | 1.0 | 8.6 | 8.5 | 0.90 | 8.4 | 0.90 | 8.4 | 0.90 | 8.4 | 0.90 | 8.5 | 0.90 |
| | 2.0 | 8.5 | 8.5 | 0.90 | 8.3 | 0.89 | 8.3 | 0.89 | 8.3 | 0.89 | 8.4 | 0.89 |
| | 3.0 | 2.1 | 3.1 | 0.79 | 3.6 | 0.80 | 3.0 | 0.79 | 1.9 | 0.78 | 3.4 | 0.79 |
| | 4.0 | 0.1 | 0.0 | 0.76 | 0.1 | 0.76 | 0.0 | 0.75 | 0.0 | 0.75 | 0.0 | 0.76 |
| | 5.0 | 0.3 | 0.0 | 0.76 | 0.0 | 0.75 | 0.0 | 0.76 | 0.0 | 0.76 | 0.0 | 0.75 |
| LEACE | 1.0 | 8.5 | 8.5 | 0.91 | 8.4 | 0.91 | 8.3 | 0.91 | 8.4 | 0.91 | 8.4 | 0.90 |
| | 2.0 | 6.5 | 7.8 | 0.88 | 7.6 | 0.88 | 7.2 | 0.88 | 7.5 | 0.88 | 7.7 | 0.88 |
| | 3.0 | 1.3 | 3.8 | 0.86 | 4.2 | 0.86 | 3.4 | 0.85 | 3.7 | 0.85 | 4.6 | 0.86 |
| | 4.0 | 0.1 | 0.4 | 0.82 | 0.2 | 0.83 | 0.5 | 0.82 | 0.3 | 0.82 | 1.0 | 0.82 |
| | 5.0 | 0.0 | 0.1 | 0.81 | 0.1 | 0.82 | 0.1 | 0.82 | 0.1 | 0.81 | 0.1 | 0.81 |

Table 16: Model Qwen2.5-7b, removal of horses

| method | strength | horses cs | motorcycles cs | bertp | cows cs | bertp | pigs cs | bertp | dogs cs | bertp | legislators cs | bertp |
|---|---|---|---|---|---|---|---|---|---|---|---|---|
| No Steering | - | 8.7 | 8.6 | - | 8.4 | - | 8.4 | - | 8.7 | - | 8.5 | - |
| CASteer | 1.0 | 8.6 | 8.5 | 0.89 | 8.4 | 0.88 | 8.4 | 0.88 | 8.7 | 0.88 | 8.6 | 0.88 |
| | 2.0 | 8.0 | 7.8 | 0.84 | 8.1 | 0.84 | 7.8 | 0.83 | 8.1 | 0.83 | 8.1 | 0.83 |
| | 3.0 | 3.6 | 4.0 | 0.72 | 5.1 | 0.73 | 4.5 | 0.73 | 4.7 | 0.73 | 3.9 | 0.72 |
| | 4.0 | 1.2 | 2.8 | 0.69 | 2.6 | 0.70 | 1.9 | 0.70 | 2.6 | 0.70 | 2.0 | 0.69 |
| | 5.0 | 0.6 | 1.0 | 0.70 | 0.7 | 0.70 | 0.3 | 0.70 | 0.7 | 0.70 | 0.9 | 0.69 |
| LEACE | 1.0 | 8.5 | 8.5 | 0.89 | 8.4 | 0.89 | 8.4 | 0.89 | 8.7 | 0.89 | 8.5 | 0.89 |
| | 2.0 | 6.9 | 8.5 | 0.89 | 8.4 | 0.88 | 8.4 | 0.88 | 8.7 | 0.88 | 8.5 | 0.88 |
| | 3.0 | 0.6 | 8.5 | 0.87 | 8.3 | 0.85 | 8.3 | 0.85 | 8.6 | 0.86 | 8.4 | 0.86 |
| | 4.0 | 0.2 | 8.3 | 0.85 | 8.1 | 0.83 | 8.1 | 0.82 | 8.4 | 0.83 | 8.2 | 0.83 |
| | 5.0 | 0.0 | 7.3 | 0.76 | 6.7 | 0.73 | 6.2 | 0.73 | 6.9 | 0.73 | 7.2 | 0.74 |

Table 17: Model Qwen2.5-7b, removal of dogs

| method | strength | dogs cs | cats cs | cats bertp | wolves cs | wolves bertp | cows cs | cows bertp | pigs cs | pigs bertp | legislators cs | legislators bertp |
|---|---|---|---|---|---|---|---|---|---|---|---|---|
| No Steering | - | 8.7 | 8.5 | - | 8.3 | - | 8.4 | - | 8.4 | - | 8.5 | - |
| CASteer | 1.0 | 8.6 | 8.5 | 0.88 | 8.3 | 0.88 | 8.4 | 0.88 | 8.4 | 0.88 | 8.5 | 0.88 |
|  | 2.0 | 8.0 | 8.1 | 0.84 | 8.0 | 0.84 | 8.1 | 0.84 | 7.9 | 0.84 | 8.3 | 0.84 |
|  | 3.0 | 3.9 | 5.0 | 0.74 | 4.6 | 0.74 | 5.4 | 0.75 | 4.8 | 0.75 | 4.6 | 0.75 |
|  | 4.0 | 1.4 | 2.2 | 0.71 | 1.7 | 0.71 | 2.4 | 0.71 | 1.6 | 0.71 | 2.1 | 0.71 |
|  | 5.0 | 0.3 | 0.7 | 0.71 | 0.4 | 0.71 | 0.7 | 0.71 | 0.4 | 0.72 | 0.8 | 0.71 |
| LEACE | 1.0 | 8.7 | 8.5 | 0.89 | 8.4 | 0.88 | 8.4 | 0.89 | 8.4 | 0.88 | 8.5 | 0.89 |
|  | 2.0 | 8.7 | 8.5 | 0.88 | 8.3 | 0.87 | 8.4 | 0.88 | 8.4 | 0.87 | 8.5 | 0.88 |
|  | 3.0 | 7.8 | 8.5 | 0.85 | 8.2 | 0.84 | 8.3 | 0.85 | 8.3 | 0.85 | 8.4 | 0.86 |
|  | 4.0 | 3.2 | 8.2 | 0.83 | 7.7 | 0.79 | 8.1 | 0.83 | 7.9 | 0.81 | 8.2 | 0.82 |
|  | 5.0 | 0.6 | 6.5 | 0.72 | 5.6 | 0.71 | 6.6 | 0.72 | 5.9 | 0.72 | 6.9 | 0.72 |

Table 18: Model Qwen2.5-14b, removal of horses

| method | strength | horses cs | motorcycles cs | motorcycles bertp | cows cs | cows bertp | pigs cs | pigs bertp | dogs cs | dogs bertp | legislators cs | legislators bertp |
|---|---|---|---|---|---|---|---|---|---|---|---|---|
| No Steering | - | 8.7 | 8.6 | - | 8.4 | - | 8.4 | - | 8.7 | - | 8.5 | - |
| CASteer | 1.0 | 8.7 | 8.6 | 0.90 | 8.4 | 0.90 | 8.5 | 0.89 | 8.7 | 0.90 | 8.5 | 0.89 |
|  | 2.0 | 8.7 | 8.6 | 0.89 | 8.4 | 0.89 | 8.5 | 0.89 | 8.7 | 0.89 | 8.5 | 0.89 |
|  | 3.0 | 8.5 | 8.5 | 0.86 | 8.4 | 0.86 | 8.4 | 0.86 | 8.6 | 0.86 | 8.4 | 0.87 |
|  | 4.0 | 2.9 | 4.2 | 0.78 | 5.0 | 0.78 | 5.0 | 0.78 | 4.3 | 0.79 | 4.3 | 0.79 |
|  | 5.0 | 0.8 | 1.5 | 0.73 | 1.2 | 0.73 | 2.0 | 0.74 | 1.4 | 0.74 | 0.6 | 0.73 |
| LEACE | 1.0 | 8.7 | 8.6 | 0.90 | 8.4 | 0.89 | 8.4 | 0.88 | 8.7 | 0.89 | 8.5 | 0.88 |
|  | 2.0 | 8.0 | 8.5 | 0.88 | 8.3 | 0.87 | 8.3 | 0.86 | 8.7 | 0.87 | 8.4 | 0.86 |
|  | 3.0 | 3.0 | 8.5 | 0.85 | 8.3 | 0.84 | 8.3 | 0.83 | 8.6 | 0.84 | 8.3 | 0.83 |
|  | 4.0 | 1.2 | 8.3 | 0.82 | 8.1 | 0.81 | 8.0 | 0.80 | 8.4 | 0.81 | 8.1 | 0.80 |
|  | 5.0 | 0.3 | 7.8 | 0.79 | 7.5 | 0.79 | 7.5 | 0.78 | 7.9 | 0.79 | 7.7 | 0.79 |

Table 19: Model Qwen2.5-14b, removal of dogs

| method | strength | dogs cs | cats cs | cats bertp | wolves cs | wolves bertp | cows cs | cows bertp | pigs cs | pigs bertp | legislators cs | legislators bertp |
|---|---|---|---|---|---|---|---|---|---|---|---|---|
| No Steering | - | 8.7 | 8.5 | - | 8.4 | - | 8.4 | - | 8.5 | - | 8.5 | - |
| CASteer | 1.0 | 8.7 | 8.5 | 0.90 | 8.4 | 0.89 | 8.5 | 0.89 | 8.5 | 0.89 | 8.6 | 0.89 |
|  | 2.0 | 8.7 | 8.5 | 0.89 | 8.4 | 0.88 | 8.4 | 0.89 | 8.5 | 0.89 | 8.5 | 0.89 |
|  | 3.0 | 8.4 | 8.5 | 0.86 | 8.3 | 0.86 | 8.4 | 0.86 | 8.4 | 0.86 | 8.4 | 0.87 |
|  | 4.0 | 3.5 | 3.7 | 0.79 | 3.1 | 0.77 | 4.2 | 0.78 | 4.2 | 0.78 | 3.4 | 0.79 |
|  | 5.0 | 1.9 | 0.9 | 0.73 | 1.6 | 0.73 | 0.9 | 0.73 | 1.2 | 0.73 | 0.4 | 0.73 |
| LEACE | 1.0 | 8.7 | 8.5 | 0.89 | 8.4 | 0.88 | 8.4 | 0.89 | 8.4 | 0.89 | 8.5 | 0.89 |
|  | 2.0 | 8.7 | 8.4 | 0.87 | 8.4 | 0.86 | 8.3 | 0.87 | 8.4 | 0.86 | 8.4 | 0.86 |
|  | 3.0 | 7.7 | 8.4 | 0.84 | 8.2 | 0.83 | 8.3 | 0.84 | 8.2 | 0.83 | 8.3 | 0.83 |
|  | 4.0 | 4.7 | 8.3 | 0.82 | 7.9 | 0.81 | 8.2 | 0.82 | 8.0 | 0.80 | 8.1 | 0.80 |
|  | 5.0 | 2.3 | 7.9 | 0.79 | 7.5 | 0.79 | 7.7 | 0.80 | 7.5 | 0.78 | 7.7 | 0.79 |

### 7.10.4 RESULTS FOR IMAGE DIFFUSION CONCEPT ERASURE

In this section, in tab. 22,25,21,25 we provide detailed breakdown of scores for all $\beta$ values and all concepts $c_s, c_i$. Pareto plots in sec. 7.10.2 were created based on the scores provided in these tables.

Table 20: Model SDXL, removal of snoopy

| method | strength | snoopy cs | mickey cs | fid | pikachu cs | fid | spongebob cs | fid | dog cs | fid | legislator cs | fid |
|---|---|---|---|---|---|---|---|---|---|---|---|---|
| No Steering | - | 74.3 | 73.1 | - | 72.6 | - | 75.1 | - | 66.3 | - | 60.8 | - |
| CASteer | 1.0 | 55.8 | 70.1 | 54.9 | 72.5 | 30.3 | 73.9 | 50.9 | 66.2 | 30.6 | 60.9 | 22.6 |
| | 1.5 | 49.9 | 67.9 | 71.8 | 72.5 | 39.9 | 72.8 | 66.0 | 66.2 | 39.4 | 60.9 | 27.5 |
| | 2.0 | 47.0 | 65.2 | 90.5 | 72.6 | 51.2 | 71.0 | 85.2 | 66.2 | 48.1 | 60.8 | 31.7 |
| | 2.5 | 45.6 | 62.2 | 111.0 | 72.5 | 65.7 | 68.6 | 109.4 | 66.1 | 58.1 | 60.9 | 35.3 |
| | 3.0 | 45.3 | 58.8 | 132.1 | 72.2 | 83.7 | 65.3 | 138.2 | 66.2 | 68.0 | 60.8 | 38.7 |
| | 4.0 | 45.3 | 53.5 | 169.0 | 71.6 | 123.4 | 59.0 | 189.5 | 66.1 | 83.3 | 60.9 | 45.8 |
| | 5.0 | 45.9 | 50.7 | 195.3 | 69.3 | 153.0 | 55.7 | 218.2 | 65.6 | 99.9 | 61.0 | 52.9 |
| LEACE | 1.0 | 56.7 | 72.2 | 35.7 | 72.9 | 21.3 | 74.1 | 42.2 | 66.3 | 20.7 | 60.6 | 26.9 |
| | 1.5 | 51.2 | 71.7 | 42.3 | 73.0 | 25.8 | 73.7 | 50.1 | 66.3 | 26.5 | 60.5 | 32.5 |
| | 2.0 | 48.3 | 71.0 | 48.2 | 73.2 | 29.6 | 73.3 | 57.8 | 66.3 | 31.6 | 60.4 | 36.4 |
| | 2.5 | 46.5 | 70.3 | 53.9 | 73.3 | 33.0 | 72.8 | 66.5 | 66.4 | 36.5 | 60.2 | 41.3 |
| | 3.0 | 45.8 | 69.6 | 59.8 | 73.5 | 36.7 | 72.1 | 75.3 | 66.4 | 40.8 | 60.0 | 46.5 |
| | 4.0 | 45.8 | 67.9 | 72.2 | 73.7 | 44.8 | 70.8 | 91.8 | 66.5 | 49.5 | 59.4 | 56.1 |
| | 5.0 | 47.0 | 66.1 | 85.9 | 73.7 | 53.8 | 69.4 | 114.1 | 66.4 | 57.1 | 58.6 | 69.2 |

Table 21: Model SDXL, removal of chihuahua

| method | strength | chihuahua cs | muffin cs | fid | dog cs | fid | wolf cs | fid | cat cs | fid | legislator cs | fid |
|---|---|---|---|---|---|---|---|---|---|---|---|---|
| No Steering | - | 75.9 | 68.2 | - | 66.3 | - | 71.8 | - | 67.5 | - | 60.8 | - |
| CASteer | 1.0 | 54.6 | 68.1 | 19.7 | 65.0 | 58.2 | 72.5 | 25.9 | 67.0 | 35.2 | 60.9 | 22.7 |
| | 1.5 | 48.5 | 68.2 | 24.0 | 61.2 | 99.9 | 72.6 | 34.0 | 66.5 | 48.9 | 60.8 | 27.8 |
| | 2.0 | 47.6 | 68.0 | 27.3 | 54.1 | 155.5 | 72.6 | 44.1 | 64.6 | 69.0 | 60.9 | 31.9 |
| | 2.5 | 47.2 | 67.9 | 31.1 | 50.7 | 177.8 | 72.2 | 61.2 | 60.5 | 102.6 | 60.8 | 35.8 |
| | 3.0 | 46.9 | 67.9 | 34.6 | 49.7 | 187.7 | 70.0 | 96.2 | 55.8 | 141.5 | 60.8 | 39.4 |
| | 4.0 | 47.8 | 67.7 | 42.2 | 49.0 | 198.2 | 62.2 | 191.6 | 50.7 | 186.3 | 60.7 | 45.7 |
| | 5.0 | 49.7 | 67.6 | 49.5 | 48.9 | 209.3 | 57.8 | 228.4 | 49.4 | 201.5 | 60.7 | 52.1 |
| LEACE | 1.0 | 55.0 | 68.2 | 20.0 | 65.8 | 35.2 | 72.3 | 17.1 | 67.4 | 22.1 | 60.9 | 21.8 |
| | 1.5 | 48.5 | 68.1 | 25.0 | 65.5 | 47.6 | 72.5 | 21.3 | 67.3 | 27.1 | 60.8 | 27.0 |
| | 2.0 | 47.4 | 68.1 | 29.0 | 65.0 | 61.1 | 72.6 | 25.0 | 67.3 | 31.4 | 60.9 | 31.0 |
| | 2.5 | 47.0 | 68.2 | 32.6 | 64.1 | 74.8 | 72.8 | 28.6 | 67.2 | 35.1 | 60.8 | 34.2 |
| | 3.0 | 47.2 | 68.1 | 36.0 | 62.7 | 92.4 | 72.9 | 32.4 | 67.1 | 38.5 | 60.8 | 36.5 |
| | 4.0 | 48.6 | 68.0 | 42.3 | 57.6 | 131.4 | 73.1 | 39.4 | 66.9 | 45.1 | 60.7 | 41.9 |
| | 5.0 | 50.2 | 68.0 | 49.4 | 53.7 | 162.6 | 73.1 | 48.3 | 66.5 | 52.5 | 60.5 | 48.3 |

Table 22: Model SDXL, removal of horse

| method | strength | horse cs | motorcycle cs | fid | cow cs | fid | pig cs | fid | dog cs | fid | legislator cs | fid |
|---|---|---|---|---|---|---|---|---|---|---|---|---|
| No Steering | - | 71.0 | 70.7 | - | 72.7 | - | 71.8 | - | 66.3 | - | 60.8 | - |
| CASteer | 1.0 | 59.3 | 70.7 | 12.9 | 71.9 | 30.1 | 71.8 | 20.8 | 65.9 | 29.9 | 61.0 | 21.3 |
| | 1.5 | 49.8 | 70.7 | 15.6 | 71.2 | 46.6 | 71.8 | 27.6 | 65.8 | 36.5 | 61.1 | 26.2 |
| | 2.0 | 48.3 | 70.6 | 17.4 | 69.2 | 79.8 | 71.9 | 36.5 | 65.7 | 42.2 | 61.0 | 30.3 |
| | 2.5 | 47.9 | 70.7 | 19.5 | 62.1 | 152.5 | 72.0 | 45.9 | 65.4 | 48.4 | 60.9 | 33.6 |
| | 3.0 | 47.8 | 70.7 | 21.7 | 54.7 | 211.1 | 72.0 | 60.1 | 65.0 | 54.7 | 60.9 | 37.0 |
| | 4.0 | 48.0 | 70.7 | 26.9 | 51.1 | 227.9 | 71.8 | 92.3 | 63.9 | 69.4 | 60.8 | 43.3 |
| | 5.0 | 49.3 | 70.8 | 35.1 | 50.4 | 238.4 | 69.8 | 138.4 | 62.2 | 89.4 | 60.6 | 49.4 |
| LEACE | 1.0 | 57.1 | 70.6 | 11.5 | 72.3 | 20.5 | 71.8 | 11.6 | 66.1 | 19.7 | 60.7 | 25.1 |
| | 1.5 | 49.6 | 70.6 | 14.0 | 72.0 | 26.0 | 71.9 | 14.1 | 66.1 | 23.7 | 60.5 | 29.6 |
| | 2.0 | 48.4 | 70.6 | 15.9 | 71.8 | 33.1 | 71.9 | 16.2 | 66.0 | 28.0 | 60.4 | 34.1 |
| | 2.5 | 48.0 | 70.6 | 17.5 | 71.4 | 39.9 | 72.0 | 17.3 | 66.1 | 30.9 | 60.3 | 38.2 |
| | 3.0 | 48.1 | 70.6 | 19.3 | 70.7 | 53.5 | 72.0 | 19.1 | 66.1 | 33.6 | 60.3 | 41.6 |
| | 4.0 | 48.4 | 70.5 | 23.0 | 64.5 | 115.1 | 72.1 | 23.1 | 66.1 | 38.6 | 59.9 | 49.2 |
| | 5.0 | 49.7 | 70.3 | 27.4 | 56.4 | 197.4 | 72.2 | 27.8 | 66.1 | 44.6 | 59.4 | 58.8 |

Table 23: Model SDXL, removal of horse

| method | strength | horse cs | motorcycle cs | fid | cow cs | fid | pig cs | fid | dog cs | fid | legislator cs | fid |
|---|---|---|---|---|---|---|---|---|---|---|---|---|
| No Steering | - | 72.1 | 70.5 | - | 73.8 | - | 73.5 | - | 68.1 | - | 60.4 | - |
| CASteer | 1.0 | 70.8 | 70.1 | 21.3 | 74.1 | 36.4 | 73.7 | 28.2 | 67.8 | 29.2 | 60.2 | 21.2 |
| | 2.0 | 52.2 | 70.9 | 45.3 | 72.0 | 93.0 | 74.1 | 49.3 | 67.4 | 44.2 | 59.8 | 36.3 |
| | 3.0 | 51.3 | 69.3 | 105.0 | 61.5 | 216.9 | 65.9 | 261.3 | 65.5 | 71.9 | 59.3 | 58.4 |
| | 4.0 | 56.7 | 62.3 | 186.2 | 59.0 | 242.8 | 67.3 | 175.3 | 62.5 | 118.4 | 58.9 | 90.1 |
| | 5.0 | 52.5 | 60.4 | 221.0 | 58.7 | 249.9 | 65.0 | 205.0 | 59.3 | 161.3 | 58.4 | 130.3 |
| LEACE | 1.0 | 71.1 | 70.5 | 7.5 | 73.8 | 14.4 | 73.4 | 11.4 | 68.0 | 9.2 | 60.4 | 11.1 |
| | 2.0 | 52.0 | 70.4 | 10.0 | 73.9 | 20.9 | 73.5 | 16.7 | 68.0 | 14.2 | 60.4 | 15.6 |
| | 3.0 | 49.7 | 70.3 | 12.2 | 74.0 | 25.8 | 73.5 | 21.4 | 67.9 | 18.4 | 60.3 | 19.1 |
| | 4.0 | 48.8 | 70.4 | 13.8 | 74.2 | 30.9 | 73.5 | 25.6 | 67.8 | 21.3 | 60.2 | 21.8 |
| | 5.0 | 53.7 | 70.3 | 15.2 | 74.1 | 36.2 | 73.6 | 29.7 | 67.8 | 24.4 | 60.1 | 24.2 |

Table 24: Model SANA, removal of snoopy

| method | strength | snoopy cs | mickey cs | fid | pikachu cs | fid | spongebob cs | fid | dog cs | fid | legislator cs | fid |
|---|---|---|---|---|---|---|---|---|---|---|---|---|
| No Steering | - | 79.7 | 76.1 | - | 74.0 | - | 79.0 | - | 68.1 | - | 60.4 | - |
| CASteer | 1.0 | 60.6 | 75.3 | 64.0 | 74.1 | 41.3 | 79.0 | 43.9 | 68.0 | 42.1 | 60.8 | 23.3 |
| | 2.0 | 46.0 | 70.5 | 168.3 | 74.3 | 103.6 | 74.7 | 146.6 | 68.0 | 74.3 | 61.1 | 38.2 |
| | 3.0 | 42.4 | 64.2 | 189.7 | 72.0 | 164.1 | 63.4 | 222.2 | 67.7 | 100.9 | 60.8 | 55.3 |
| | 4.0 | 40.9 | 58.5 | 202.0 | 62.6 | 204.2 | 55.7 | 258.7 | 66.8 | 116.9 | 60.4 | 74.8 |
| | 5.0 | 40.9 | 55.4 | 208.1 | 55.5 | 231.6 | 52.9 | 276.5 | 65.1 | 127.6 | 60.0 | 94.8 |
| LEACE | 1.0 | 57.0 | 76.1 | 18.2 | 74.1 | 6.7 | 79.0 | 13.9 | 68.1 | 17.3 | 60.3 | 9.1 |
| | 2.0 | 44.8 | 76.2 | 30.5 | 74.1 | 11.7 | 78.9 | 19.3 | 68.1 | 25.3 | 60.2 | 13.6 |
| | 3.0 | 41.6 | 76.1 | 49.0 | 74.2 | 16.4 | 75.2 | 200.5 | 68.0 | 32.2 | 60.2 | 16.8 |
| | 4.0 | 40.9 | 75.6 | 73.4 | 74.2 | 21.3 | 78.7 | 29.0 | 68.0 | 38.0 | 60.1 | 19.5 |
| | 5.0 | 41.4 | 74.4 | 109.4 | 74.2 | 26.1 | 78.7 | 34.1 | 68.1 | 44.1 | 60.0 | 22.0 |

Table 25: Model SANA, removal of chihuahua

| method | strength | chihuahua cs | muffin cs | fid | dog cs | fid | wolf cs | fid | cat cs | fid | legislator cs | fid |
|---|---|---|---|---|---|---|---|---|---|---|---|---|
| No Steering | - | 76.4 | 66.3 | - | 68.1 | - | 73.2 | - | 68.5 | - | 60.4 | - |
| CASteer | 1.0 | 75.6 | 66.6 | 19.8 | 67.4 | 49.4 | 73.6 | 25.6 | 68.3 | 32.5 | 55.3 | 268.2 |
| | 2.0 | 49.5 | 66.8 | 30.8 | 59.9 | 143.6 | 73.4 | 53.4 | 66.4 | 65.2 | 60.5 | 33.7 |
| | 3.0 | 48.9 | 67.1 | 44.1 | 52.6 | 214.4 | 64.6 | 265.3 | 58.6 | 151.2 | 60.4 | 48.6 |
| | 4.0 | 49.5 | 67.0 | 58.0 | 52.3 | 223.8 | 62.0 | 263.8 | 54.6 | 205.0 | 60.2 | 71.9 |
| | 5.0 | 50.4 | 66.3 | 76.7 | 53.2 | 233.5 | 59.8 | 282.5 | 53.6 | 220.9 | 59.8 | 102.3 |
| LEACE | 1.0 | 73.0 | 66.3 | 5.8 | 68.0 | 28.1 | 73.2 | 6.4 | 68.5 | 10.4 | 60.4 | 9.9 |
| | 2.0 | 49.3 | 66.2 | 9.7 | 67.8 | 47.1 | 73.3 | 9.7 | 68.5 | 16.0 | 60.4 | 14.7 |
| | 3.0 | 47.3 | 66.2 | 12.6 | 67.6 | 68.1 | 73.3 | 12.3 | 68.6 | 20.6 | 60.4 | 17.9 |
| | 4.0 | 47.2 | 66.1 | 15.2 | 67.1 | 88.7 | 73.3 | 14.7 | 68.6 | 24.6 | 60.3 | 20.9 |
| | 5.0 | 48.5 | 66.1 | 17.7 | 66.2 | 113.6 | 73.3 | 16.8 | 68.7 | 27.5 | 60.3 | 23.0 |

