# OpenReview forum: "Optimal Affine Framework for Steering Generative Models"
_ICLR.cc/2026/Conference — Submitted to ICLR 2026_

### Official Review · Reviewer_MrFE · 2025-10-21

**Soundness:** 2
**Presentation:** 1
**Contribution:** 3
**Rating:** 2
**Confidence:** 3

**Summary:**

The paper proposes a theoretical connection between affine concept erasure and concept steering, and then adapts affine erasure to the task of concept switching. In addition, a new approach, MiDSteer is proposed, and experiments are conducted to compare it to the prior techniques in both LLMs and generative image models.

**Strengths:**

The proposed technique makes sense and the results show the improvement versus the state of the art techniques.

The theoretical bridging of the different approaches is a sdolid contribution.

**Weaknesses:**

My score is currently low because of the, to me, confusing presentation. If this could be rectified, then the score could improve. For example, it is unclear what LEACE is. It is not well introduced, and we have to infer what it is. Similarly, CASteer is mentioned just above equation (18), but it is not clear what it is and how it differs to LEACE. Even MidSteer itself is not well introduced. There is a tangential definition of it at line 310, which refers back to Thm 5.

The results in section 4.2.1 experiments are for CASteer and LEACE. These are existing techniques and not the (new) MiDSteer. Why is this included?

Minor:
- "subscript" of $s^c^ should be "superscript"
- miscellaneous grammatical issues, e.g. "of the concept C in generation result of the model"
- Figure 1 is not referenced in the text as far as I could see.
- there seem to be two betas. One in line 322 and a different one (?) in eqn (20). That is confusing. (and the reference to eqn (21) in line 323 should presumably to to eqn (20))
- the heat map colour for beta in the figures does not work well
- in line 471, should "4" be "Fig 4"?

**Questions:**

Is the constraint in Theorem 5 on the rank being full satisfied in practise? Was that investigated? If it is not, how does it impact the results?

In section 4.2.1, unrelated concepts are used for testing. How is the lack of relation to the erased concept established? And would it be interesting to investigated related concepts vs unrelated concepts to see the impact of erasure on the different degrees of relatedness?

---

> ### Author Response · Authors · 2025-11-26
> **Rebuttal to Reviewer MrFE**
>
> We thank the reviewer for praising our **solid theoretical contribution** and **improvements over SOTA**. Below, we respond to the main issues the reviewer raised.
>
> **1.On the confusion of presentation.**
>
> We thank the reviewer for pointing out the lack of clear introductions to LEACE, CASteer, and MidSteer. We have substantially revised the presentation to address this.
>
> First, we now introduce **CASteer** and **LEACE** early and explicitly, including clear definitions, intuition, and their role in prior work. Although both were originally proposed for **concept erasure**, they are, to our knowledge, the only steering-based **affine** methods with closed-form solutions for concept manipulation in generative models. They are widely used for activation-level control in both LLMs and diffusion models.
>
> We also clarify how they differ:
> - **CASteer** (Eq. 3–4) applies a Householder reflection—a symmetric concept flip—which makes it the closest existing mechanism to switching.
> - **LEACE** (Eq. 6–7) is the optimal affine operator for removing covariance with a concept, forming the strongest principled baseline for erasure or replacement.
>
> We then introduce **MidSteer** cleanly and independently, rather than referencing it tangentially. The revised Section 4.3 explains that MidSteer is a **general affine framework** that unifies erasure, flipping, and switching. CASteer and LEACE appear as special cases under this unified view, which justifies using them as baselines. Section 4.2.1 further highlights why erasure-only or symmetric methods are insufficient for **directed** concept switching.
>
> These revisions aim to make the presentation clear, self-contained, and easier to follow, resolving the confusion raised by the reviewer.
>
> **2.On the experiments of CASteer and LEACE.**
>
> We have updated the experiments section to focus exclusively on switching results.
>
> Note, however, that LEACE is a special case of MidSteer when MidSteer is applied to erasure. As shown theoretically in Sec. 4.3 of the updated paper, improvements of LEACE over baselines in the erasure setting therefore also imply corresponding advantages for MidSteer. We have updated the erasure plots to reflect this relationship.
>
> Our earlier version included erasure experiments comparing Vanilla Steering and LEACE to highlight LEACE’s superiority over vanilla steering (CASteer), important because vanilla steering is widely used for erasure tasks, whereas LEACE remains underrepresented in the literature.
>
> **3.Minor issues.**
>
> We have largely resolved the minor issues in the updated manuscript.
>
> **Q1.Is the constraint in Theorem 5 on the rank being full satisfied in practise? Was that investigated? If it is not, how does it impact the results?**
>
> In Theorem 5 (Theorem 7 in the updated paper), the requirement that $\Sigma\_{XZ\_1}$ be full rank is necessary for the proof: it allows us to use the closed-form expression for the Moore–Penrose pseudoinverse in the full-rank case, enabling the construction of $\Lambda\_W$.
>
> Practically, this condition means that the source concepts $c\_i$ to be switched must not be linearly dependent. For example, if $c\_1 = c_2 + c\_3$, with $c_2$ and $c_3$ being mutually exclusive , then $Z\_1$ = { $c\_1$, $c\_2$, $c\_3$} has rank at most 2.
>
> In our experiments, we manipulate a single source concept into a single target concept. Under this setting, the full-rank condition reduces to requiring that the concept being steered is non-constant (i.e., not always present or always absent in the dataset). This condition is satisfied in all experimental setups.
>
> **Q2.In section 4.2.1, unrelated concepts are used for testing. How is the lack of relation to the erased concept established? And would it be interesting to investigated related concepts vs unrelated concepts to see the impact of erasure on the different degrees of relatedness?**
>
> For each source concept $c\_s$, we selected three semantically related concepts $c\_i$ and one unrelated concept (“legislator’’ in all cases). The related concepts are chosen so that they should remain unaffected by concept switching while still being meaningfully connected to $c\_s$. For example, for $c\_s$=“horse,’’ the related concepts are other four-legged animals (“dog,” “pig,” “cow’’).
>
> Appendix Sec. 7.9 provides a detailed breakdown of CS/BERT/FID scores for each concept, allowing the reader to assess how switching affects concepts with different levels of semantic relatedness.

---

> > ### Comment · Reviewer_MrFE · 2025-11-27
> >
> > Thank you for your response. The presentation is now a lot clearer and that certainly helps.
> >
> > But now I have a few queries/concerns arising from this, again probably from a lack of understanding of the approach
> > - on line 51, you mention concept "addition". I am not sure this is defined, and does not seem to appear later in the paper anyway
> > - Around line 109, you talk of "targeted steering". It is not clear if this is distinct from steering in general
> > - Concept switching is mentioned in line 110 and yet flipping is mentioned in 109 - related and yet different concepts. I think more clarity is needed. e.g. equation (4) to me seems associated with a concept flip rather than a switch between two concepts and yet the function is referred to as f_{switch}. On that, I am lacking intuition over exactly what concept flipping corresponds to e.g. I understand what it might mean for a concept, but not for an object.
> > - Further to the point above, in the experiments you switch a horse for a motorcycle. Including with the existing techniques. Yet I thought the argument was that they (the older techniques) were only for erasure or flipping, not switching.
> >
> > Minor query:
> > - On line 156, you refer to Z=j, and yet Z takes values in a k-dimensional binary space. What is j?

---

> ### Author Response · Authors · 2025-11-27
> **Answer to the new questions**
>
> We thank the reviewer for the new questions. We have further updated the pdf to solve these concerns.
>
> **'Addition' and 'targeted steering'**
>
> We thank the reviewer for the helpful comment. We removed the reference to “addition” in line 51 and clarified the ambiguous phrase “target steering” in line 109. Although “steering” is often used in the literature to denote adding concepts or behaviours to models, our work focuses exclusively on erasure and switching. Accordingly, we revised line 51 and Section 3.1 to restrict the discussion to these forms of concept manipulation.
>
> **Concept switching is mentioned in line 110 and yet flipping is mentioned in 109 - related and yet different concepts. I think more clarity is needed...**
>
> We thank the reviewer for highlighting the misalignment between “flipping” and “switching.” We updated line 110 to refer to switching rather than flipping. The initial draft used both terms, “flipping” for the LEACE formulation and “switching” for MidSteer, because Eq. 4 and Theorem 5 (LEACE-Switch) impose a bidirectional swap constraint ($c_1 \leftrightarrow c_2$), which can be interpreted as concept flipping. In the revised version, we consistently describe Eq. 4 and Theorem 5 as performing concept switching to avoid terminological confusion. As clarified in lines 259–265, our objective is concept switching: we first examine how vanilla steering (Eq. 4) and the LEACE-Switch extension (Theorem 5) address this task, noting their limitations, and then introduce MidSteer, which is explicitly designed for precise concept switching.
>
> **Further to the point above, in the experiments you switch a horse for a motorcycle. Including with the existing techniques. Yet I thought the argument was that they (the older techniques) were only for erasure or flipping, not switching.**
>
> As stated in Sec. 3.1, vanilla steering (e.g., CASteer for images) supports both concept erasure (Eq. 3) and concept switching (Eq. 4). LEACE was originally proposed only for concept erasure. In this paper, we extend LEACE to the switching setting, deriving LEACE-Switch (Theorem 5, Sec. 4.2.1). We then analyze its limitations (L316–L333) and introduce MidSteer (Theorem 7, Sec. 4.3), a framework designed to address these issues. In our experiments, we compare all three methods for concept switching: vanilla steering, LEACE-Switch, and MidSteer, and show that MidSteer resolves the limitations of LEACE-Switch and outperforms both alternatives, achieving equal or higher switching accuracy while minimally altering model representations.
>
> **On line 156, you refer to Z=j, and yet Z takes values in a k-dimensional binary space. What is j?**
>
> Z=j refers to Z being a one-hot vector of size k having 1 at index j. We borrowed the definition of guardedness and notation from LEACE.

---

> > ### Comment · Reviewer_MrFE · 2025-11-27
> >
> > Thank you for your response.
> >
> > In the paragraph at line 280 onwards, the switch from Z to 1^k-Z is said to represent the switch from c_1 to c_2. How does this work ie how is Z related to BOTH c_1 andf c_2. Are the concepts required to be related by being mutual inverses? If so, how does that work in your horse-motorcycle example?

---

> > > ### Author Response · Authors · 2025-11-28
> > > **Response to the new question**
> > >
> > > Yes. For LEACE-Switch (Sec. 4.2.1), constraint (9) assumes that the binary concept variable $Z$ partitions the dataset, i.e., $(P(Z=1) + P(Z=0) = 1)$ when switching between two concepts. Under this assumption, LEACE-Switch has a theoretical guarantee that an optimal transformation exists. We note this in the “Limitations” section of Sec. 4.2.1 (L316–L332). In practice, however, LEACE-Switch can still be applied when $(P(Z=1) + P(Z=0) ≠ 1)$ by estimating $Z_1$ and $Z_2$ for $c_1$ and $c_2$ and substituting 1 - $Z_1$ with $Z_2$. Our experiments confirm that LEACE-Switch works empirically for distinct concepts $c_1$ and $c_2$, though it exhibits the limitations predicted by the theory.
> > >
> > > Given these limitations, we introduce a new constraint for switching between two concepts $c_1$ and $c_2$, formulated in Sec. 4.3, and derive MidSteer (Theorem 7), a new theoretical solution for concept switching.
> > >
> > > We include LEACE-Switch in the paper for two reasons. First, it is the natural generalization of LEACE (originally for erasure) to the switching setting. In Theorem 6, we show that vanilla steering for switching (Eq. 4) arises as a special case of LEACE-Switch, just as vanilla steering for erasure (Eq. 3) is a special case of LEACE for erasure. This highlights why the limitations of LEACE-Switch motivate the need for a different framework, MidSteer.
> > >
> > > Second, there are practical scenarios where the assumption $P(Z=1) + P(Z=0) = 1$ holds and a bidirectional swap $c_1 \leftrightarrow c_2$ is required, for instance, dataset augmentation procedures that create controlled paired samples (e.g., male ↔ female face transformations). In such settings, LEACE-Switch remains a valid and usable method.

---

> ### Comment · Reviewer_MrFE · 2025-11-28
>
> Thanks. That explanation may be good to incorporate in the paper, e.g. rather than introduce the argument of 4.2.1 and then note at the end that it relies on such an assumption, make it clear from the start. Otherwise it, at least to me, is confusing as to how there are two concepts present but a single Z.
>
> Thanks for the discussion and updates to the paper. In view of the changes and improvements I shall consider revising.

---

> > ### Author Response · Authors · 2025-11-28
> > **Thank you**
> >
> > We thank the reviewer again for their constructive criticism during the entire review process. We genuinely believe that this helped us significantly improve the paper.
> >
> > We are encouraged to see that the reviewer is considering revising their score.

---

### Official Review · Reviewer_bmDB · 2025-11-02

**Soundness:** 2
**Presentation:** 1
**Contribution:** 1
**Rating:** 2
**Confidence:** 4

**Summary:**

Topic: Steering intermediate representations of generative models

Definition of steering: adding a steering vector to the intermediate representations to control the generated results.
* concept deletion = Erasing unwanted behaviors or concepts
* concept switching = changing a concept with another

Problem statement:
* Steering is empirically developed without theory.
* Naive steering often perturbs unrelated features.
* Affine concept erasure does not solve concept addition or switching.

Contribution:
* a theory of concept steering: steering is a special case of LEACE, a closed form method for affine concept erasing in neural networks
* Minimal disturbance concept steering (MiDSteer), an improved version of concept steering

**Strengths:**

Originality: not sure

Quality:
1. L329 The experiments cover various models: Llama 2, Qwen 2.5, SDXL, and SANA.

Clarity:
1. L353, L371, L400 The task and desiderata are clearly noted.

Significance: not sure

**Weaknesses:**

1. The previous knowledge and the proposed knowledge should be separated. Which parts are the contribution? It would help the readers recognize the significance of this paper.
2. Significance of the contribution should be apparent to the readers. Why are the theorems and proofs non-obvious?
3. L171 Guardedness should be defined in the paper for integrity, even though it is cited. The connection between guardedness and Theorem 1 should be explained. Please be kind to the readers.
4. L310 MiDSteer, the proposed method, is not proposed.
5. LEASE should be L054 cited and L196 explained.
6. Figure 4: The images and the caption do not match: the dog is still there in CASteer
7. Sentences should be easier to be understood. Please remove redundant or uninformative words and write precisely.

**Questions:**

My questions are apparent in the weaknesses. Resolving them in the rebuttal may improve my rating.

---

> ### Author Response · Authors · 2025-11-26
>
> We thank the reviewer for praising our **experiments** and **the desiderata**. We genuinely thank the reviewer for their suggestion in improving the presentation of the paper. We have followed their advise, substantially rewriting the paper.
>
> **1.Prior Knowledge and contributions should be separated.**
>
> We thank the reviewer for pointing out the lack of clear introductions to LEACE, CASteer, and MidSteer. We have substantially revised the presentation to address this.
>
> First, we now introduce **CASteer** and **LEACE** early and explicitly, including clear definitions, intuition, and their role in prior work. Although both were originally proposed for **concept erasure**, they are, to our knowledge, the only steering-based **affine** methods with closed-form solutions for concept manipulation in generative models. They are widely used for activation-level control in both LLMs and diffusion models.
>
> We also clarify how they differ:
> - **CASteer** (Eq. 3–4) applies a Householder reflection—a symmetric concept flip—which makes it the closest existing mechanism to switching.
> - **LEACE** (Eq. 6–7) is the optimal affine operator for removing covariance with a concept, forming the strongest principled baseline for erasure or replacement.
>
> We then introduce **MidSteer** cleanly and independently, rather than referencing it tangentially. The revised Section 4.3 explains that MidSteer is a **general affine framework** that unifies erasure, flipping, and switching. CASteer and LEACE appear as special cases under this unified view, which justifies using them as baselines. Section 4.2.1 further highlights why erasure-only or symmetric methods are insufficient for **directed** concept switching.
>
> These revisions aim to make the presentation clear, self-contained, and easier to follow, resolving the confusion raised by the reviewer.
>
> **2.Why are the theorems non-obvious?**
>
> As can be seen from the theorem proofs they are hardly trivial. Our work is a theoretical work bridging the gap between optimal affine methods such as LEACE and steering methods such as CASteer. Obviously, to show this theoretical connections we needed to mathematically prove it.
>
> **3.Guardness should be defined?**
>
> Thanks for raising this issues. We have done so in the updated version (Definition 1, lines 154-173).
>
> **4-5.MidSteer should be better proposed, LEACE should be cited and explained.**
>
> We thank the reviewer for the helpful feedback. We have substantially revised the paper to improve clarity and readability. The former “Methodology’’ section is now split into **Preliminaries** and **Theoretical Results**. In *Preliminaries*, we provide a clearer introduction to affine guardedness and reference LEACE as Theorem 3. We also explicitly introduce MidSteer and reference it as Theorem 7.
>
> **6.Figure 4: The images and the caption do not match: the dog is still there in CASteer**
>
> While the dog remains present, it has significantly changed, which is not desirable. Only the horses should change to motorcycles, the other concepts should remain unchanged (as it happens in MidSteer).
>
> **7.Sentences should be easier to be understood. Please remove redundant or uninformative words and write precisely.**
>
> We have done this in the revised version.

---

### Official Review · Reviewer_7vKT · 2025-11-03

**Soundness:** 2
**Presentation:** 2
**Contribution:** 2
**Rating:** 4
**Confidence:** 3

**Summary:**

The authors provide a theoretical connection between representation steering affine concept erasure/switching. In particular, they first show that concept steering is a special case of LEACE, which is a closed-form method for affine concept erasure. Next, the authors study the task of concept switching. By arguing that vanilla concept steering can switch the two targeted concepts, e.g. untruthfulness to truthfulness and vice versa, they propose MIDSteer to only allow one way mapping.

**Strengths:**

* The paper is generally well-written.
* I think the problem of concept switching is a more controlled variant of concept erasure, which is nice.
* Empirical results for LLMs show superior concept switching results compared to other methods.

**Weaknesses:**

* The result section on LLMs seem to lack qualitative results, while the result section on diffusion models lack quantitative results.
* While MIDSteer perform better than other methods for concept steering, I am not sure if CASteer and LEACE are appropriate baselines since they are purely designed for concept erasure.
* I think the paper would benefit from more justifications on why concept switch is an interesting problem or why is it more preferred than concept erasure.

**Questions:**

* Can the authors provide some qualitative results on LLMs output when MIDSteer, CASteer, and LEACE are applied?
* What concepts are used to measure erasure for the LLM experiments?
* Can the authors provide Pareto efficiency frontiers plot for the SDXL experiments?
* What is the purpose of Section 4.2.1, I do not see MIDSteer being compared for concept erasure, yet the authors are comparing it with the same methods for concept switching?
* For concept switching, I would be interested to see the Pareto plot using the y-axis as the concept score on c2 and on c1 separately, rather than just the difference between concept scores of c2 and c1.

---

> ### Author Response · Authors · 2025-11-26
> **Rebuttal to Reviewer 7vKT**
>
> We thank the reviewer for calling our paper **well-written**, and for praising the **superiority** of our results. Below, we answer to the reviewer's concerns.
>
> **1.Qualitative results in LLMs and quantitative results in diffusion.**
>
> We thank the reviewer for raising this. We updated Sec. 5.2 to include quantitative results for both LLMs and diffusion models. Note that more quantitative results are presented in Sec. 7.9 in the appendix. We also added example of qualitative results on LLM in the appendix Sec. 7.3
>
> **2.CASteer and LEACE are erasure methods.**
>
> We agree that CASteer and LEACE were introduced for concept erasure rather than switching. However, to our knowledge, no prior work explicitly studies concept switching. CASteer and LEACE remain the only steering-based affine methods with closed-form solutions for concept manipulation in generative models, and they are widely used in activation-level control for both LLMs and diffusion models. Our goal is to unify erasure, flipping, and switching within a single affine framework. Under this view, CASteer (Eq. 3–4) and LEACE (Eq. 6–7) are special cases of our general affine operators and therefore constitute the most natural baselines.
>
> Concretely:
> - CASteer’s “switching’’ rule (Eq. 4) is a Householder reflection, i.e., a symmetric concept flip, making it the closest existing mechanism to switching.
> - LEACE is the optimal affine operator for removing covariance with a concept, providing the strongest principled baseline for erasure or replacement.
>
> Because our contribution generalizes affine erasure methods to directed concept switching, comparisons to CASteer and LEACE are methodologically appropriate and highlight the limitations of symmetric flipping and erasure for directional tasks.
>
> In the revised version, Sec. 4.3 clarifies that MidSteer subsumes erasure, flipping, and switching, and Sec. 4.2.1 explains why erasure-only methods fail in switching settings: they impose symmetric transformations that do not distinguish forward from reverse direction, a limitation MidSteer resolves by design.
>
> **3.Why switching is preferred to erasure?**
>
> First, we show in Sec. 4.3 that concept erasure is a special case of concept switching: LEACE’s erasure operator arises directly as a restricted instance of our MidSteer framework. Thus, our formulation generalizes the erasure task.
>
> Many erasure settings can be recast as steering problems. For example, nudity removal in image generation can be expressed as steering from $c\_1$= “nudity’’ to $c\_2$ “in clothes.’’ Importantly, images containing $c\_2$ should not be mapped back to $c\_1$, making MidSteer preferable to LEACE.
>
> Concept switching also enables broader applications than erasure, such as:
> - **Dataset augmentation and domain adaptation:** e.g., controlled male ↔ female face transformations or generating paired samples for supervised tasks.
> - **Style transfer:** e.g., academic ↔ casual, dry ↔ humorous, bright ↔ dark.
> - **Disambiguation:** e.g., steering “the city’’ → “Paris.’’
>
> Unlike vanilla steering, MidSteer preserves non-target semantics (Sec. 4.2.1), which is valuable when generating samples that differ in only one attribute. The asymmetric property of MidSteer, allowing $c\_1$ to $c\_2$ but not $c\_2$ to $c\_1$ is beneficial in these settings. For augmentation, one may want to modify only a specific entity (e.g., one face in an image) without altering others that already exhibit $c\_2$. For style transfer, converting academic → casual should not cause initially casual texts to flip to academic.
>
> We also hypothesize that concept steering could support fact editing in LLMs, though this requires additional structure and is left for future work.
>
> **Q1.Can the authors provide some qualitative results on LLMs output when MIDSteer, CASteer, and LEACE are applied?**
>
> We have added example of qualitative results for LLM in the appendix, Sec. 7.3
>
> **Q2.What concepts are used to measure erasure for the LLM experiments?**
>
> We thank the reviewer for pointing to this. We have added a description of the prompts used for concept switching to sec.5.1 of the paper, and a description of the prompts used for concept erasure to sec.7.10 of the paper
> We test erasure of the following concepts $c\_1$="Horse", $c\_2$ ="Dog", $c\_3$ ="Chihuahua". Corresponding testing concepts that we measure the difference in LLM/diffusion model generation on are $t\_1$=("Motorcycle", "Cow",  "Dog", "Pig", "Legislator"), $t\_2$=("Cat", "Cow",  "Wolf", "Pig", "Legislator"), $t\_3$=("Muffin", "Cat",  "Dog", "Wolf", "Legislator").
>
> **Q3.Can the authors provide Pareto efficiency frontiers plot for the SDXL experiments?**
>
> We added Pareto efficiency frontier for SDXL in Sec. 5.2. More plots can be found in the appendix Sec. 7.9.

---

> > ### Author Response · Authors · 2025-11-26
> > **Continuing the rebuttal**
> >
> > **Q4.What is the purpose of Section 4.2.1, I do not see MIDSteer being compared for concept erasure, yet the authors are comparing it with the same methods for concept switching?**
> >
> > We have updated the experiments section to focus exclusively on switching results.
> >
> > Note, however, that LEACE is a special case of MidSteer when MidSteer is applied to erasure. As shown theoretically in Sec. 4.3 of the updated paper, improvements of LEACE over baselines in the erasure setting therefore also imply corresponding advantages for MidSteer. We have updated the erasure plots to reflect this relationship.
> >
> > Our earlier version included erasure experiments comparing Vanilla Steering and LEACE to highlight LEACE’s superiority over vanilla steering (CASteer), important because vanilla steering is widely used for erasure tasks, whereas LEACE remains underrepresented in the literature.
> >
> > **Q4. For concept switching, I would be interested to see the Pareto plot using the y-axis as the concept score on c2 and on c1 separately, rather than just the difference between concept scores of c2 and c1.**
> >
> > We provided additional Pareto plots (in 9 cases) with the y-axis showing concept scores for $c\_2$ and $c\_1$ separately in Appendix Sec. 7.9.

---

### Official Review · Reviewer_1GAs · 2025-11-04

**Soundness:** 3
**Presentation:** 2
**Contribution:** 2
**Rating:** 6
**Confidence:** 2

**Summary:**

This paper introduces a unified theoretical framework for optimizing affine mappings for concept steering in neural representations. They show that the existing approach for concept deletion and flipping can be viewed as special-case solutions for a general constrained optimization problem. Then, they propose MidSteer, a general framework for concept steering. The main novelty compared with the prior works is that they whitens activations and apply steering transformation on the standardized representation space.

**Strengths:**

The work elegantly connects existing techniques for concept deletion, flipping, and transfer within a single constrained-optimization framework, offering novel insights and guiding future work. MidSteer generalizes prior works by removing the assumption of standardized activations, making the framework applicable to real, anisotropic model embeddings.

**Weaknesses:**

From an implementation standpoint, MidSteer reuses the same affine transformation derived in earlier work, with $\beta$ now treated as a hyperparameter, and the only non-trivial technical novelty seems to be activation standardization. Similarly, the theoretical contributions, though sound, follow relatively directly from existing frameworks without introducing substantially new analytical insights. Happy to be corrected on this.

The presentation of the final algorithm can be improved. I could not find any discussion of the computational cost of estimating $\Sigma_{X, X}$. I suggest that the authors add the pseudo-code for this algorithm. What will the dimension of $X$ be for the language model? Is it seq_len x hidden_dim? Would the algorithm require instantiating the whole matrix of $\Sigma_{X, X}$?

I do not directly work in this field, so I cannot comment on how significant the experiment results are. Therefore, I set my confidence to 2.

**Questions:**

See weakness.

---

> ### Author Response · Authors · 2025-11-26
>
> We thank the reviewer for calling our method **elegant**, offering **novel insights**, **guidance for future work**, while being applicable to **real-world** scenarios. Below, we address the issues the reviewer raised.
>
> **1.On better explaining the theoretical novelty.**
>
> We thank the reviewer for the comment. **Theorem 5 (LEACE-Switch)** is derived from Bellrose et al. (Theorem 3), but it has two limitations (L316–L333):
>
> 1.  It performs a bidirectional swap ($c_1 \leftrightarrow c_2$), which is often undesirable.
> 2.  It assumes $c_1$ and $c_2$ span the dataset, which rarely holds.
>
> We therefore formalize a more general task: map $c_1 \to c_2$ while leaving $c_2$ unchanged, without requiring span coverage. Section 4.3 (**Theorem 7**) introduces **MidSteer**, an affine solution that uses separate cross-covariance matrices for $c_1$ and $c_2$. This produces a transformation distinct from LEACE, with **LEACE as a special case** (L361–L364). Experiments show MidSteer outperforms Theorem 5 (LEACE) and vanilla steering (CASteer).
>
> We have updated the manuscript accordingly and added Section 4 to clearly separate prior work from our contributions.
>
> **2.The presentation of the final algorithm should be improved.**
>
> To estimate the covariances we use the **Welford algorithm** on a broad sample of prompts unrelated to the steering concepts (see Sec. 5.1, “Details on LLM experiments”). We have added pseudocode for this algorithm to the appendix Sec. 7.1.
>
> Formally, given X with the dimension batch_size $\cdot$ seq_len $\cdot$ hidden_dim, the algorithm estimates the covariance matrix $\Sigma_{XX}$ of size hidden_dim $\cdot$ hidden_dim.
>
> 1. It does this by maintaining sample-level statistics of size O(hidden_dim^2) in memory.
> 2. It takes O(batch_size $\cdot$ seq_len $\cdot$ hidden_dim^2) time to update them for the output of a particular layer on a particular batch.
>
> In practice, estimating the covariances for **50,000 samples** for SANA 1.6 finished in **under 15 minutes**, and for Qwen2.5 14B in **under 30 minutes** on a single Nvidia H100.

---

### Author Response · Authors · 2025-11-26
**Global response**

Dear AC and reviewers,

We thank all the reviewers for their very constructive reviews, raising several issues with the paper. Below we summarize the main points we addressed.

1) Most of the concerns were about the presentation of the paper, and the lack of clear separation before prior art and our contributions. We have heavily updated the paper (please see the new pdf) to clearly separate these two things.

2) We have performed or pointed to all the extra required experiments asked by the reviewers.

---

### Meta-Review · Area_Chair_nXAY · 2025-12-11

**Summary:**

## Reviewer 1GAs

Limited contribution from an implementation standpoint, noting that MidSteer reuses an existing affine transformation and that activation standardization is the only non-trivial technical novelty. Theoretically, the contributions are seen as following relatively directly from LEACE without introducing substantially new analytical insights. The reviewer also raises concerns regarding the algorithm's presentation and the reporting of its computational cost. Finally, the reviewer acknowledges not working directly in this field, which limits their ability to assess the significance of the experimental results.

## Reviewer 7vKT

The reviewer is concerned about the lack of qualitative LLM results and quantitative diffusion model results. The reviewer questions the appropriateness of CASteer and LEACE as baselines for concept switching and suggests more justification for the importance of concept switching. Additionally, the reviewer requests qualitative LLM outputs, clarification on concepts used for LLM erasure, Pareto efficiency frontiers for SDXL, and a clearer purpose for Section 4.2.1. They also suggest separate Pareto plots for c1​ and c2​ concept scores.

## Reviewer bmDB

This reviewer primarily focuses on presentation and clarity, highlighting several weaknesses and offering explicit questions for the authors to address. Key criticisms include:
* Lack of Clarity on Contributions: Reviewer comments that the paper needs to clearly separate prior knowledge from novel contributions and explain the significance and non-obviousness of its theorems and proofs.
* Missing Definitions and Explanations: It indicates that terms like "Guardedness" require definition and connection to theorems, and proper citation and explanation of LEASE.
* Specific Errors: The reviewer notes that MidSteer is incorrectly stated as being proposed, and there's a mismatch between Figure 4's images and its caption (e.g., the dog still appearing in CASteer).
* General Writing Style: The paper advises that sentences should be clearer, more precise, and free of redundant words.

## Reviewer MrFE

The reviewer finds the presentation of the paper confusing, pinpointing to the null presentation of CASteer and its confusion with LEACE. Reviewer also complains about experiments on section 4.2.1 being for CASteer and LEACE instead for the proposed method MidSteer. Finally, the reviewer arises other minor errors that the authors claim to have revised. Additionally, both in the review and in during the rebuttal conversation, the reviewer asks several questions for clarification including the constraint in Theorem 5 on the rank being full satisfied in practice, or the fact that in the experiments authors compare with previous existing techniques focus for erasure or flipping instead of for switching, which is the tested technique, among others.

**Reviewer Concerns:**

## Reviewer 1GAs

The authors address the limited theoretical novelty by clarifying that MidSteer introduces separate cross-covariance matrices for c1​ and c2. However, they do not directly respond to the limited contribution from an implementation standpoint or the concern about not introducing substantially new analytical insights.

Regarding the algorithm's presentation, the authors add pseudocode in Section 7.1 and include approximations of the execution time. However,  the execution time is not well-defined in the paper (just approximates), and while the rebuttal specifies execution on a single Nvidia H100, this detail is not indicated in the manuscript.


## Reviewer 7vKT

The authors address the lack of quantitative diffusion model results by adding Appendix Section 7.9.2, which includes Pareto Charts for Image Diffusion Concept Switching. However, the authors' claim of clear superiority for MidSteer over other steering approaches is not always evident in some figures (e.g., Fig 32, Fig 33, Fig 40, Fig 41, Fig 42, Fig 43), a point that requires clarification. They also include a single qualitative example for LLMs in the Appendix, which only minimally addresses the reviewer's concern.

Regarding the appropriateness of CASteer and LEACE as baselines, the authors attempt to justify their use as the only steering-based affine methods for concept manipulation, claiming no prior work explicitly studies concept switching. However, existing literature shows other methods using affine techniques for concept manipulation (e.g. [ITI](https://arxiv.org/abs/2306.03341), [CAA](https://arxiv.org/abs/2312.06681), [LinEAS](https://arxiv.org/abs/2503.10679)).

Finally, while the authors justify the need for concept switching in the rebuttal, they do not include this justification in the manuscript. This means they do not fully address the reviewer's concern that the paper would benefit from more explanation on why concept switching is an interesting problem.


## Reviewer bmDB

The authors appear to have made a substantial effort to address the reviewer's concerns. They have revised the paper to introduce CASteer early, clearly, and separately from MidSteer. Regarding the non-obviousness of theorems and proofs, the authors partially address the concern, stating that the proofs are "hardly trivial" but do not elaborate further. The authors include the definition of "Guardedness," thus alleviating the reviewer's concern. They also provide an explanation for Figure 4 and claim to have removed redundant or uninformative words in the revised version, although wordy sentences can still be found in the manuscript.


## Reviewer MrFE

The authors addressed the reviewer's major concern regarding poor presentation by first introducing CASteer and LEACE early and explicitly providing clear definitions. Separately, the authors introduce MidSteer rather than referencing it tangentially. However, it should be noted that these changes resulted in significant revisions to the originally submitted paper.

The authors also changed the presented experiments to focus solely on concept switching, thereby better highlighting MidSteer's contributions. Regarding the minor concerns, most were fully addressed by the authors, while others were only partially addressed, such as the concern about comparing MidSteer to older techniques not originally designed for concept switching. Lastly, the reviewer suggested incorporating a clarification for the argument presented in Section 4.2.1 concerning multiple concepts within a single Z matrix; however, this clarification has not yet been introduced into the manuscript. Finally, the reviewer thanked the authors for the updates to the paper and suggested they would consider revising their score.

**Reviewer Scores:**

*   **Reviewer 1GAs**
    *   **Original Score:** 6.
    *   **Final Score**: 6.
    *   **Reason:** The authors partially addressed this reviewer's concerns (see above), and given this is already the highest score, I doubt this reviewer would increase it further.

*   **Reviewer 7vKT**
    *   **Original Score:** 4
    *   **Final Score:** 4.
    *   **Reason:** After the rebuttal there are still doubts about the superiority claim of MidSteer over other approaches and the choice of baselines.

*   **Reviewer bmDB**
    *   **Original Score:** 2
    *   **Final Score:** 4.
    *   **Reason:** The authors did a honest effort in improving the paper presentation.

*   **Reviewer MrFE**
    *   **Original Score:** 2
    *   **Final Score:** 4
    *   **Reason:** The authors did a honest effort in improving the paper presentation.

## Decision
Reject. I increased bmDB and MrFE's scores by one point to reflect the authors effort in improving clarity and presentation. However, even in that scenario, the overall score would tend towards rejection.

---

### Decision · Program_Chairs · 2026-01-26

Reject